# Preparations and Applications of MXene–Metal Composites: A Review

**Maaz Ullah Khan** [1] (ID), **LiJing Du** [1], **Shuai Fu** [2], **Detian Wan** [2], **Yiwang Bao** [2], **Qingguo Feng** [1], **Salvatore Grasso** [1] (ID) **and Chunfeng Hu** [1,*] (ID)

1   Key Laboratory of Advanced Technologies of Materials, Ministry of Education, School of Materials Science and Engineering, Southwest Jiaotong University, Chengdu 610031, China; engrmaaz213@gmail.com (M.U.K.); dulijing3933@163.com (L.D.); qfeng@swjtu.edu.cn (Q.F.); s.grasso@swjtu.edu.cn (S.G.)
2   State Key Laboratory of Green Building Materials, China Building Materials Academy, Beijing 100000, China; shuaifu19@163.com (S.F.); dtwan@ctc.ac.cn (D.W.); ywbao@ctc.ac.cn (Y.B.)
*   Correspondence: chfhu@live.cn

**Abstract:** MXene, an advanced family of 2D ceramic material resembling graphene, has had a considerable impact on the field of research because of its unique physiochemical properties. MXene has been synthesized by the selective etching of MAX via different techniques. However, with the passage of time, due to the need for further progress and improvement in MXene materials, ideas have turned toward composite fabrication, which has aided boosting the MXene composites regarding their properties and applications in various areas. Many review papers are published on MXene and their composites with polymer, carbon nanotube, graphene, other carbon, metal oxides and sulfides, etc., except metal composite, and such papers discuss these composites thoroughly. In this review article, we illustrate and explain the development of MXene-based metal composites. Furthermore, we highlight the synthesis techniques utilized for the preparation of MXene composites with metal. We briefly discuss the enhancement of properties of the composites and a wide range of applications as an electrode substance for energy storage devices, electrochemical cells, supercapacitors, and catalytic and anti-corrosive performance. Major obstacles in MXene and metal composite are mentioned and provide future recommendations. Together, they can overcome problems and enable MXene and composites on commercial-scale production.

**Keywords:** MXene; metals; composite; properties; applications

## 1. Introduction

The development of new advanced materials is highly demanded by the aerospace and automotive industries, which come up with weight savings, boost energy efficiency, resist severe structural loadings, and increase tribological performances. To accomplish such particular features, the material should have high specific strength, elastic modulus, and stiffness additional to improved functional characteristics [1]. Two-dimensional (2D) materials have been described as an acceptable constituent in future electrical properties, and their mechanical properties are entirely predominant for several approaches. The exploration of the mechanical properties and associated atomic structure are slightly troublesome because of their atomic breadth and planar character [2]. Two-dimensional materials mostly possessed a layered structure, which made them distinct from three-dimensional (3D) crystalline and one-dimensional (1D) nanowires. Two-dimensional substances are composed of covalent bonds in every layer and are packed cooperatively with van der Waals interactions. This crystal structure can provide these 2D substances with distinctive elastic, fracture, surface, and interfacial properties [3]. The developing method of two-dimensional (2D) graphene was examined to be complex and costly, which restricted its feasible practical implementations [4]. MXene, a newly produced class of 2D materials from Drexel University, has so far achieved superior consideration in the science world. Commonly, MXenes are

2D transition elements such as carbides, nitrides, or carbonitrides with a typical formula of $M_{n+1}X_nT_x$, where M represents transition element, X describes carbon (C) or nitride (N), and T show the combined activity of the active surface functional groups, such as O, OH and F [5]. MXene could readily be adopted by the selective etching of the "A" layer from the MAX phase, such as members from class 13 or 14 of the periodic table. Numerous experimental results show that the MAX phase can be etched as MXene only when A is Al atom. This is because the bonding force between Al atom and M atom and MX bond (a mixture of covalent bond, ionic bond and metal bond) is relatively weak, which provides the possibility of Al element spalling [6]. The etching agent used for the selective etching of the A atom layer in the MAX phase is hydrofluoric acid or lithium fluoride and hydrochloric acid mixed solution. Compared with these two methods, MXene laminates prepared by HF etching are clear and evenly spaced. However, HF is a highly corrosive acid, and MXene laminates prepared by HF etching often contain a certain amount of defects (such as holes), which will adversely affect the application of MXene [7]. The etching, as well as the manufacturing, of MXene conductive hydrogels involves both labor-intensive and time-consuming processes that cannot be scaled up. As a result, the difficulty of confirming long-term stability and large production limits the applications of MXene conductive hydrogels [8]. However, MXene is prepared by the method of lithium fluoride and hydrochloric acid. Although the lamellar morphology is not obvious and there are many small flake products attached to the surface of MXene, MXene nanosheets (less than 5 layers) of a high quality, high yield and large transverse size can be obtained in the following ultrasonic stripping. Therefore, this method is suitable for preparing flake MXene. The surface activity of MXene prepared by this liquid etching method is very high, and can react rapidly with water, fluorine ions, oxygen, and so on, in the solution to reduce the energy of the entire system. For instance, out of the several theoretically stated MXenes, the $Ti_3C_2$, $Ti_2C$, $Nb_2C$, $V_2C$, $Ti_3CN$, $Mo_2C$, and $Ta_4C_3$ members have been efficiently developed. Amongst them, $Ti_3C_2T_x$ is one of the most general and extensively deliberated MXenes. MXene has already displayed the surprising possibility of energy applications, principally as electrode materials in batteries and supercapacitors. The variable surface chemistry, graphene-like morphology, and redox ability with metal-like conductivity made MXene a promising 2D candidate for distinct approaches. Since MXenes comprise harmless still rich elements such as Ti, C and N with their degenerated products ($CO_2$ and $N_2$), which are also harmless, MXenes could further be employed for environmental applications [4,9].

MXene is a novel class of two-dimensional materials, which is generated by etching the Al layer of $Ti_3AlC_2$ MAX phase with HF solutions under gentle mode. MXene has achieved substantial consideration because of its superior hydrophilicity, physiochemical stability, electrical conductivity, and favorable environmental characteristics. It has been stated that, when MXene is employed as an assisting substrate, the properties of composites (containing electro-catalytic activity, phosphate removal, and peroxymonosulfate activation) enhance significantly. It was found that, in comparison with pure $Co_3O_4$, the sandwiched $Co_3O_4$/MXene composite exhibited superior catalytic activity for peroxymonosulfate activation to degenerate BPA, thus prompting that the use of MXene as a substrate can effectively increase the catalytic activity of active components. Therefore, it is anticipated that MXene could be used as a support of $Fe_2CoTi_3O_{10}$ for activation of peroxymonosulfate [10]. The adsorption of albumin, which staved off re-aggregation of the few-layer nanoplates, resulted in stable colloidal solutions after delamination of manifolded MXenes into minute fine nanoplates. Monodisperse colloids were created using cascading centrifugation, which can be used to synthesize MXenes for biomedical purposes. Albumin coated MXenes may find uses in a variety of disciplines, involving medicine, biology, pharmaceuticals, and environmental engineering, where protein adsorption upon nanomaterial planes performs a remarkable function [11].

The multi-ions were electrostatically intercalated into $Al^{3+}$ pre-intercalated $Ti_3C_2T_x$ MXene within the constrained area created by neighbouring MXene layers. These ions' intercalation can keep MXene's 2D feature and provide a way to adjust the interlayer envi-

ronment at the atomic level. Multi-ion interactions within the restricted MXene interlayer can create a steric impediment and electrostatic barrier to electrolyte ion transport and storage. The intercalated electrode design can be guided by understanding this mechanism and investigating the interaction between the multi-ion and MXene [12]. The adsorption efficiency of unmodified MXenes is heavily influenced by the development circumstances, as they play a key role in adjusting inter-layer spacing and specific surface area, both of which affect MXene chemistry for numerous metal ions. Surface functional group enhancement, specific surface area enlargement, structural stability enhancement, and, in many instances, surface charge protonation have all been used to increase MXenes' ability to remove heavy metals. Before adsorbent production, density functional theory calculations may aid in anticipating the adsorption procedure and adsorption reedition. Adsorption and interfacial chemical transformation are regarded to be the most prevalent adsorption operations for MXene adsorbents, although electrostatic attraction, surface complication, and ion exchange had the most familiar adsorption mechanisms for MXene [13]. The MXene/PPy complex grains seem to have an escalated adsorption selectivity, allowing them to detach methylene blue from a mixture while simultaneously blending cationic and anionic dyes. The MXene's stability is considerably improved, and the MXene/PPy compound grains exhibit almost no oxidation. This research demonstrates a new method for fabricating MXene-based adsorbents with excellent stability, and the composite particles show promise in effluent water cleanup [14]. Zhang, J. et al. studied the influence on electrical properties and structural changes caused by utilizing alkalization and calcination post-treatment in order to replace functional groups from the surface of MXene $Ti_3C_2$ nanosheets. The development of the interpolation of Na ions and the rise of layer spacing among 2D MXene $Ti_3C_2$ layers can be seen after eliminating F-modified groups by alkalization, and the final products remain as smooth and regular as nanosheets. The distinctive morphology of the 2D nanostructured materials imparts a large surface-to-volume ratio; consequently, the more active reaction sites on the surface and greater chemical activity are pivotal for feasible implementations such as catalytic, energy storage and electrode materials. At 400 and 600 °C, the XRD apex points remain nearly similar even without calcination, illustrating that the actual structure of the 2D MXene slab is still maintained after calcination. The high temperature treatment caused the little change, which is associated with a surface reaction of MXene $Ti_3C_2$ nanosheets, suggesting the elimination of the OH functional groups. Additionally, rising the calcination temperature to 800 °C, the enormous quantity of rutile phase and the small amount of anatase titania had been recognized in the XRD samples, implying that MXene $Ti_3C_2$ proceeded along with the oxygen contaminant in vacuum or O/OH groups ceased in the nanosheet surface. The conductivity of the MXene $Ti_3C_2$ after calcination at 400 °C is about 70% more compared to the sample without calcination. The calcination temperature rises to 600 °C, the conductivity improved further to 2410 S/cm, around three times greater than that of the MXene $Ti_3C_2$ without calcination (850 S/cm). Furthermore, the conductivity rate is considerably more than the previous report (1500 S/cm); thus, it could be attributed to diminish functional groups on the plane of the 2D nanosheets by calcination. Additionally, the evolution of small conduction trails imparts successfully to upgrade the electronic properties owing to the rise in compactness after calcination [15]. MXene nanoparticles placed on carbon fiber surfaces improved fiber surface energy, wettability, and surface roughness, resulting in significantly improved interfacial strength and flexural characteristics of fiber-resin [16].

$Mo_2C$ MXene exhibited a little molar volume, super-high electrical conductivity, approvable thermal conductivity, less thermal expansion coefficient and intense mechanical strength. It is suggested that $Mo_2C$ MXene possessed a wide range of applications, such as it could be employed as conductive material formed on its super-high electrical conductivity as well suitable thermal conductivity. MXene also has extreme conductivity and the structure strength to temperature difference and strains, enabled it to be employed as a substratum for additional surface systems. Furthermore, MXene with small molar volume, high adsorption capacity of ions and super-high conductivity made it a suitable material

utilized in batteries and supercapacitors application [17]. MXene and GO nanosheet enhance electrical and mechanical properties. Because of these favorable properties together with excellent electrochemical performance and flexibility, these new strands are suitable candidates for more progress of textile-based storage devices for wearable technologies and diminished electronics [18,19]. The work of Alhabeb, M. et al. showed that the MXene membranes exhibited the effectual Young's modulus of $333 \pm 30$ GPa, and the breaking strength of $17.3 \pm 1.6$ GPa. $Ti_3C2$ MXene possessed a Young's Modulus of 502 GPa according to the molecular dynamic simulations study. As anticipated, the analytically calculated value for the $Ti_3C_2T_x$ of $333 \pm 30$ GPa was smaller due to the presence of defects and surface functionalization. Furthermore, the graphene and GO (1050 GPa versus 210 GPa) showed no dramatic trend compared to the case of the divergence in the Young's moduli of the "ideal" $Ti_3C_2$ and the analytically perceived $Ti_3C_2T_x$. It was indicated that the mechanical properties of one-atom-thick monolayer graphene are greatly influenced by surface functionalization contrary to thicker $Ti_3C_2T_x$ flakes [20]. The properties of $Ti_3C_2T_x$ MXene films such as high strength and increased electrical conductivity were obtained by employing sizable MXene grains and a scalable blade coating operation. The results obtained for free-standing thin films were tensile strength with ~570 MPa, Young's modulus with ~20.6 GPa, and favourable electrical conductivity of ~15,100 S·cm$^{-1}$. The electromagnetic interference (EMI) shielding performance of these films was excellent [21]. Wang, Y. et al. investigated strong MXene sheets possessing high conductivity that were fabricated by the sequential bridging of hydrogen and ionic bonding. Interplanar spacing was reduced and MXene nanosheet alignment was enhanced through the ionic bonding agent. However, the hydrogen bonding agent aided in enhancing the interplanar sheet and reduced MXene nanosheet alignment. Both ionic and hydrogen bonding successive application optimized properties such as toughness, tensile strength, oxidation resistance in a wet circumstance, and resistance to sonication disintegration and mechanical exploitation [22]. It was suggested that $Nb_4C_3T_x$ MXene is a potential candidate for the primary component of structural composites, preventive coatings, membranes, textiles, and other applications. The mechanical properties of 2D $Nb_4C_3T_x$ MXene and 3D cubic NbC specified that the substantial empirical details on bulk carbides could be beneficial for recognizing novel MXene substances with enriched functioning behavior [23]. Luo, K. et al.'s targeted theoretical study was carried out on electrical and mechanical properties of MXene multilayer structures under strain modulation. In considering semi-conductors, it is crucial and effectual to control band gaps. The computation outcomes reveal that $Ti_2CF_2$ and $Ti_2C(OH)_2$ remain metallic under various strains, and so the oxygen ceased. $Ti_2C$ MXene might be acceptable for requisition in pressure sensing appliances. Moreover, strain–stress curves and electronic structures are calculated under strains. From the results, it governed the transformation from semiconductors to metals for $Hf_2CO_2$, and $Zr_2CO_2$ is governed to be permitted under compressive strains [24]. Mu, H. et al. studied the influence of etching temperature and ball milling on the development and capacitance of $Ti_3C_2$ MXene. $Ti_3C_2$ MXene capacitance was not affected by etching temperature, while the capacitance of MXene could significantly emhance via ball milling. The improvement of behavior is assigned to excessive carbon content for superior conductivity and the rapid transfer of electrons, as well as a broad surface area for more approach of aqueous electrolyte to the electrodes, collaboratively [25]. MXene nanofiller surface energies have a significant impact on the mechanical strength and reliability of each polymer-based appliance and compound to which they are implemented. According to contact-angle measurements, the surface energy values of 10-layer MXene coatings range between 47.98 and 64.48 mJ/m$^2$. The quantity of coating layers and liquids utilized had an effect on the wettability qualities. In comparison to a pure MXene coating, the surface energy of epoxy with 1 wt.% arbitrarily scattered MXenes enhanced. The interfacial adherence among the MXene grains and the epoxy is large, following the function of adhesion values (92.14–123.6 mJ/m$^2$). The findings in this paper imply that adding MXene to MXene–epoxy-fibre complexes, MXene–polymer

grazing, and polymer-based sensors might improve their mechanical properties, which can be used in a variety of multifunctional applications [26].

MXene's high-density oxygen functional groups, which are essential for initiating ZIF-8 crystal nucleation, were fused into a continuous layer on the surface. Even after 4 days of intense oxidation at 85 °C and 85% RH, ZIF-8/Ti$_3$C$_2$T$_x$ MXene preserved 98% of its initial electromagnetic interference shielding efficiency. The improved stability was due to ZIF-8's hydrophobic microporous structure, which effectively prevented water molecules from permeating while also ending MXene's dangling connections with Zn ions [27]. When the filler concentration was just 2 wt.%, the Ti$_3$C$_2$T$_x$ MXene/polybenzoxazine (PBZ) composites demonstrated good comprehensive properties, with a 22.1% increase in tensile strength, a 50% increase in impact strength, and a 67.3% increase in thermal conductivity. This research shows how to scatter hydrophilic nanosheets in a hydrophobic polymeric matrix using H–bonds for the creation of nanocomposites with improved characteristics [28]. The electrical conductance of the resultant nanostructure had significantly boosted, and the mechanical characteristics (e.g., tensile strength and tangent modulus) showed substantially upgraded synergism, thanks to the bridging and interrelated assemblies of the hybrid fillers. In addition, the network established between the hybrid fillers and the elastomeric macromolecules [29].

Manufacturing Ti$_3$C$_2$T$_x$ MXene–epoxy composites and examining their structure and fracture surfaces confirmed the modeling results. The binding energies of MXene and epoxy are essentially unaffected by MXene type (Ti$_2$CT$_x$ or Ti$_3$C$_2$T$_x$). Due to an increase in favourable electrostatic interactions, the bond between Ti$_3$C$_2$T$_x$ and epoxy gets stronger when the hydrogen coverage of the Ti$_3$C$_2$T$_x$ surface decreases. MXene–epoxy composites have a higher Young's modulus than plain epoxy, which results from stress transfer between the matrix and the nanofiller; the modulus increases linearly with filler loading up to 1 vol.%. Due to filler aggregation, the modulus increases less as the filler content increases. Both experimental and computational analyses of the fracture surfaces revealed void formation near the margins of the particles in MXene–epoxy composites during strain. Based on these findings, we predict that MXene fillers will boost epoxy toughness and mechanical behavior [30]. The etching method of the MAX phase Ti$_3$AlC$_2$ by the mixture of HCl and NH$_4$F strengthens in the generation of MXene Ti$_3$C$_2$. It remarkably affected this operation through two explored factors: the content of the hydrofluoric acid salt and the treatment time of the MAX phase in solution with the etching agent. The analysis indicated that the often-suitable conditions are: concentration of NH$_4$F in solution of 3 M and the treatment time of 160 h. The fascinating observation is that an increase in the duration of the process and concentration of the etching agent creates a decline in the intensity of the (002) reflection analogous to MXene. Perhaps this influence is created by the slight decomposition of laminated slabs of the MXene phase into individual Ti$_3$C$_2$ sheets [31].

The impact of several etching agents was examined on the exterior chemistries of Ti$_3$C$_2$T$_x$ multi-layers derived from the similar MAX phase group. They decided to focus on three etching agents, HF, LiF/HCl and FeF$_3$/HCl, all of which were under both normal and severe circumstances, as these give very dissimilar surface chemistries.

(1) Differing the concentration of HF confesses for varying the F content and thus the capacity to insert water. It manifested that less HF concentration allows for the introduction of water layers, apparently balanced by H$_3$O$^+$. Although, the cleaning on this specimen, a decisive act for MXene incorporation, requires an irreversible decline of the entered water quantity. The HF concentration further permits improving the dispersal of terminal groups (O/OH ratio) and the stability against surface oxidation. These surface chemistries are captivating for many applications. For example, at lesser HF concentrations, the evolution of MXenes with a greater activity is favored likely due to its little F content. At higher HF concentrations, the stability against oxidation is favored for a minimum of one month.

(2) Etching with LiF/HCl arises in producing conductive clays enabling the processing of these materials. Soft etching conditions—low temperatures and durations—caused

the generation of $Ti_3C_2T_x$ with wide flakes and capacitances. If the etching problems are severe (high temperature and duration) the emerging flakes are minor and more deformed, directing to boost hydrogen evolution reaction activity.

(3) Although the LiF is replaced by $FeF_3$, the kinetics of the MAX-to-MXene modification is boosted, even when the starting F/Al ratio is similar, due to the oxidation properties of $Fe^{3+}$, which are interesting when one is looking to lessen the universal cost of the MXene fabrication. Conversely, this process does not permit a spontaneous delamination in water, as with the LiF/HCl etching method.

This technique further admits the development of $Ti_3C_2T_x@TiO_{2-x}F_{2x}$ powders. The adjusting of the synthesis variables (temperature, duration, Fe concentration) grants the control of the mass of $TiO_{2-x}F_{2x}$ formed during synthesis. This oxyfluoride compound possessed the anatase structure. Manufacturing this type of composite with controlled composition in a one-pot synthesis is fascinating for various applications, such as photocatalysis and batteries [32]. The resulting composite sheet has good conductivity (71.91 S/cm) and electromagnetic interference (EMI) shielding properties (28.82 dB). The composite plate, which had a breadth of 29 m, had an extremely good SSE/t (12,422.41 dB cm$^2$·g$^{-1}$). This research demonstrates the significant benefits of MXene nanoparticles and nanofiber plate in the fabrication of shielding substances, as well as a new technique for producing ultra-thin and high-performance EMI shielding composite film [33].

Chen, X. et al. prepared two kinds of 2D MXene, namely, $Ti_3C_3$ and $Nb_2C$, using the hydrothermal etching process. It was found that as-prepared MXene possessed a larger BET-specific surface area compared to MXene prepared by the conventional HF etching method, which will aid in the higher adsorption performance of cationic dye [34]. Preethi, J. et al. employed MXene for the adsorption elimination of phosphate and nitrate ions from water. It can be concluded that the ions of nitrate and phosphate were resourcefully adsorbed by MXene from the aqueous environment [35]. Sreedhar, A. et al. studied MXene as a strong methylene blue adsorbent in wastewater. These consequences demonstrated that the mainly F-terminated $Ti_3C_2T_x$ MXene is extremely favorable as a strong, recyclable adsorbent for the elimination of methylene blue in wastewater [36]. Morsin, M. et al. focused on the utilization of MXene to study its application in promoting antifungal activity. The outcomes suggested that the d-$Ti_3C_2T_x$ MXene nanosheets displayed magnificent antifungal properties by impeding the mycelium and spore germination of tricoderma reesei because of cell physical destructions prompted by the d-$Ti_3C_2T_x$ MXene nanosheets [37]. Pt-immobilized partially etched MXene/MAX hybrid monolith, as high-performance catalysts, was constructed for hydrogen evolution reaction through the spontaneous redox reaction between $[PtCl_6]^{2-}$ and MXene. This scheme holds the benefits of the better stability of MAX phases in acidic solution, large electrical conductivity and strong bonding among the MXene and MAX phases. The catalyst exhibits minimum overpotential vs. reversible hydrogen electrode (43 mV for 10 mA/cm$^2$) based on slight Pt loadings (lower than 8.9 mg/cm$^2$) [38]. $Ti_3C_2T_x$ MXene manufacture by microwave incorporated the MAX phase and displayed an original size of ~80 nm with crystalline fractures. They inspected the impact of spinning speed on optoelectronic properties of $Ti_3C_2T_x$ MXene transparent conductive electrodes. Rising the spinning speed from 1000 to 4000 caused an increase in transparency (T550) (from 72% to about 94%) and sheet resistance (Rs) (from 2010 to 23,660 $\Omega$/sq.). A figure of merit value of 2.027 $\pm$ 0.163 was computed from data fitting of T550 vs. Rs (R$^2$ > 0.97) for transparent conductive electrodes [39]. Dong, S. et al.'s study was conducted on the preparation of flexible multi-scaled MXene for supercapacitor applications. The $Ti_3C_2T_{x-}10$ flexible (the electrode with 10% mass portion nanoparticles) demonstrated a worthy specific capacitance of 372 F·g$^{-1}$ at 1 A·g$^{-1}$, which was more than that of $Ti_3C_2T_x$ film, and an essential cyclic stability up to 95% capacitance retention after 5000 cycles. The symbolic advancement in electrochemical performance was chiefly because of the open sandwich-like structure of the flexible electrode supported by multi-scale $Ti_3C_2T_x$, which stipulated immense surface area and greater active sites [40].

Zalnezhad, E. et al. assembled an advanced 3D MXene-NiCo$_2$S$_4$ nanostructure as a binder-free electrode for chemical capacitors. Low specific capacity and current density can limit the use of the system in supercapacitor applications. These issues can be overcome with a promising design of the microstructure. The synthesized MXene-NiCo$_2$S$_4$ electrode demonstrated valuable electrochemical achievement, with a large specific capacity (596.69 C·g$^{-1}$ at 1 A·g$^{-1}$) as well as excellent cyclic stability (maintained 80% of the primary capacity after 3000 cycles). The cyclic stability was mainly because of the unique structure of the titanium carbide substructure, which not only aided an abundant surface area but also withstood the volumetric strain because of the application of charges with the redox reaction. Moreover, an ASC model built with MXene-NiCo$_2$S$_4$ as a positive electrode and AC as a negative electrode revealed an elevated energy density of 27.2 Wh·kg$^{-1}$ and a high-power density of 0.48 kW·kg$^{-1}$ [41]. Zhao, J. et al. focused on Nb$_2$CT$_x$ MXene preparation treated with lithium fluoride and HCl for applications of a supercapacitor. The obtained results display the excellent crystalline degree and structural arrangement for as-prepared MXene as well as good electrochemical progress. However, a conductive agent such as CNT aids the further enhancement of electrode performance. The Nb$_2$CT$_x$/CNT electrode delivered an energy density of 154.1 µWh·cm$^{-2}$ and a supreme power density of 74,843.1 µWh·cm$^{-2}$, which is better compared with other focused MXene-supported supercapacitors [42]. Min Y. et al. had illustrated a simple, efficient and scalable process to manufacture Ti$_3$C$_2$T$_x$. Mxene supported stretchable electrodes for supercapacitors. The produced elastic supercapacitor provided an areal specific capacitance of 33.3 mF·cm$^{-2}$ at a scan rate of 10 mV·s$^{-1}$. It could be expanded to 30% without evident capacitance break down, and remained 90% of its original capacitance after 3000 extending cycles to a highest strain of 30% [43]. Shi, L. et al. studied 2D MXene treated with three various etching agents, such as HF, LiF/HCl and tetramethylammonium hydroxide (TMAOH). It was found that HFTi$_3$C$_2$T$_x$ comprised both particles and sheets, with eO and eF directing lapse, while LH-Ti$_3$C$_2$T$_x$ had principally consisted of a minute film, holding a huge amount of eO/eOH terminations, and TM-Ti$_3$C$_2$T$_x$ was also self-possessed films with a greater size, including Al(OH)$_4^-$ terminations. The preferences in the stability and conductivity of the three were TM-Ti$_3$C$_2$T$_x$, LH-Ti$_3$C$_2$T$_x$ and HF-Ti$_3$C$_2$T$_x$, and the flexibility was reversed, which showed that excellent lamellar size and firm microstructure were advantageous for improving conductivity and stability, while the interlamellar association is the dominant constituent to the flexibility [44]. Electrostatic interaction and ion-exchange are the key processes in methylene blue elimination, according to mechanistic studies. In addition, a Ti$_3$C$_2$T$_x$ MXene nanosheet composite membrane had been produced and used for methylene blue (MB) removal by physical separation, with fine elimination efficiency and dye water flux. The topical Ti$_3$C$_2$T$_x$ Mxene suspension appears to be an acceptable MB adsorbent in water [45]. The findings suggest that manipulating MXene's termination group is a viable way to improve the membrane's desalination performance. More advanced production processes are required to accurately manage the adjourning group and the interlayered spacing of MXene nanochannels in order for MXene to be widely used as a desalination membrane [46].

Composites are a type of material that combines the major benefits of each material in terms of property and functionality and can be discovered back to earliest times when grass stems were used to reinforce clay as the principal building material. They are usually the consequence of a combination of factors, structural design and optimization on many scales and layers of elemental materials. The evolution of such a notable achievement and current functions is primarily because of intricate synergistic interactions. At various levels, there is an effect, an interface effect, and a scale effect, which together provide the fundamental, research adaptation, and development direction of modern material science, performance, photothermal conversion ability, and so forth [47]. Furthermore, by creating unusual morphologies (e.g., hollow MXenes), altering the localized structure of MXenes (e.g., doping with other atoms), modifying surfaces (e.g., heat treatment), improving the MXene layer spacing (e.g., inserting cations or organic molecules), and so

on, the performance of MXenes can be improved [48]. The optimal PAA-MXene/PAN membrane has permeate fluxes of 271.26, 516.34, and 300.83 L·m$^{-2}$·h$^{-1}$ and rejections of 98.92%, 99.52%, and 98.12% to DR80, AB8GX, and AB90 at 0.1 MPa, respectively, which is a significant improvement over the MXene-based membrane. Furthermore, the PAA-MXene/PAN membrane is more promising for low-pressure filtration and has a longer lifespan. All of these factors combine to make the PAA-MXene/PANmembrane a potential wastewater treatment nanofiltration membrane [49].

Faradaic deionization (FDI) focusing on faradaic electrodes including Ti-MXene (Ti$_3$C$_2$T$_x$) has been presented for brackish water desalination to improve the desalination performance and energy efficiency of capacitive deionization (CDI), but scalability and massive cost in the material's synthesis is still a problem for the mass manufacturing of the electrode, which inhibits CDI scaling up [50]. The creation of a new non-segregated and dense char layer by the catalytic charring and physical hindrance effect of PCS and MXene, according to the gas and condensed phases studies, is primarily responsible for the improved flame retardancy [51]. The results show that including LDH prevented MXene nanosheets from stacking and increased MXene dispersion in the epoxy (EP) matrix, giving EP composites outstanding thermal and flame-retardant qualities. The higher fire safety of EP composites was related to the catalytic charring and attenuation action of transition metal oxides, as well as the barrier effect of nanosheets and the cooling effect, the gas-phase dilution influence of LDH [52]. The enormous specific surface area of Nb$_2$C MXene can be used as a substrate for the growth of Co$_3$O$_4$ nanoparticles on its surface, forming a 3D cross-linked structure that can provide efficient electron transport channels between Co$_3$O$_4$ nanoparticles and Nb$_2$C MXene. Furthermore, the synergistic impact of MXene and Co$_3$O$_4$ results in the Co-MXene electrode having excellent electrochemical stability [53]. N-wrinkled MXene's and flexible nanostructure performed well in maintaining the Ni-rich NCM811 cathode. As a result, using the crinkled and resilient N-MXene supplement to increase the overall performance of NCM811 Ni-rich compound for macroscale applications of high-energy-density lithium ions batteries should be a more competitive, simple, and efficient technique [54]. The author discovered that different polymer-modified MXene compounds with selectivity, stimuli-responsiveness, contrast enhancement, and sensitivity can be used in biomedical applications, such as photothermal therapy, drug delivery, diagnostic imaging, biosensing, bone regeneration, and antibacterial activity. The current obstacles of nanoparticles include mass production, storage, structure composition precision, in vivo retention, and long-term biosafety, among others, which all obstruct their widespread use in nanomedicine and must be overcome [55]. When compared to pure PW70, the specific heat capacity of the PW70/MXene nanocomposite with a mass fraction of 0.3 wt.% increased by 43%. With a mass fraction of 0.3 weight percent, the thermal conductivity of the PW70/MXene nanocomposite is increased by 16%. MXene nanoparticles have a unique two-dimensional planar structure with well-formed layers, resulting in high-promising thermophysical characteristics. Future research should focus on the influence of the number of layers, thickness, and size. thermal conductivity at elevated temperatures must also be investigated to see how it changes with temperature [56].

Ti$_3$C$_2$ flakes with a well-formed multi-layered structure and homogeneous dispersion in oil medium have the ability to improve thermophysical properties. With the addition of nano additions, the NF's viscosity and density increased by 13.28 and 1.01%, respectively. At higher temperatures, however, viscosity and density reduced dramatically, and rheological behavior was Newtonian up to 100 s$^{-1}$ shear rates. The SO/Ti$_3$C$_2$ NF on hybrid PV/T produces optimal results, including an overall thermal efficiency of 84.25% at 0.07 kg/s discharge. In addition, when the NF was used as a cooling fluid instead of water, the surface temperature reduced by 14 degrees Celsius. By replacing the water/alumina cooling fluids in the hybrid PV/T system with SO/Ti$_3$C$_2$ NF, the system's electrical output was increased by 15.44% at a mass flow rate of 0.07 kg/s. Regardless of the fluids used in this investigation, the variation of heat transfer coefficient with mass flow rate showed an increasing tendency with increasing flow rate. Compared to the alumina/water

NF, MXene/soybean oil achieves a maximum augmentation of 14.3% at 0.06 kg/s [57]. The loading concentration of the MXene-OPO nanofluid is 0.01, 0.03, 0.05, 0.08, 0.1, and 0.2 wt.%. At 25 °C, MXene-OPO nanofluid with a 0.2 wt.% loading concentration has a 68.5% higher thermal conductivity than pure OPO. When the temperature is raised from 25 to 50 degrees Celsius, the viscosity of nanofluid with 0.2 weight percent MXene is reduced by 61%. When compared to PVT with $Al_2O_3$-water-based nanofluid, MXene-based nanofluid improves thermal efficiency by around 16% at a 0.07 kg/s flow rate. PVT with MXene nanofluids improves the heat transfer coefficient by about 9% when compared to PVT with $Al_2O_3$-water heat transfer fluid. When compared to a stand-alone PV module, MXene nanofluid may decrease PV temperature by 40% [58]. When loaded with MXene, the dielectric constant augmentation effect is seen in other polymers as well. We show that the charge accumulation generated by creating tiny dipoles at the interfaces between the MXene sheets and the polymer matrix under an external electric field is substantially responsible for the increased dielectric constant [59]. Si/MXene composite sheets have excellent lithium storing properties. Highly conductive MXene networks can provide a stiff current collector, improve Si/MXene conductivity, accommodate huge volume expansion, provide more active sites, and make ions transport more efficient. Meanwhile, the silicon nanospheres can prevent MXene sheets from re-stacking. This research could lead the way for large-scale Si anode use in LIBs [60]. The corrosive impediment of the aqueous epoxy plating was dramatically increased by $Ti_3C_2$ MXene@PANI complexes, overcoming the accelerating corrosive issue of $Ti_3C_2$ MXene. The results show that by combining the barrier impact of $Ti_3C_2$ MXene nanosheets with the passivation influence of PANI, Ti3C2 MXene@PANI compounds with less electrical conductance can accomplish prolonged and effective corrosion preservation in waterborne epoxy coating for Q235 steel [61].

MXene synthesis via selective-etching-produced stacks of many single-layered MXene flakes, which are then converted to single-layer flakes following the delamination of multilayered MXenes. The delamination of layered MXenes mainly entails the chemical deposition of MXenes containing significant organic molecules to extend their interfacial gap, lowering the interaction within particular MXene layers dramatically. The hydrophilicity of electrode material needs special attention, which could badly affect the performance of the electrode system. However, MXene proved suitable electrode material, possessing good hydrophilicity with functional groups on its surface, such as O, OH and F. The development of heterogeneous catalysts comprising a ternary transition metal was required. Secondary aqueous and nonaqueous metal batteries undergo dendrite growth, which is the major obstacle in its way. However, MXene and metal composite contributed well in designing dendrite-free metal-based batteries. In order to boost up the performance of MXene-based ceramic materials regarding their method of preparation, properties and application in various fields have urged researchers to investigate MXene with their composite materials. A lot of work has been conducted on MXene and its composites with different materials; each MXene-based composite showed outstanding performance for a specified purpose. However, to the best of our knowledge, there is no detailed review presented about MXene and metal composite. Therefore, we focused on a detailed review of knowledge over MXene and metal composites, which will be helpful in further research progress in the improvement of composites. This study unlocked the synthesis of MXene and metal composite and their improvement in properties. Moreover, this study presented the application of MXane and metal composite such as electrochemical performance, catalytic activity and corrosive performance.

## 2. Preparation of MXene–Metal Composites

This section of the paper focuses on the preparation of MXene and metal composites as shown in Figure 1.

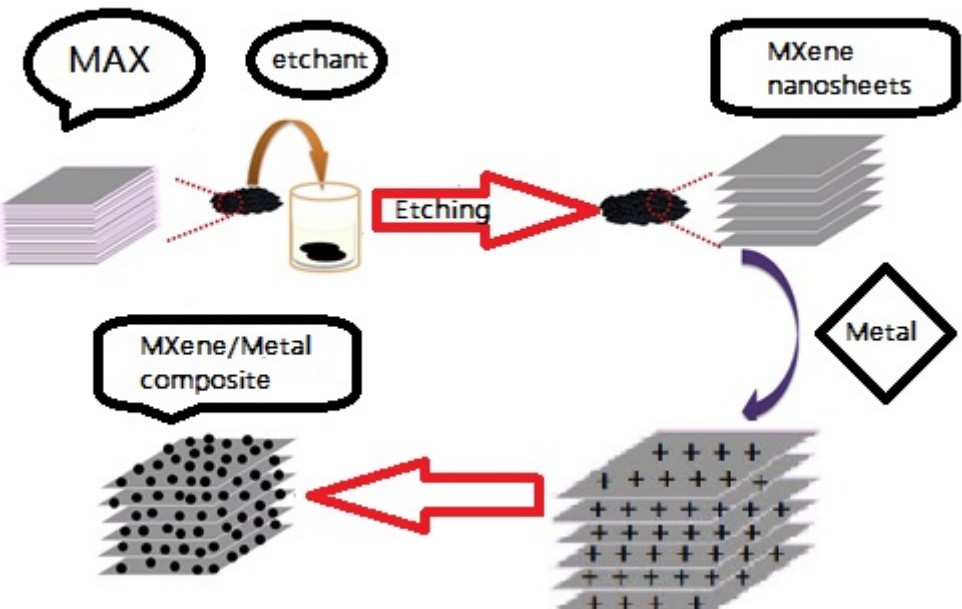

**Figure 1.** Schematic illustration of MXene and metal composite synthesis procedure.

### 2.1. Au/Ti₃C₂Tₓ Nanocomposite

Au/Ti$_3$C$_2$T$_x$ nanocomposite was synthesized by a chemical reduction method. In this process, reducing agent was added to reduce solution containing metal, which was further assisted through stirring. After this, Ti$_3$C$_2$T$_x$ MXene was put in the aforementioned mixture. The emerging compound had been ultrasonicated for 30 min. After completion of a reaction, the solution was drained utilizing filter paper made of cellulose with 0.1 μm pore size after being rinsed three times with deionized water. The filtered material was then dried for 2 h in a vacuum oven at 80 °C [62].

### 2.2. RhNi/MXene Nanocatalyst

The RhNi/MXene nanocatalyst was prepared via the one-step wet chemical method. To create a homogeneous dispersion, in a two-neck round-bottom flask, 100 mg MXene is dissolved in 2 mL water (30 mL) and sonicated for 30 min. Then, for 20 min, gently stir in 100 L rhodium chloride mixture (0.8 mmol/mL) and 100 L nickel chloride solution (0.2 mmol·L$^{-1}$) into the aforementioned MXene solution with electromagnetic stirring (speed of 220 rpm). Then, 24 mg sodium borohydride (NaBH$_4$) (1.3 mol·L$^{-1}$) dissolved in 0.5 mL 2.0 M NaOH solution is promptly added to the above-mentioned mixture and vigorously stirred for three hours at 0 °C, employing an ice-bath to keep the condition low enough to suppress RhNi nanoparticle accumulation. The RhNi/MXene nanocatalysts are made by centrifuging and washing with deionized water [63].

### 2.3. Ti₃C₂/DNA/Pd/Pt Nanocomposite

To create dsDNA, the dsDNA had been dispersed in distilled water, warmed at 95 °C for 20 min, and then rapidly chilled in an ice water bath. After that, 1 mL of 1 mg/mL Ti$_3$C$_2$ distribution was blended along 1 mL solution of DNA, sonicated for 30 min in a cold-water tub, then centrifuged for 10 min at 10,000 rpm. In 7.5 mL deionized water, the sediment was re-dispersed. In total, 1 mL of 0.01 M PdCl$_2$ had combined with Ti$_3$C$_2$/DNA procedure well agitated for 20 min to make PdNP-modified MXene nanosheets. Then, in an ice water bath, 100 μliters of 0.1 M NaBH$_4$ was steadily added following the sonication process for 30 min. After that, 1.2 mL of 0.01 M H$_2$PtCl$_{6.6}$H$_2$O underwent stirring for 20 min, followed by 400 mL of 0.1 M NaBH$_4$ being gently put into an ice water bath under stirring for 30 min. After three centrifugations, the Ti$_3$C$_2$/DNA/Pd/Pt nanocomposite was finally produced [64].

### 2.4. Ti₃C₂Tₓ /Ni MXene Composite

Ingredients were dissolved using magnetic stirring followed by two successive stirring steps and then heated in a stainless-steel autoclave lined with Teflon. Deionized water and 100% ethanol were used to rinse the products many times. Eventually, the dark powders had been dehydrated overnight [65].

### 2.5. Ti₃C₂Tₓ/Al MXene Composite

The pressureless sintering method was used to prepare composites of $Ti_3C_2T_x/Al$, and the process was then followed by hot extrusion. Process conditions such as 650 °C and 1 h in Ar were used for pressureless sintering, and a temperature of about 450 °C was chosen for the hot extrusion process. Properties of composites were evaluated on the basis of adding $Ti_3C_2T_x$ aggregate from 0.5 wt.% to 3 wt.%. As the content of $Ti_3C_2T_x$ increased to 3 wt.% in the composites, the Vickers hardness (0.52 GPa) and the tensile strength (148MPa) significantly improved by 92% and 50%, sequentially in contrast with pure Al [66].

### 2.6. Ti₃C₂ MXene@Au@CdS Composite

The self-reduction process was used for the preparation of $Ti_3C_2$ MXene@Au composite. A solution of $Ti_3C_2$ MXene (100 mg) and ultrapure water (100 mL) was prepared by stirring. Then, $HAuCl_4$ (3 mL/0.1 mol·$L^{-1}$) was put gently in solution carrying constant stirring to set off the self-reduction. The process was carried out for 30 min and then the suspension was centrifuged and cleansed with ultrapure water persistently. Finally, the $Ti_3C_2$ MXene@Au composite was obtained by lyophilization at −60 °C for 48 h [67].

### 2.7. FLM/Al Composite

Firstly, the beaker was filled with Al powder and deionized water was added, which was mechanically stirred and ultrasonicated in an ice-bath. Afterward, MXene was added dropwise into the Al suspension followed by stirring process. The suspension was then filtered and dried in a vacuum furnace and the desired Mxene/Al powder mixture was achieved. Lastly, the Spark plasma sintering (SPS) process was employed to consolidate the MXene/Al powder mixture [68].

### 2.8. Ag-Ti₃C₂Tₓ and Ag-Nb₂CTₓ Composites

Simultaneous self-reduction and oxidation methods had been used to make metal nanoparticle adorned oxidized MXene ($Ti_3C_2T_x$ and $Nb_2C_{Tx}$) ternary composites. MXene $Ti_3C_2T_x$ had been homogeneously diffused in deionized water and sonicated in a bath for 30 min. Following magnetic stirring and sonication, silver nitrate was dispersed individually in deionized water. Then, the as-prepared $AgNO_3$ suspension was dropped into MXene solution dropwise with vigorous stirring, and the solution was stirred for another 30 min followed by sediment collection. After collecting sediments, they were cleaned with water and ethanol before being vacuum dried for 24 h at 40 °C. The same procedure was applied for the preparation of Ag-$Nb_2CT_x$ [69].

## 3. Properties of MXene and Metal Composites

This section highlights characteristic properties of MXene and metal composites, which includes microstructure, mechanical properties, electromagnetic adsorption and wettability.

### 3.1. Microsructure

Figure 2 shows a diagram of the Au/MXene composite combination. Figure 2a shows $Ti_3C_2T_x$ (MXene), Au nanoparticles alongwith Au/MXene nano-composite powder XRD patterns. The Te pattern of $Ti_3C_2T_x$ MXene closely resembles the XRD pattern of HF-treated $Ti_3C_2T_x$ reported in the literature. Each of the diffracted apexes within nanoparticles and XRD specimens were gathered. A truecubic close-packed form of Au can be easily indexed from the solution. The reflections of Au-MWNT nanocomposite material may be seen in the XRD pattern. Au is present, as is $Ti_3C_2T_x$ MXene. The Figure 2b TEM results confirm

that nanoparticles of Au had been uniformly distributed on the exterior of an exfoliated very fine and explicit MXene nanosheet.

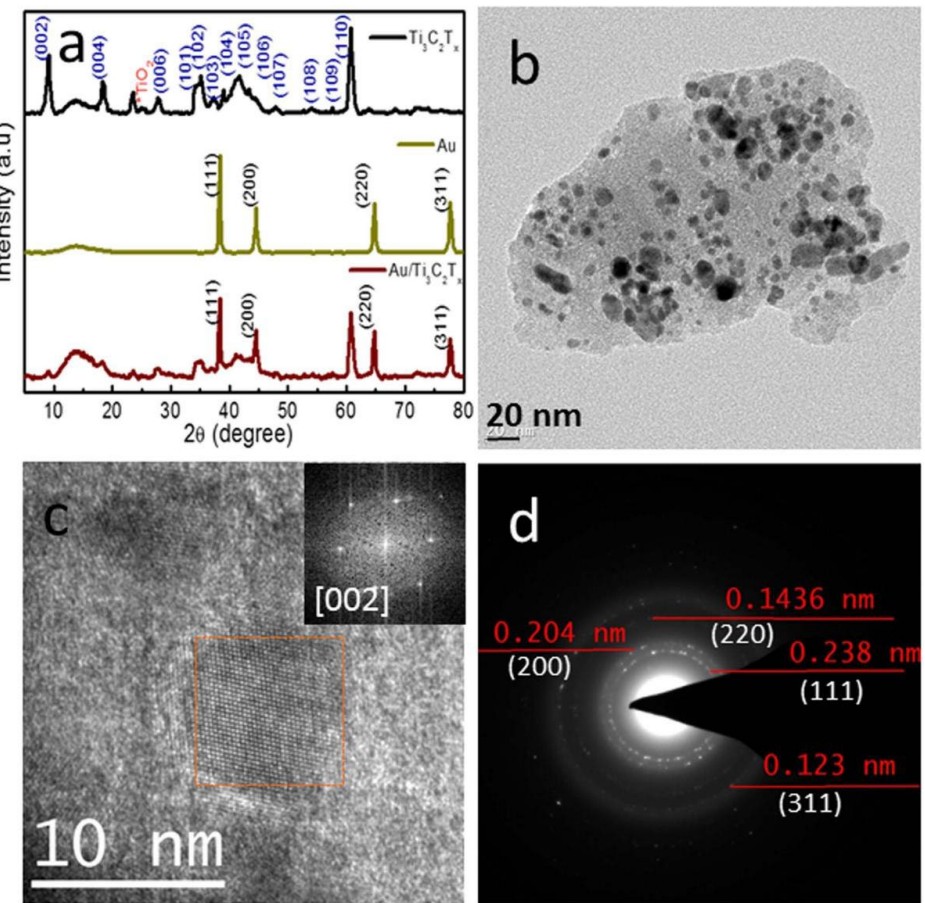

**Figure 2.** (**a**) Powder X-ray diffraction pattern of $Ti_3C_2T_x$ MXene, Au nanoparticles and Au/Mxene nanocomposite (**b**) TEM and (**c**) HRTEM images of Au/MXene nanocomposite. Inset of (**c**) shows the FFT pattern taken from the marked area. (**d**) SAED pattern corresponding to the marked area in (**c**). Reprinted with permission from [62]. Copyright 2016 Springer Nature.

Moreover, the TEM sample development procedure demanding ultra-sonication, the mostly Au nanoparticulate keep affixed to the MXene slabs, demonstrates intense contacts found linking the Au nanoparticles and MXene nano-sheets. Au particles are 6–8 nm in size on average. As an inset, the FFT (fast Fourier transform) specimen of a chosen section acquired along the (002) zone axis from the HRTEM figure of Au/ $Ti_3C_2T_x$ composite exhibited in Figure 2c is displayed. Figure 2c shows the equivalent preferred region electron diffraction (SAED) pattern, while Figure 2d shows the SAED pattern. The pattern represents the nanogold's polycrystalline structure. The SAED pattern can have diffraction rings. Utilizing round Hough diffraction analysis, these data were marked to the FCC cubic Au structure, and they agree well with the PXRD findings. The crystalline character of the Au nanoparticles is readily visible in the HRTEM image in Figure 2c [62].

To further examine the morphologies of RhNi NPs, TEM as well as HRTEM are utilized as shown in Figure 3. The RhNi/MXene TEM and HRTEM pictures (Figure 3a–c) showed that the discrete RhNi nanoparticulate having an average dimension of 2.8 nanometer were homogeneously distributed across the plane of MXene. Metal as well as non-metal elements are equally distributed all over the MXene, as seen by the elemental mappings of Ni, Rh, C, O, and Ti (Figure 4). The production of mono-dispersed RhNi NPs improves catalytic properties for the breakdown of $N_2H_4H_2O$ [63].

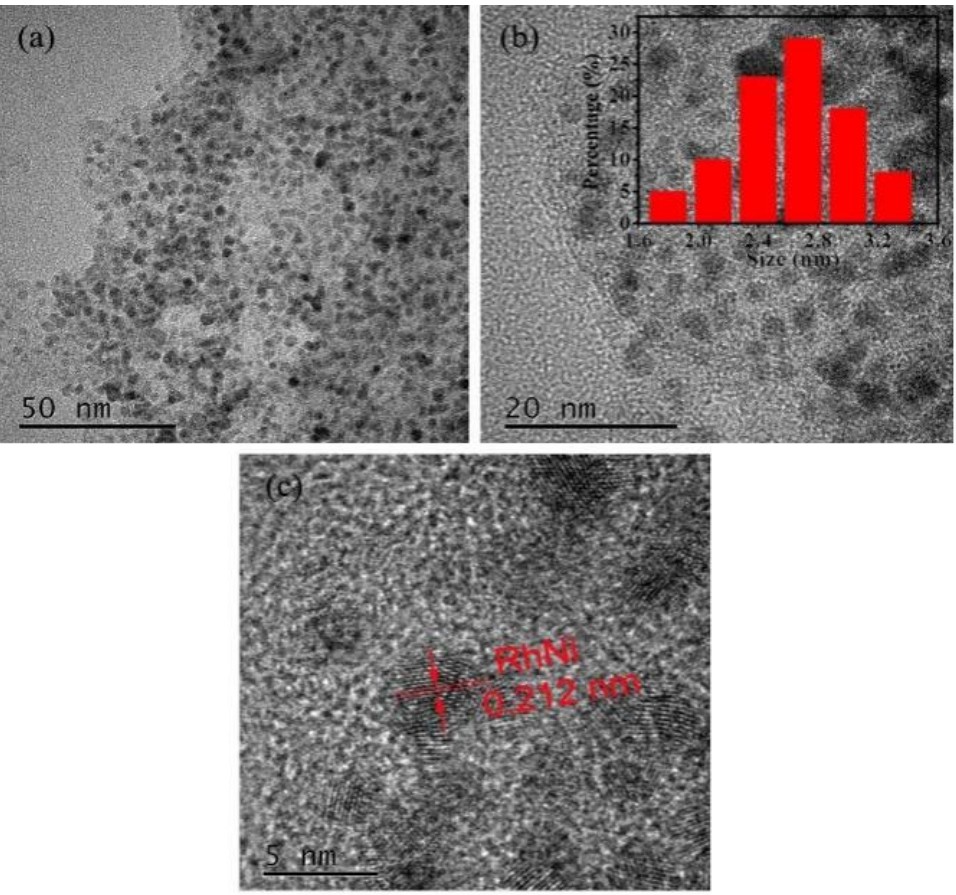

**Figure 3.** (**a**,**b**) TEM and (**c**) HRTEM images of the $Rh_{0.8}Ni_{0.2}$/MXene. Reprinted with permission from [63]. Copyright 2018 John Wiley and Sons.

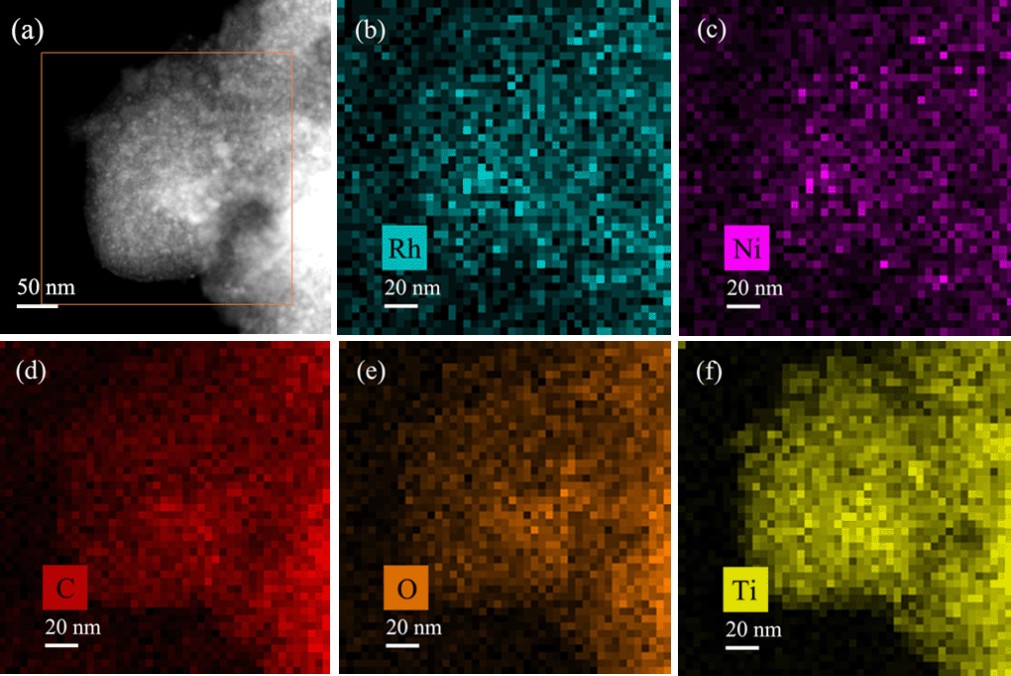

**Figure 4.** The STEM-HAADF images of the $Rh_{0.8}Ni_{0.2}$/MXene (**a**) and the corresponding elemental mapping images for Rh (**b**), Ni (**c**), C (**d**), O (**e**) and Ti (**f**). Reprinted with Permission from [63]. Copyright 2018 John Wiley and Sons.

It can be confirmed through XRD analysis (Figure 5a,b) that modified groups of −OH are formed when treated with HF and Ni spheres successively embedded between the $Ti_3C_2T_x$ MXene sheets and make a laminated arrangement, which leads to interface polarization. The spaces in the middle of $Ti_3C_2T_x$ nanosheet layers were distinctly seen in each laminated assembly, which was compatible with XRD analysis. Some of the Ni nanoparticles were found on the edge of each film perforate due to the limited growing room, the interspace between each lamella, and Ni prepared by the hydrothermal process has a greater average sphere dimension compared to pure Ni [65].

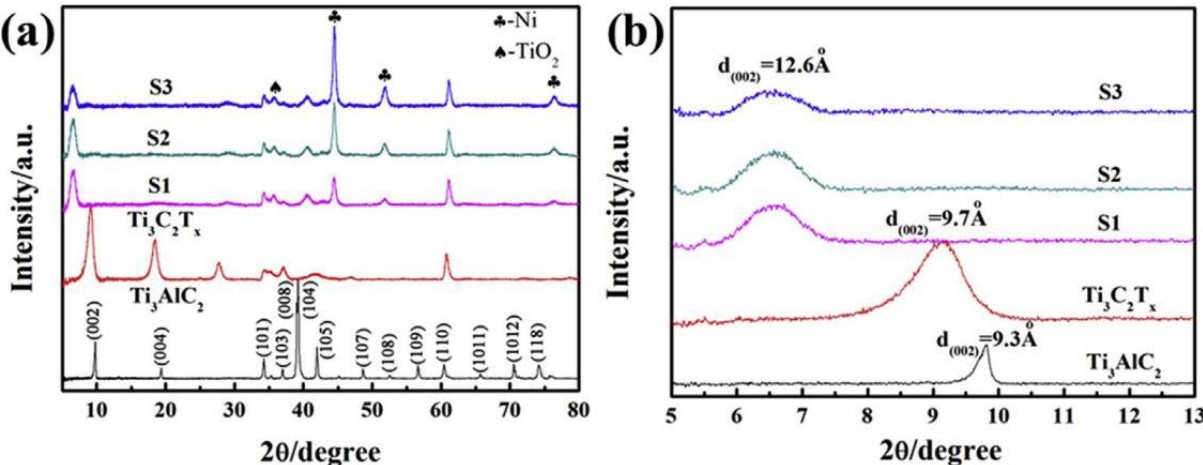

**Figure 5.** (**a**) XRD patterns of $Ti_3AlC_2$, $Ti_3C_2T_x$ and $Ti_3C_2T_x$/Ni-sphere hybrids. (**b**) Zoomed-in views of (**a**) in the 2θ range of 5–13°. Reprinted with permission from [65]. Copyright 2019 Elsevier.

SEM analysis showed that an equivalent dispersion of $Ti_3C_2T_x$ in Al matrix composite was obtained and no agglomeration of $Ti_3C_2T_x$ particles was found (light gray) (Figure 6a). Similarly, EDS tests (Figure 6b) confirmed that $Ti_3C_2T_x$ scattering in the Al matrix was predominantly ceased with –F and –O modified groups because of the composition of Ti, C, F and O in the $Ti_3C_2T_x$ particle [66].

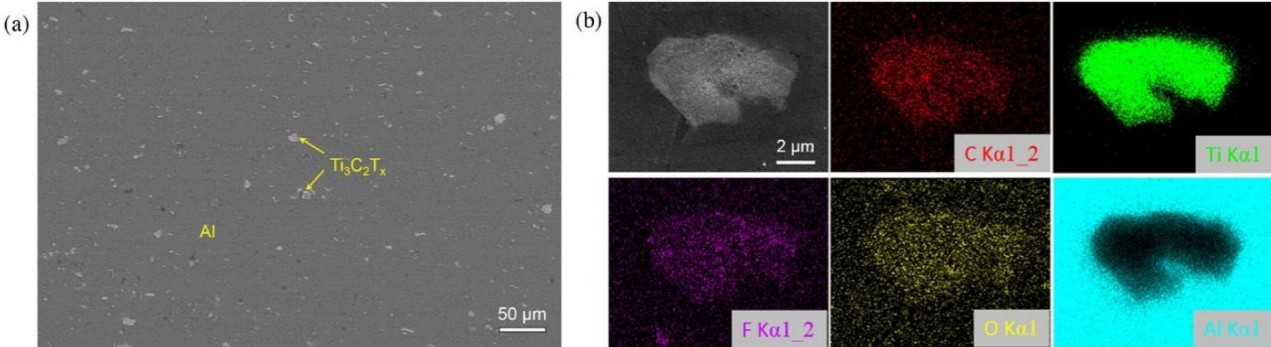

**Figure 6.** (**a**) Backscattered SEM micrograph of the polished surface of 3 wt.% $Ti_3C_2T_x$/Al composite. (**b**) EDS maps for the distribution of C, Ti, F, O and Al. Reprinted with permission from [66]. Copyright 2020 Elsevier.

TEM (Figure 7a–c) indicating a mean diameter of about 150 nm was noted for the CdS nanoparticles with a uniform spherical structure. MXene showed a typical layered structure with a uniform thickness. The Au and CdS particles are regulated and evenly moved to the MXene's and each layer's surface. Figure 7e,f of MXene@Au@CdS exhibited no accumulation, specifying that MXene is a fine substrate for dispersion growth. Furthermore, tight connections existed between MXene, Au, and CdS, which provide charge separation

and transfer. EDS analysis confirmed (Figure 8) an MXene@Au@CdS heterostructure that is compatible with TEM data [67].

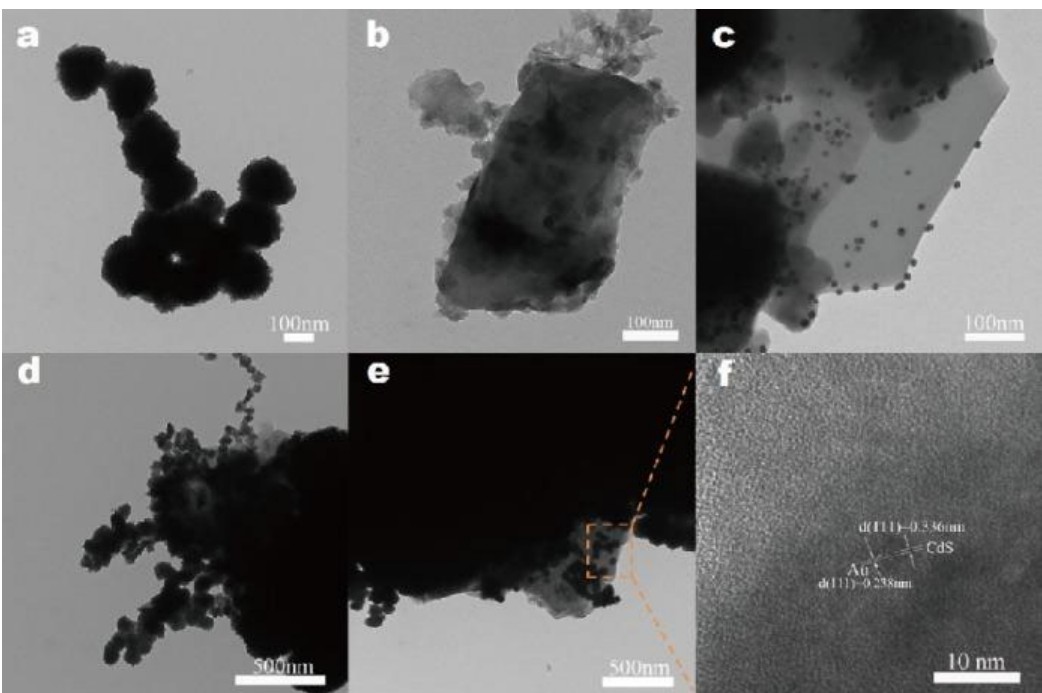

**Figure 7.** TEM images of (**a**) CdS, (**b**) MXene, (**c**) MXene@Au, (**d**) MXene@CdS, (**e**) MXene@Au@CdS and HRTEM image of (**f**) MXene@Au@CdS [67].

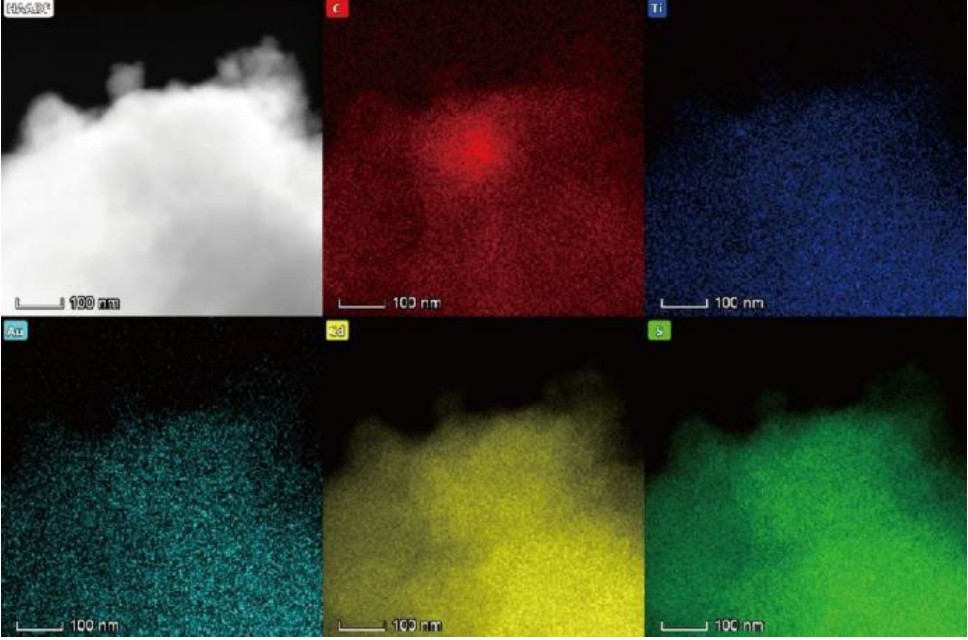

**Figure 8.** TEM images and elemental mapping images of MXene@Au@CdS nanocomposite [67].

The specific surface area and porosity of the resulting specimens had been established using a nitrogen adsorption–desorption technique, as shown in Figure 9. The H$_3$ isotherm in pure MXene is typical, exhibiting its mesoporous characteristics. The Brunauer–Emmet–Teller (BET) surface areas, pore volumes, and pore diameters of the processed materials are illustrated in Table 1. The BET surface area, pore volume, and pore size of MXene

(MXene@Au@CdS) suggested that after feeding with Au@CdS nanomaterials, the MXene maintains a porosity and has a proportionally high discrete surface.

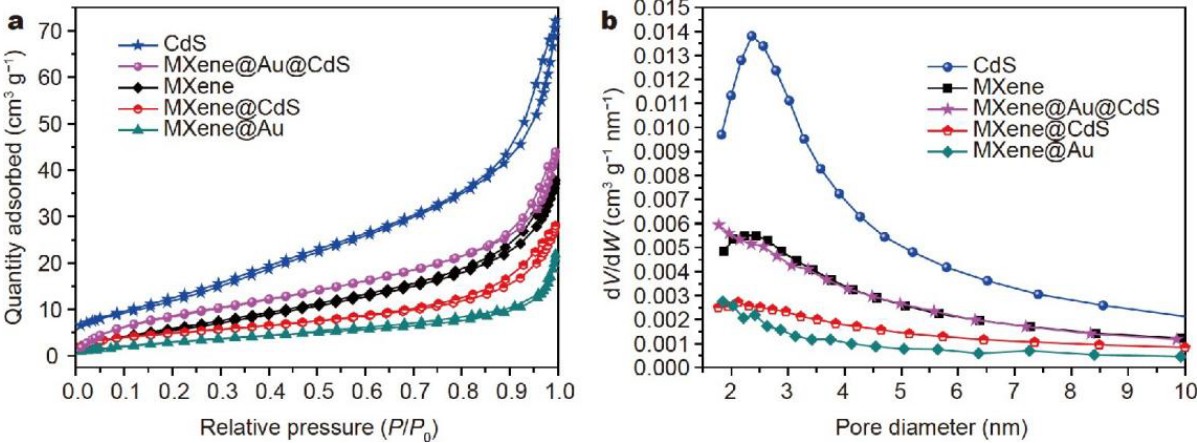

**Figure 9.** N$_2$ adsorption–desorption curves (**a**) and the pore size distributions (**b**) of MXene, MXene@Au, MXene@CdS, MXene@Au@CdS and CdS [67].

**Table 1.** The BET data of prepared samples [67].

| Catalysts | Surface Area (m$^2 \cdot$g$^{-1}$) | Pore Volume (cm$^3 \cdot$g$^{-1}$) | Pore Size (nm) |
|---|---|---|---|
| MXene | 22.2672 | 0.0563 | 9.7948 |
| MXene@Au | 12.2502 | 0.0316 | 9.4052 |
| MXene@CdS | 17.8133 | 0.0421 | 9.0111 |
| MXene@Au@CdS | 34.5170 | 0.0622 | 7.3110 |
| CdS | 45.4100 | 0.1096 | 9.1249 |

The UV–visible diffuse reflectance spectra (DRS) patterns of MXene, MXene@Au, MXene@CdS, MXene@Au@CdS, as well asCdS were examined, as mentioned in Figure 10. The absorption intensity of MXene steadily reduces as the wavelength increases, which could be ascribed to the peculiar absorption of carbonic substances.

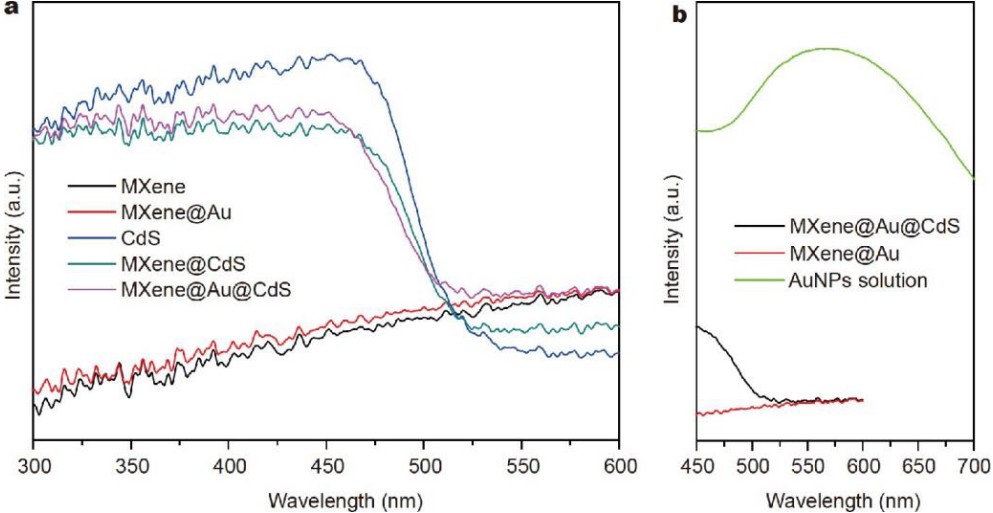

**Figure 10.** UV–Vis DRS spectra of MXene, MXene@Au, MXene@CdS and MXene@Au@CdS (**a**); enlarged curves of MXene@Au, MXene@Au@CdS and AuNPs solution (**b**) [67].

Due to the modest amount of MXene injected, the absorbance intensity of MXene in the DRS spectrum is relatively low. In addition, as seen in Figure 10a, the CdS specimen

exhibits an abrupt absorption edge at 475 nm, as expected. The absorption spectrum of the MXene@CdS compound has focused at around 475 nm, demonstrating the CdS property. The resulting MXene@Au@CdS curve displays the spectrum properties of CdS as well as MXene. Furthermore, because the number of MXenes injected during the self-reduction event is rather large, the absorption apex of the produced Au nanoparticles is shielded. As a result, as shown in Figure 10b, there would be no noticeable surface plasmon resonance (SPR) apex for Au nanomaterials in the DRS data. According to the foregoing findings, the produced ternary complex MXene@Au@CdS conclusively increases CdS absorption in the visible region, which is helpful for the future photocatalytic hydrogen generation reaction [67].

The microstructure of 0.26 vol.% few-layered MXene/Al (FLM/Al) compound at the EBSD-inverse pole was investigated as shown in Figure 11a–f. The shape of Al particles at the beginning was spindle-shaped and possessed a fiber texture down the extrusion behavior because of the metallic plastic flow. The wrinkled few-layered MXene (FLM) incorporated within an Al particle might be benefited by metallic plastic flow towards the stretching and flattening behavior. The few-layered MXene (FLM) platelets were scattered and predominantly oriented along the extrusion direction, according to the long cross-sectional image of the FLM/Al compound.

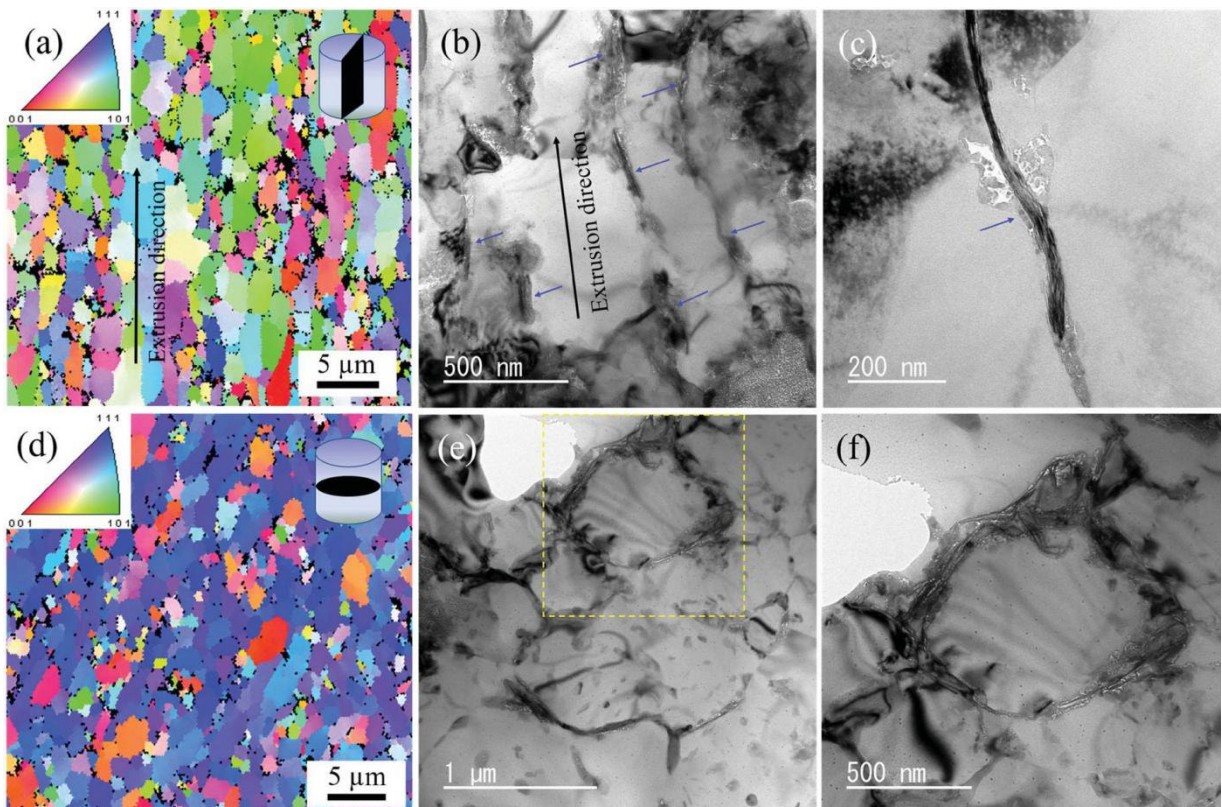

**Figure 11.** Microstructures of the 0.26 vol.% FLM/Al composite in the (**a**–**c**) longitudinal and (**d**–**f**) transverse cross-sections: (**a**,**d**) EBSD-IPF maps; (**b**,**c**,**e**,**f**) TEM images. (**f**) is taken from the yellow square in (**e**). Reprinted with permission from [68]. Copyright 2020 Taylor and Francis.

The few-layered MXene (FLM) containing 0.13 vol% or 0.26 vol.% reduced the size of pure Al to 2.45 or 1.96 μm, while the mean particle magnitude of true Al had dictated as 2.92 micrometer. During the densification process, the pinning of few-layered MXene (FLM) platelets generated this grain finesse. The Al component was loaded into the gap of the FLM films, according to other HRTEM-EDS evaluation.

Regardless of the FLM-Al interface's lack of wettability, the few-layered MXene (FLM) had closely associated with the Al, specifically the Al₂O₃-coating, which was unbound by

nanovoids or contaminants. Interfacial arrangement of FLM/Al composite was studied by inserting Ti-rich film (~18nm) in the $Al_2O_3$ separate two Al grains and demonstrated the production of an $Al-Al_2O_3-FLM-Al_2O_3-Al$ multiple interphase. During powder mixing in water, the surface oxidation caused the $Al_2O_3$ layer to become slightly dense compared with the emerging aluminum powder (Figure 12a–c) [68].

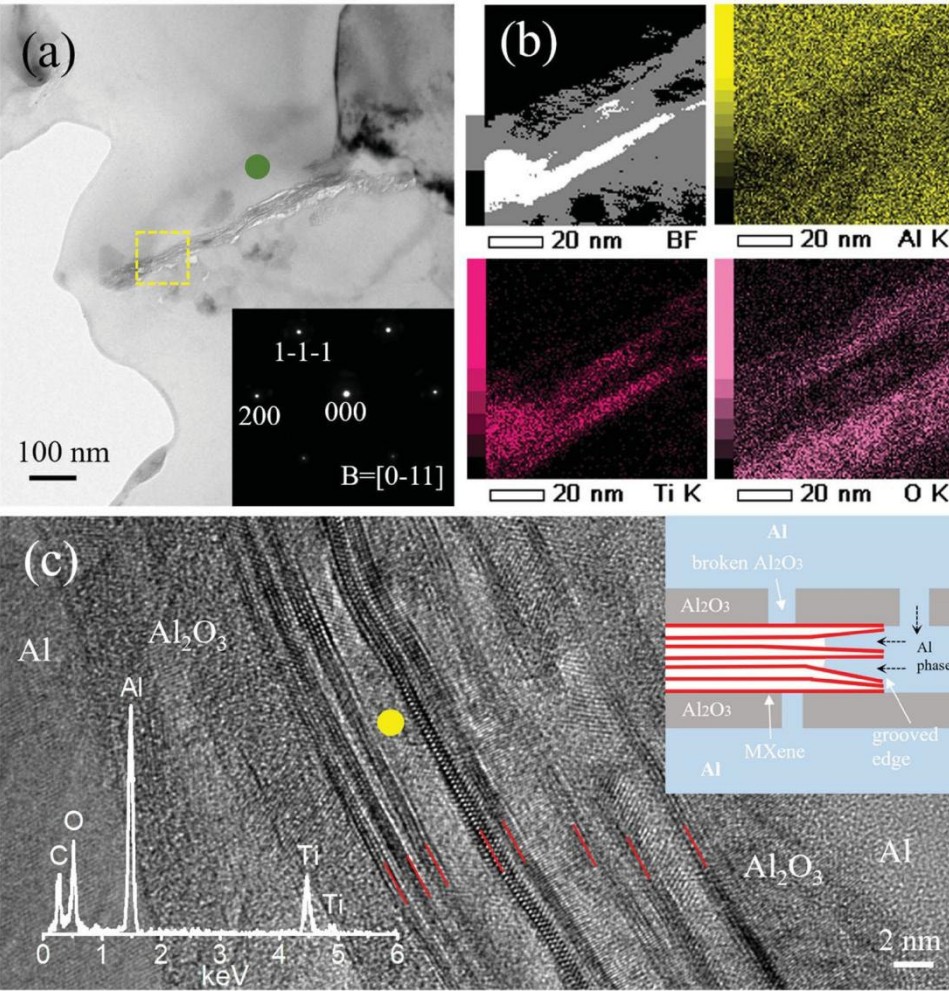

**Figure 12.** (**a**) TEM image of a 0.26 vol.% FLM/Al composite in the longitudinal cross-section; (**b**) ABF-STEM image and corresponding EDS maps of elemental Al, Ti, and O taken from the yellow square in (**a**); (**c**) HRTEM image of the FLM-Al interface. Inset in (**a**) shows the SAED pattern of Al taken from green spot. Insets in (**c**) show the EDS analysis taken from the yellow spot, and the schematic of the Al filled into the grooved edge of the FLM, respectively. Reprinted with permission from [68]. Copyright 2020 Taylor and Francis.

It was noted (inset of Figure 13) that the hybrid colloidal solutions such as Ag, Au and Pd@MXene achieved better dispersion in water. In dilute aqueous medium, the delaminated MXene showed peaks at 225 and 275 nm. Moreover, the MXene colloid displayed maximal absorption in the ultraviolet region around 225 and 325 nm.

As the SPR peaks within the visible range at 440 and 558 nanometers observed for Ag and Au, MXene hybrids have shown that there were nanoparticles of Ag and Au in a colloidal mixture, and they were beneficial for electromagnetic (EM) improvement. Besides this, the Pd nanoparticle (NP) surface plasmon resonance (SPR) bands lay at 230 nm in the ultraviolet region, owing to the interaction with the accumulated Pd NPs; this may be termed a red-shifted MXene peak.

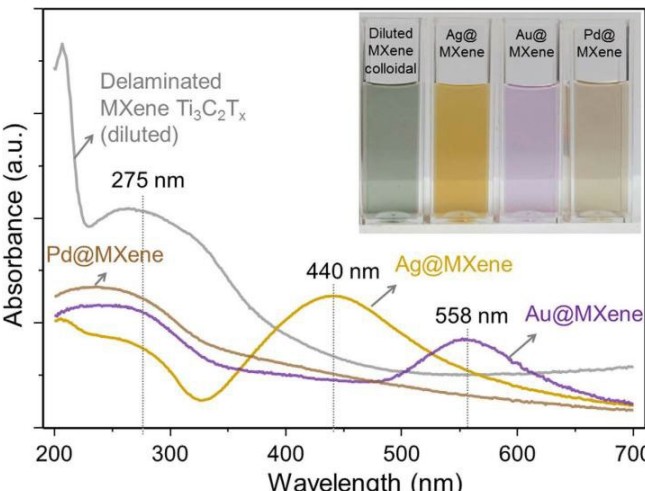

**Figure 13.** UV–Vis spectroscopy analysis on the delaminated $Ti_3C_2T_x$ nanosheets (MXene) and the colloidal suspension of MXene–metal hybrid materials. A photograph of solutions of the diluted MXene colloid, Ag, Au and Pd NPs@MXene colloids is presented in the inset (left to right, correspondingly). Reprinted with permission from [70]. Copyright 2016 Springer Nature.

Individual as well as little-sheet MXene flakes were observed in a sample of bare MXene with a size range of 2–3 μm, and the thickness was from one to a few nanometers (Figure 14a–d). The Ag nanoparticles were rounded with sizes of 10–70 nm, which is difficult to maintain on MXene because the highly reactive $Ag^+$ ions probably encounter fast reduction under ultrasonication, even if the concentration of precursor is kept low (0.1 mM $AgNO_3$ solution). Contrastingly, a better homogeneous size distribution was proposed for rounded Au NPs (40–50 nm) at the same conditions, indicating a moderate depletion of $Au^{+3}$ ions. The Pd@MXene showed layer-like flattened grains above the MXene flakes, demonstrating that the reduction step was opposite compared with Ag and Au. This detailed analysis confirmed the appearance of processed metal nanoparticles on MXene. It can be seen from the XRD results (Figure 15) that Ag and Au hybrids have packed exfoliation of MXene and no crystallographic assembling of MXene layers as well as achieve successful hybridization.

Additionally, the hybridization mechanism of the Pd@MXene hybrid is different and all these findings are in line with the shape analysis noted by TEM. The delaminated MXene (Figure 16) IR peak at 3742 $cm^{-1}$ was assigned to the –OH functional group. Besides this, NP@MXene hybrids have IR peaks at 3400–3800, 1661, and 1211 $cm^{-1}$, which are attributed to $OH/H_2O$ adsorbed on the surface of nanoparticles (NPs), and which were not visible in delaminated MXene flakes [70].

They carried out Raman evaluation on MXene (Figure 17a) prior to nanoparticle (NP) hybridization (Figure 17b–d) to illustrate the surface-enhanced Raman spectroscopy (SERS) reactivity of the NP@MXene composite opposed to the methylene blue (MB) analyte molecule. They soaked the glass-coated specimens in a $10^{-6}$ M ethanol mixture of MB and then dried them. They employed hybridizing NPs with improved size of particle and quantity by putting a 5-fold upraised (0.5 mM) amount of every derived metal complex throughout the functionalization operation to enhance the signal-to-noise ratio of the observed Raman spectra. This could lead to a rapid depletion of the precursor, resulting in an irregular and wide-sized NP deposit on the MXene plane (potentially aggregated). MXene and MXene hybrids (Ag@, Au@, and Pd@MXene) were placed onto a glass dopant to create self-assembled monolayers of MB. Figure 17a shows the characteristic Raman peaks of partly oxidised $Ti_3C_2T_x$. The oxidation could have occurred as a result of ethanol or MB reactions, or as a result of the large laser power (35 mW). Despite the fact that MB molecules had adsorbed on MXene, no Raman spectrum characteristics of methylene blue could be seen in Figure 17a. This is because of the lack of NPs on the MXene plane that were

required for increasing the Raman signal of MB. Because of the substantially greater Raman spectral cross-section of the overlying methylene blue molecules, the Raman characteristics of MXene are no longer visible in the NP@MXene composites. The SERS spectrum for the 3-NP hybrids (with various intensities) show the distinctive apexes of MB about 443 and 1615 cm$^{-1}$, which have been ascribed to the C–N–C sketchy curve and C–C extending, sequentially, showing the molecules had adsorbed upon the surfaces.

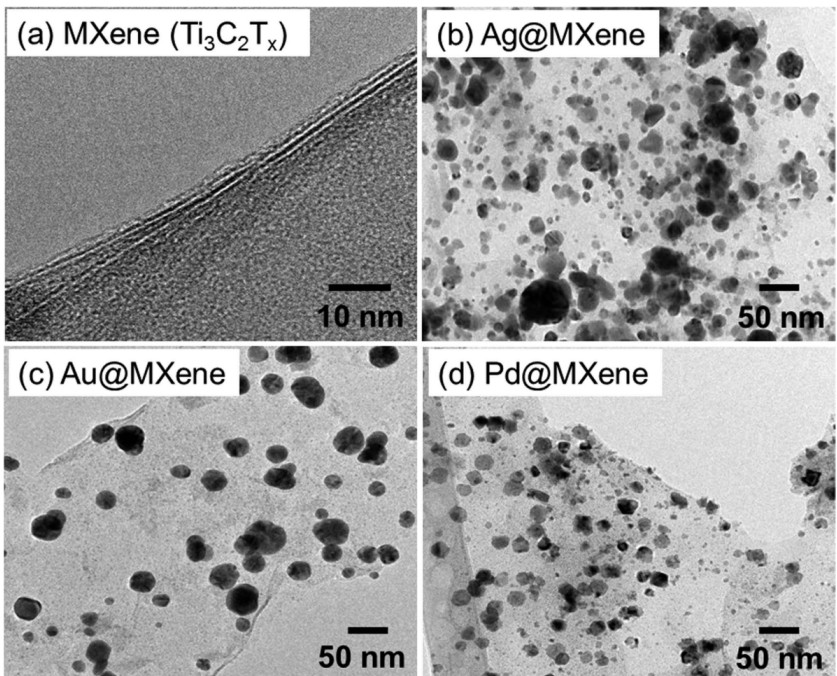

**Figure 14.** (**a**) High-resolution TEM image of MXene nanosheets and low-resolution TEM images of (**b**) Ag@MXene, (**c**) Au@MXene and (**d**) Pd@MXene hybrid nanosheets. Reprinted with permission from [70]. Copyright 2016 Springer Nature.

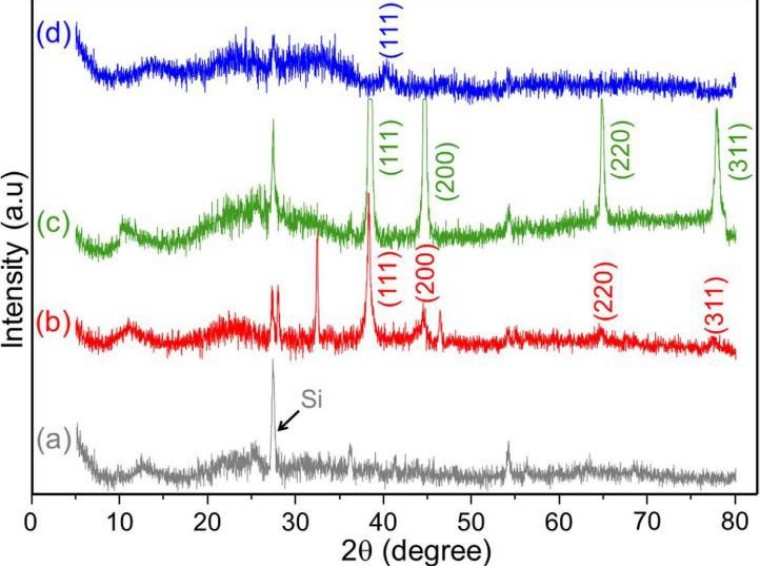

**Figure 15.** The powder X-ray diffraction patterns of (**a**) delaminated MXene nanosheets, and after hybridization with (**b**) Ag, (**c**) Au and (**d**) Pd nanoparticles. Reprinted with permission from [70]. Copyright 2016 Springer Nature.

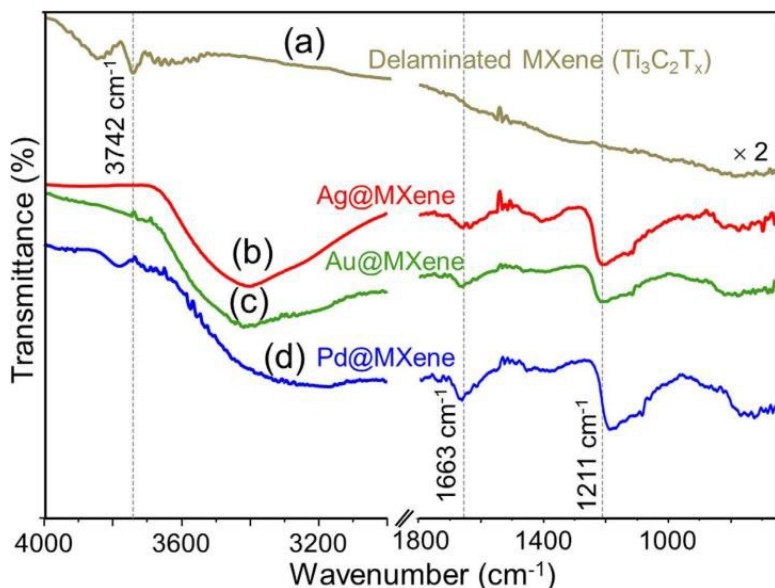

**Figure 16.** ATR-FTIR spectroscopy analysis of delaminated MXene nanosheets (**a**) (Ti$_3$C$_2$T$_x$) and (**b**) Ag@, (**c**) Au@ and (**d**) Pd@MXene hybrids. Reprinted with permission from [70]. Copyright 2016 Springer Nature.

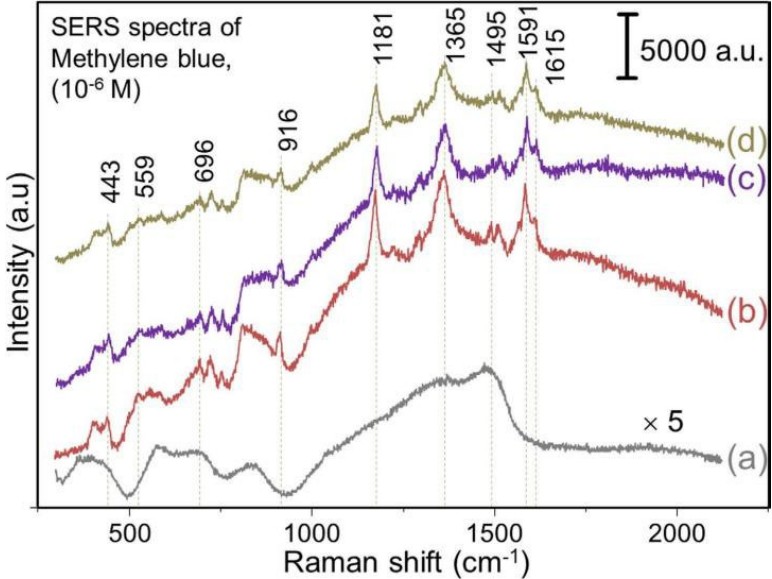

**Figure 17.** (**a**) Raman spectrum of Ti$_3$C$_2$T$_x$ after soaking in MB dispersed in ethanol and subsequent drying. SERS spectra of MB with (**b**) Ag@, (**c**) Au@ and (**d**) Pd@MXene. Reprinted with permission from [70]. Copyright 2016 Springer Nature.

The surface-enhanced Raman spectroscopy (SERS) feature patterns for δ (C–S–C) and ν (C–S) at 559 and 1181 cm$^{-1}$, correspondingly, indicate that adsorbed MB molecules had connected to the MXene hybridization plane through the sulfur–metal link, supplementing C–S bond strength. Various bands, especially for Ag@MXene composite, may be found, including (C–H) at 696 cm$^{-1}$, sym(C–N) at 1365 cm$^{-1}$, asym(C–N) at 1495 cm$^{-1}$, aromatic asym(C–C) at 1516 cm$^{-1}$, and sym(C–C) at 1591 cm$^{-1}$. Those MB spectra emerge with modest shifts in Au and Pd@MXene composites, demonstrating the CM character of such SERS impacts (Figure 17c,d). As a result, the produced 2D MXene-modified surfaces should have a high SERS enhancement. In order to compute the improvement components of 2D MXene-composite specimens, 1% liquid MB (in ethanol) had been utilized as a reference for obtaining a Raman spectrum. The improvement components computed for Ag@, Au@,

and Pd@MXene, for example, are 1.50 105, 1.17 105, and 9.61 104, respectively. This shows that the proposed material could be used in SERS applications. The SERS enhancement process of chemisorbed probe particles on a particular surface $Ti_3C_2T_x$ hybrid (in the instance of Au@ $Ti_3C_2T_x$) is being investigated in depth (chemically modified as well as electromagnetically modified) [70].

The surface structure of the rough Al and $Ti_3C_2T_x$ powders, as well as the microstructures of the forged specimens, are shown in Figure 18. As illustrated in Figure 18a, the raw Al particles were spherical, and every $Ti_3C_2T_x$ particle has stacked multilayers. The polished surface after sintering at 650 °C reveals that the $Ti_3C_2T_x$ particles are mostly found at the Al particle interfaces, as seen in Figure 18b. In the specimen forged at 650 °C, no more new phases were found. As illustrated in Figure 18c, the rupture surface evidently exhibits many layers of $Ti_3C_2T_x$ particles. A TEM picture of a multilayered $Ti_3C_2T_x$ particle is shown in Figure 19a. The Al signal in the $Ti_3C_2T_x$ grain is revealed by EDS examination (see Figure 19b), showing that aluminum atoms have diffused into the $Ti_3C_2T_x$ multilayers. $Ti_3C_2T_x$ has active places for aluminum crystalline formation in the interlayer gap zones. The element F was found in the $Ti_3C_2T_x$ multilayers, even though O had mostly been found between the $Ti_3C_2T_x$ MXene and aluminum particles. The presence of F within the multilayers shows in such a way that $Ti_3C_2F_2$ is the most common plane coating lapse of MXene. $Ti_3C_2T_x$–O group might proceed with Al to generate $Al_2O_3$.

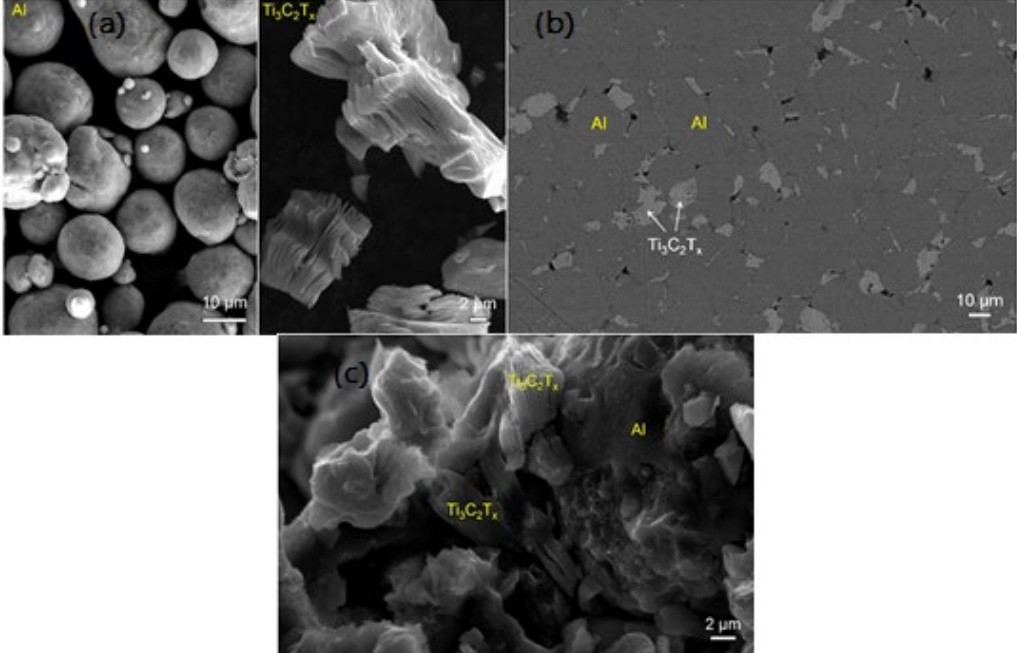

**Figure 18.** SEM micrographs of the initial powders (**a**), and polished surface (**b**) and fracture surface (**c**) of the samples sintered at 650 °C in Ar for 1 h. The left- and right-hand-side micrographs in (**a**) show the morphologies of Al and $Ti_3C_2T_x$ powders, respectively. Reprinted with permission from [71]. Copyright 2018 materials MDPI.

The microscopic TEM images of a $Ti_3C_2T_x$/Al specimen forged at 650 °C are shown in Figure 20. These $Ti_3C_2T_x$ flakes have a thickness range from a few nanometers (nm) to tens of nanometers (nm), showing that the finer particle has been made up of a minimum of two $Ti_3C_2T_x$ layers, as shown in Figure 20a. The $Ti_3C_2T_x$ particles have Al filling the interspaces. The inter-layer array of $Ti_3C_2T_x$ is about 0.855 nm, according to a high-resolution TEM (HRTEM) picture (Figure 20b). Because of the elimination of modified groups following heat processing at 650 °C, this value is lower compared to the value of 1.17 to 1.28 nm for the beginning $Ti_3C_2T_x$ flakes. As demonstrated in Figure 20c, in the HRTEM pictures of the $Ti_3C_2$/Al interface, the lattices of the $Ti_3C_2$ and Al areas are directly associated. The user interface is simple and consistent. At the contact, neither precipitates nor amorphous

patches are visible. At 650 degrees Celsius, the $Ti_3C_2$/Al contact appears to be chemically and structurally robust [71].

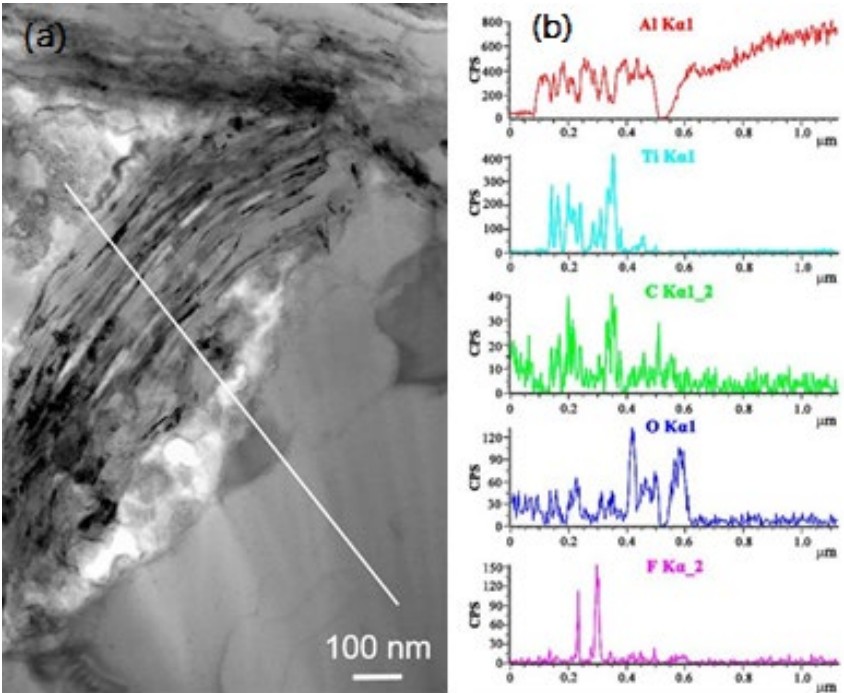

**Figure 19.** TEM image (**a**) and EDS results of $Ti_3C_2T_x$/Al sample sintered at 650 °C (**b**). The EDS scan analysis along the line in (**a**) showing the elemental distribution of Al, Ti, C, O, and F. Reprinted with permission from [71]. Copyright 2018 materials MDPI.

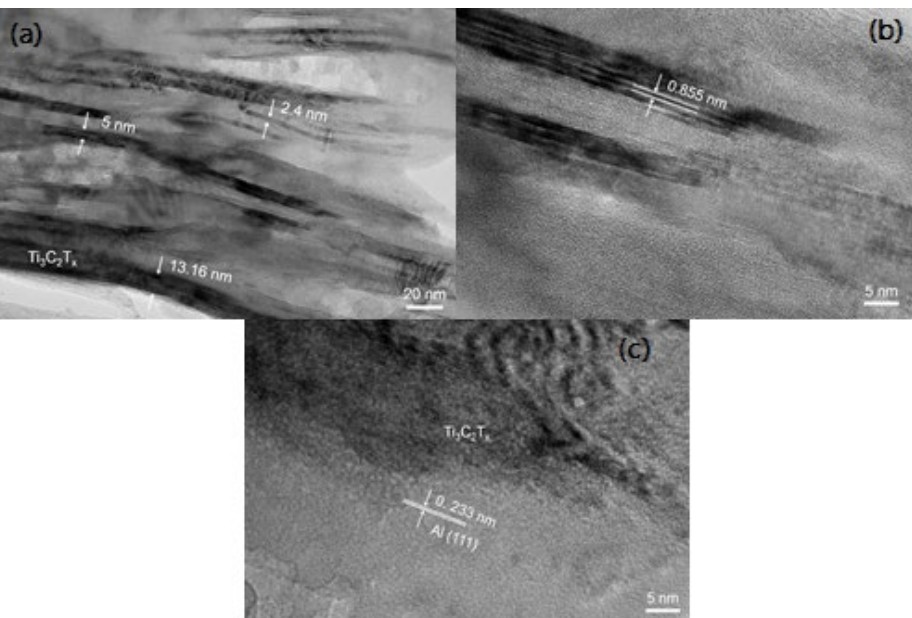

**Figure 20.** TEM images of the sample sintered at 650 °C. (**a**) A TEM micrograph showing stacked multilayers of $Ti_3C_2T_x$, (**b**) a high-resolution TEM (HRTEM) micrograph of $Ti_3C_2T_x$, and (**c**) a HRTEM micrograph of $Ti_3C_2T_x$/Al interface. Reprinted with permission from [71]. Copyright 2018 materials MDPI.

The $Ti_3C_2T_x$ MXene@Zn paper had a homogeneous Zn topping over the classical film 3D assembly, but the exposed Zn foil had pristine Zn with a polished and condensed 2D flattened construction, as observed in SEM pictures of Figure 21a. This layered structure

may aid in the acceleration of fast electron and ion transport, as well as providing a broad surface area with an adaptable place for Zn accumulation. As shown in Figure 21b–d, EDS examination revealed good dispersion of Zn over carbon and titanium of $Ti_3C_2T_x$ MXene. The position of the C 1s and Ti 2p XPS results for MXene($Ti_3C_2T_x$) and $Ti_3C_2T_x$MXene@Zn were the same, but $Ti_3C_2T_x$ MXene@Zn with large valence Ti and C–C bonds became durable, indicating an improvement in valence and binding energy. While demonstrated in Figure 21e–g, a Zn peak of 1021.45 eV also indicates the successful manufacturing. The difference between Zn foil and MXene sheet is evident, indicating that the structure of Zn foil is smooth and flat in two dimensions. After 1 h, a large amount of erect and sharpened flaky Zn with a dimension of around 10m developed on the boundary. The continuous accumulation of charges caused by the initial sharp dendrite aids in accelerating the growth of dendrite [72].

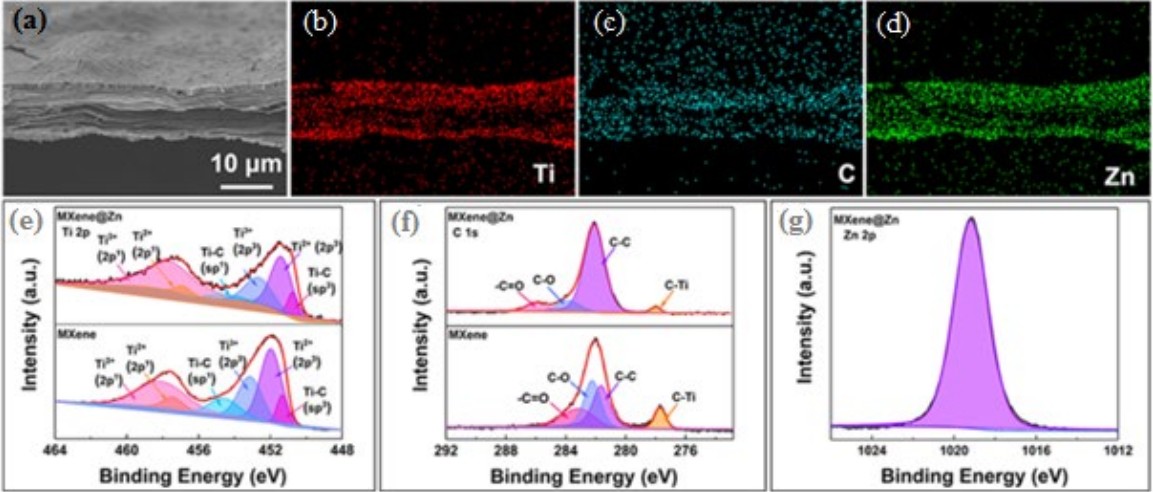

**Figure 21.** (**a**) Cross-sectional SEM image and (**b–d**) corresponding elemental distribution of $Ti_3C_2T_x$ MXene@Zn paper: (**b**) Ti (**c**) C, (**d**) Zn. High-resolution XPS spectra for (**e**) Ti 2p, (**f**) C 1s of Ti3C2Tx Mxene and $Ti_3C_2T_x$ Mxene@Zn, and (**g**) Zn $2p_{3/2}$ of $Ti_3C_2T_x$ Mxene@Zn. Reprinted with permission from [72]. Copyright 2019 American Chemical Society.

When the deposition time is increased to 10 h, closely packed Zn flaky agglomerates and also some Zn flake with a magnitude of around 40m enclose the bare Zn surface, as seen in Figure 22a–j. As the deposition period is increased up to 20 h, progressively submissive disrupted aggregates and dendrites grow on the Zn foil, resulting in a rough plane. A weak and extensive dendritic texture can be seen in the cross-sectional SEM picture (Figure 22e–i) on the naked Zn foil; a Zn dendritic with a thickness of 58m represented a significant number of sharp and upright Zn dendrites. The fast development of Zn dendrites revealed an irregular Zn coating process approach on bare Zn foil, indicating unsteady distortion behavior resulting from competing $H_2$ evolution interference. Furthermore, the dendrites may cause safety issues as well as a variety of side reactions, limiting the use of recharged zinc-based devices in an aqueous electrolyte. The surface of the $Ti_3C_2T_x$ MXene paper, on the other hand, remains smooth and flat after 1 to 20 h of plating. Even after 20 h of Zn deposition, no substantial protuberances or filaments are visible. The stacked $Ti_3C_2T_x$ MXene component and a thinner deposition matrix with a thickness of about 2 m was visible in the cross-section SEM picture of the $Ti_3C_2T_x$ MXene@Zn anode. These visual results demonstrate that the $Ti_3C_2T_x$ MXene sheet efficiently suppressed the development and modification of Zn dendrites [72].

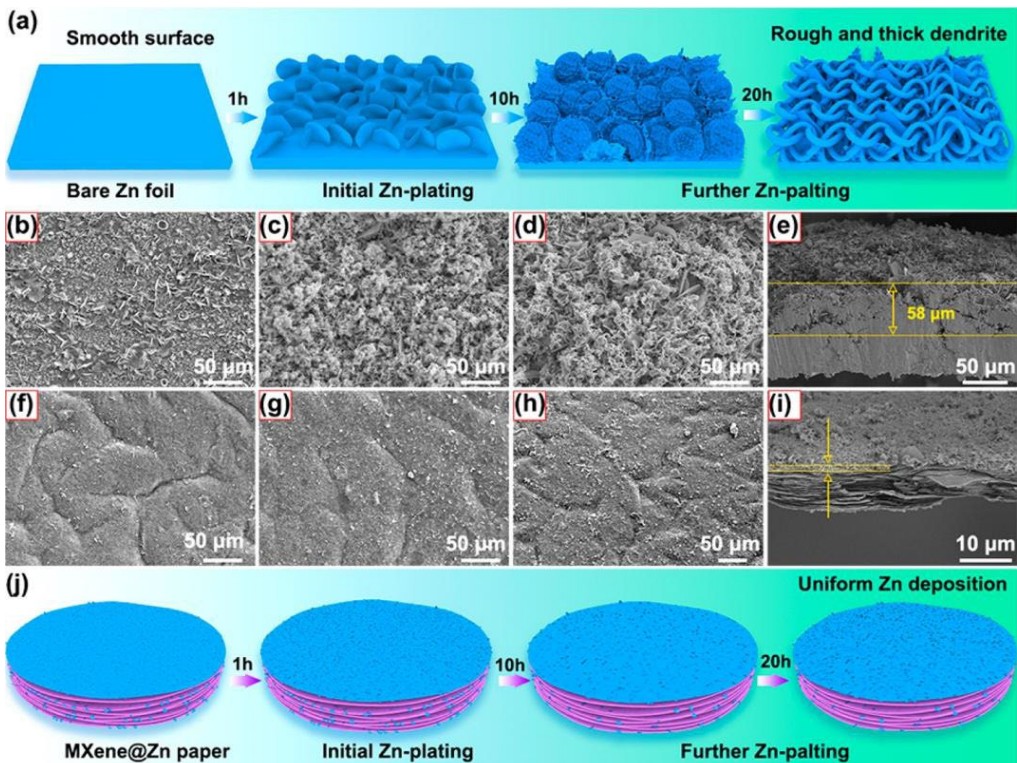

**Figure 22.** Morphological evolution and schematic illustration of Zn deposition. Schematic deposition on the (**a**) Zn foil and (**j**) $Ti_3C_2T_x$ MXene@Zn paper. Typical top-view SEM images of Zn deposition in 2 M $ZnSO_4$ electrolyte after plating capacity of (**b**) 1, (**c**) 10, and (**d**) 20 mAh·cm$^{-2}$ on Zn foil, and (**f**) 1, (**g**) 10, and (**h**) 20 mAh·cm$^{-2}$ on $Ti_3C_2T_x$ MXene@Zn paper at a current density of 1 mA·cm$^{-2}$, respectively. Cross-sectional SEM images of (**e**) commercial Zn foil and (**i**) $Ti_3C_2T_x$ MXene@Zn paper corresponding to panels d and h, respectively. Reprinted with permission from [72]. Copyright 2019 American Chemical Society.

On account of the inclusion of the $Ti_3C_2T_x$ MXene fragment, they were able to discover parameters that might originate liquid metal gelation using liquid Ga like a dummy approach. Ga melts at 29.8 °C and has an interfacial tension of 0.711 J/m$^2$. As illustrated in Figure 23a, the content of $Ti_3C_2T_x$ film in liquid Ga was built up repeatedly at 45 °C, with the ultimate concentration of $Ti_3C_2T_x$ planes in liquid Ga fix at 13 volume percent. Figure 23b suggests the frequency dependency of G′ and G″ in the direct elastomeric arrangement for $Ti_3C_2T_x$ dispersion in liquid Ga at $y_0 = 0.05\%$. Because $y_0$ is modest, it only has a minor effect on the suspension's equilibrium structure. The analogous coefficients calculated for liquid Ga covered along a narrow surface of local oxide are at least 50 times bigger than G′ and G″ for $Ti_3C_2T_x$ grains suspensions (Figure 23b). As a result, we attribute the composite's viscoelastic behaviors to the existence of MXene grains and consider the oxide surface impact to be a minor disturbance. The shear moduli of the composite are almost frequency independent, with G′ > G″ over the frequency scale tested. The creation of a colloidal gel is consistent with this. By executing continuous shear in uni-direction, the gel structure decreases with time, indicating thixotropic behavior, which is also seen in the gel system.

The behavior of ceramic grains diffused in liquid metals can be predicted using the observed aggregate and crystal events towards MXene flakes in liquid Ga. At small volume proportions, nanoflake agglomeration causes an apparent state detachment of $Ti_3C_2T_x$ in Ga, but at a large volume portion, extensive particle networks emerge. $Ti_3C_2T_x$ in Mg-Li and $TiC_{0.9}$ microparticles in Al-Mg compound showed essentially comparable behavior. To produce a homogeneous distribution of particles, significant particle loading was required in all circumstances. The enlarged particle networks gave enough stiffness

to form metals immediately in their molten state; moreover, they generated consistent composites (Figure 23c). The amount of in situ clustered crystallites within the zone among the solid and liquid lines in the equilibrium phase diagram controls the viscoelastic characteristics of metal slurries in a traditional semisolid casting. However, this approach has two key drawbacks: (i) it requires precise temperature gradient control, which is hard to achieve on a wide range, and (ii) alloys must be of a specific configuration. Bcc Mg-Li compounds are intriguing principles to explore the influence of ceramic compounds over the structural as well as mechanical properties because of their low density and good mechanical qualities. The concentration of exfoliates had increased until the liquid Mg-Li alloy displayed notable gel-like viscous behavior and it became suitable for holding its pattern against gravity, deployed on rheological investigations with MXenes in liquid Ga.

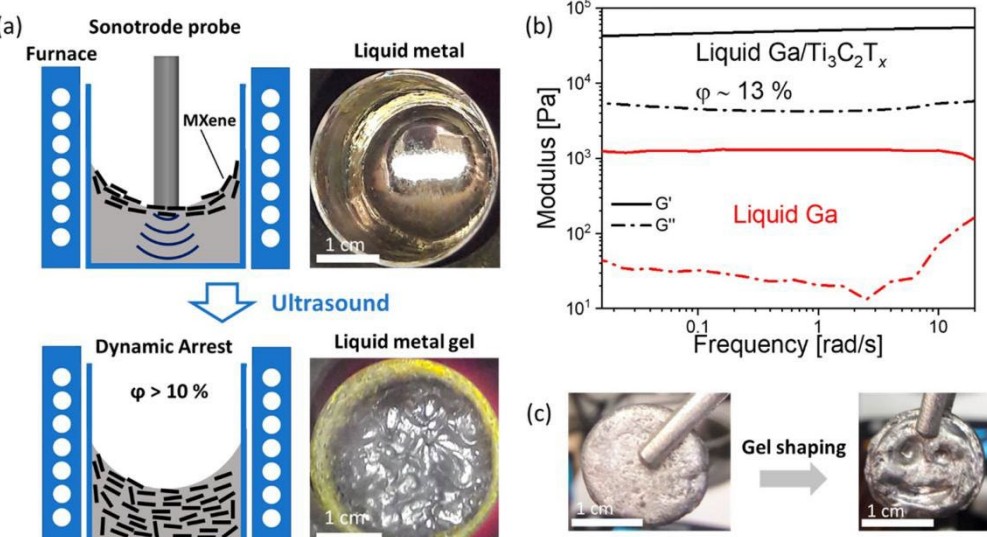

**Figure 23.** (**a**) Schematic for the preparation of $Ti_3C_2T_x$ MXene dispersions in lightweight metals and photographs of the crucibles. (**b**) Real (G) components of the complex modulus describing viscoelastic properties of liquid Ga with and without $Ti_3C_2T_x$ MXene flakes. (**c**) Demonstration of shaping the gelated composite of $Ti_3C_2T_x$ Mxene in Al-Mg alloy above the alloy melting temperature of 432 °C. Reprinted with permission from [73]. Copyright 2019 American Chemical Society.

　　High-energy ball milling (HE ball milling) was used to make MXene-Cu composite powders. Figure 24a depicts the overall look of the powders. As the ball milling duration is increased, the grain size of the composite powders enhances noticeably. After 12 h of ball milling, the cloud has reached millimeter level. The increase in particle size suggests that during the HE ball milling operation, the pulverized Cu went through critical inelastic deformation and cold welding. The optical microscope photographs of the synthesized powder milled subsequently at 3, 6, 9, and 12 h are shown in Figure 24b–e. After 3 h of milling, the Cu particles visibly link together via deformation and cold welding, as shown in Figure 24b. The grain measurement of the composite powder is around 300 m, which is substantially more than the initial Cu powder's 40 m. The cold-welding process may be seen as some small particles bond together to form a larger one, as indicated by the white arrows. As illustrated in Figure 24c, the holes still present within the composite powder particle after 6 h of milling, implying a lack of distortion and cold welding.

　　After 9 h of grinding, a few bigger grains with nearly no flaws can be seen, as indicated by the white arrows in Figure 24d. The plastic deformation and cold-welding processes further thoroughly conducted when the milling duration was increased to 12 h. As demonstrated in Figure 24e, the particle size continues to rise as the flaws in the particles decrease. SEM was used to study the shape and to further understand the MXene dispersion process, and researchers looked at the interior microstructure of the composite particles. The shape of the MXene-Cu combination after 1 h of milling is shown in Figure 25a. A minor plastic

deformation occurred in the Cu particles; these distortions had a sufficient link to the well-adjusted Cu particles. The MXene scraps can be seen visibly amongst Cu grains, showing that these powders are mixed mostly during the first step of the ball milling operation. Since the grinding duration extended to 2 h, the composite powder's inelastic deformation worsened, resulting in flat composite fragments, as illustrated in Figure 25b. There were no individual MXene flakes discovered, implying that the MXene scraps were encased in such composite fragments. After 3 h of milling, the interior microstructure of the composite flakes is shown in Figure 25c. MXene flakes on a micrometer size were still present in the composite particles. The around 100 nm MXene particles have been visible in Figure 25d, as indicated by the white arrows, implying that the bigger size MXene scraps will be clarified more by high-intensity ball milling. The earliest μ-scale MXene flakes vanished when the milling period was increased to 12 h, including the submicron MXene fragments distributed equally in composite grains, as seen in Figure 25e. Even the smallest MXene particles can be seen in the magnified image shown in Figure 25f [74].

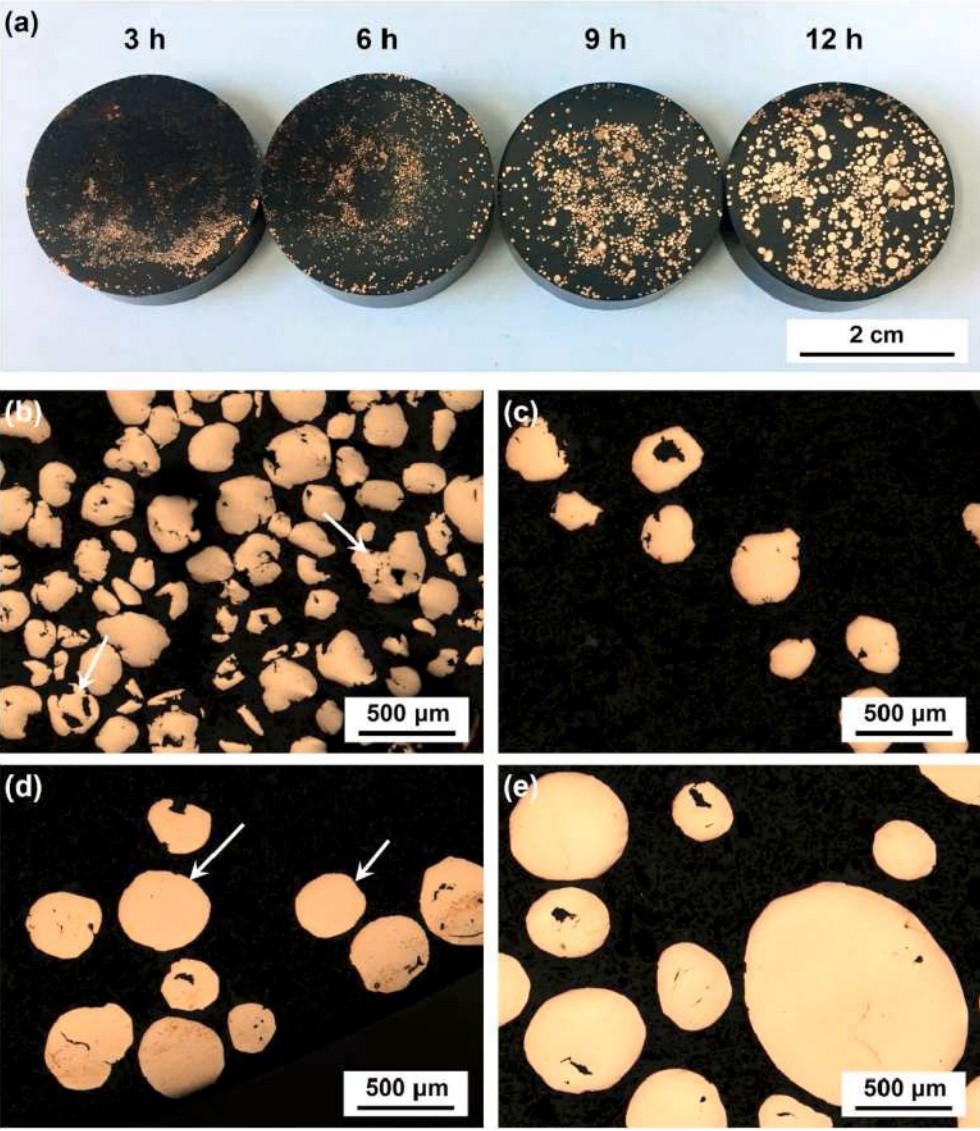

**Figure 24.** (**a**) The optical photograph of 3MXene-Cu composite powders embedded in bakelite with 3, 6, 9 and 12 high-energy ball milling, (**b**–**e**) the optical microscope photograph corresponding to the composite powder in (**a**): (**b**) 3 h, (**c**) 6 h, (**d**) 9 h and (**e**) 12 h, respectively [74].

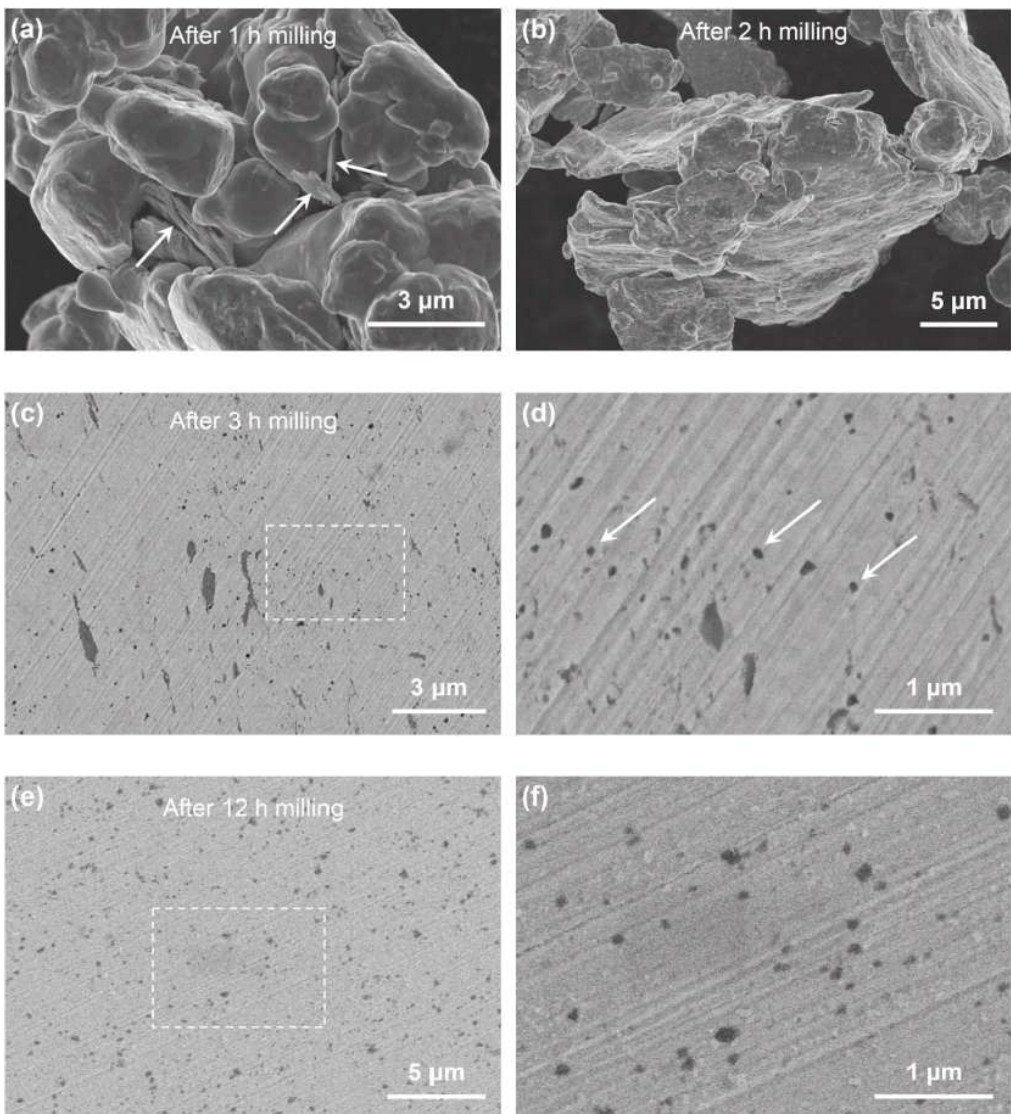

**Figure 25.** (**a**,**b**) The secondary electron SEM images of the 3MXene-Cu composite powder morphology: (**a**) 3MXene-Cu-2 h (**c**–**f**), the backscattered electron SEM images of the internal structure of 3MXene-Cu composite powders; (**c**) 3MXene-Cu-12 h, which were observed using the composite powders embedded in bakelite; (**d**) and (**f**) are the enlarged images of the rectangular areas marked in (**c**) and (**e**), respectively [74].

Figure 26a,b illustrated the microstructures of 3MXene/Cu-3 h and 3MXene/Cu-12 h compounds, correspondingly. The final MXene/Cu composites have a microstructure that is extremely similar to the equivalent MXene-Cu composite fragments, as seen in Figure 25c–e. MXene particle distribution and particle size in the Cu matrix are unaffected by the sintering process. The nanoscale MXene particles scattered in the Cu grid are shown in Figure 26c, which is a typical TEM picture of the 3MXene/Cu-12 h composite. There was also a single massive MXene fragment, around 200 nm in size.

The agglomerated particle is clearly made up of nanoscale MXene particles, as can be seen. The microstructure backs up the results from the SEM in Figure 25f. Figure 26d depicts the microstructure of the contact linking the single nanoscale MXene fragment and the Cu grid. The interface was clean, with no contaminants or flaws present, indicating robust interfacial adhesion. Figure 26e demonstrates that the SAED specimen is comparable to the region in Figure 26d. The dispersed particles had asserted to be cube-like TiC composition based on the calibration outcomes of the diffraction spots. MXene's stability was highly connected to the tempering heat, as previously stated. The MXene/Cu complex had been

synthesized at temperatures exceeding 1040 °C for more than 50 min, allowing the original MXene to convert entirely into a cubic TiC structure. For the time being, it is still referred to as MXene particles due to the unknown stoichiometry of the newly synthesized cube-like TiC, and particularly it had changed from MXene. The powdered MXene–Cu complex was added to a glazed graphite mold, and vacuum hot-pressed sintering at 1040 °C for 30 min was carried out with a uniaxial pressure of 25 MPa. The temperature of sintering had been raised at a heat rate of 1.5 °C/min at about the same pressure, and the height of each sintered specimen was observed in real time using a displacement sensor with a precision of 0.01 mm. The temperature-rising routine came to an end once the displacement enhanced by 0.2 mm, and the furnace began cooling. Simultaneously, the pressure dropped to 5 MPa, which remained constant until the temperature of the furnace dropped to 550 °C. Generally, the displacement of 0.2 mm appeared most commonly at the temperature ranging from 1070~1090 °C, which could stimulate densification of the materials [74].

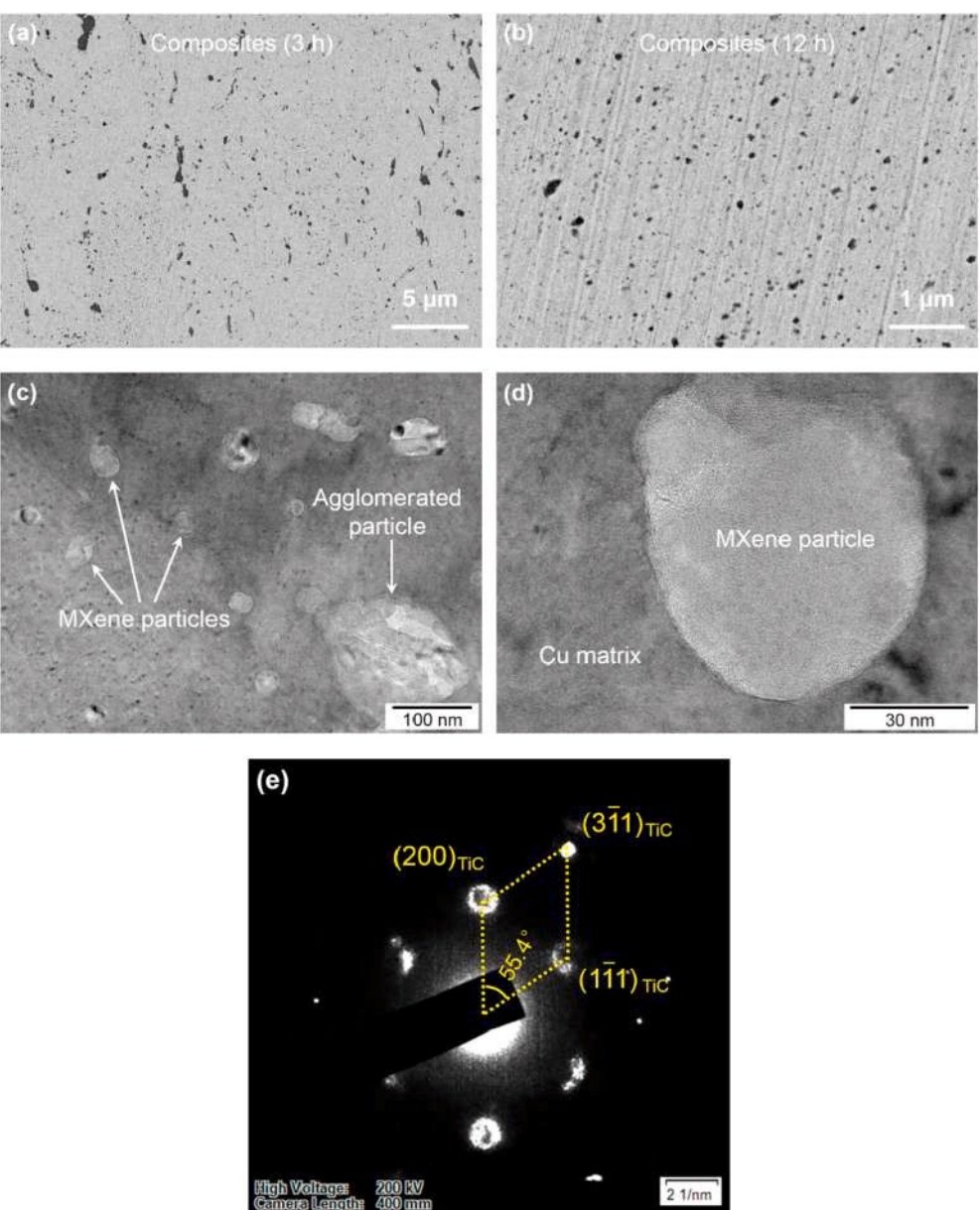

**Figure 26.** (**a**,**b**) The backscattered electron SEM images of microstructure of MXene/Cu composites: (**a**) 3MXene/Cu-12 h, (**b**–**e**) the TEM images of the microstructure of 3MXene/Cu-12 h, (**d**) the microstructure of the interface between one MXene nanoparticle and the Cu matrix, and (**e**) the selected area electron diffraction (SAED) pattern corresponding to the area of (**d**) [74].

This study was the first to use molecular-level stirring and chemical reduction to make Ni-MXene hybrids. Ball milling the hybrids and pure Cu granulate at high temperatures produced the Ni-MXene-Cu composite powder. The vacuum hot-pressing sintering of the Ni-MXene/Cu composite talc was used to further develop the initial Ni-MXene/Cu composites, as seen in Figure 27a. The white arrows indicate that only a few MXene flakes contain nickel particles. As the Ni concentration of MXene flakes rises, a high number of Ni particles develop and are scattered across the surface, as illustrated in Figure 27b. The fragment dimension is around 30 nm, and the fragments are spread equally, as seen in Figure 27c. The morphology of 15Ni-MXene hybrids is shown in Figure 27d. When comparing Figure 27b,c, it is clear that the Ni particle size of 15NiMXene is significantly greater than that of 10Ni-MXene, reaching around 100 nm. The findings also show that higher Ni concentration could allow the Ni particles to mature more fully. The 10Ni-MXene hybrids were chosen for the final Ni-MXene/Cu complex production because the nano Ni particles were produced and dispersed homogeneously at the MXene plane, as well as to prevent the hard emulsion reinforcement impact induced by very high Ni loading.

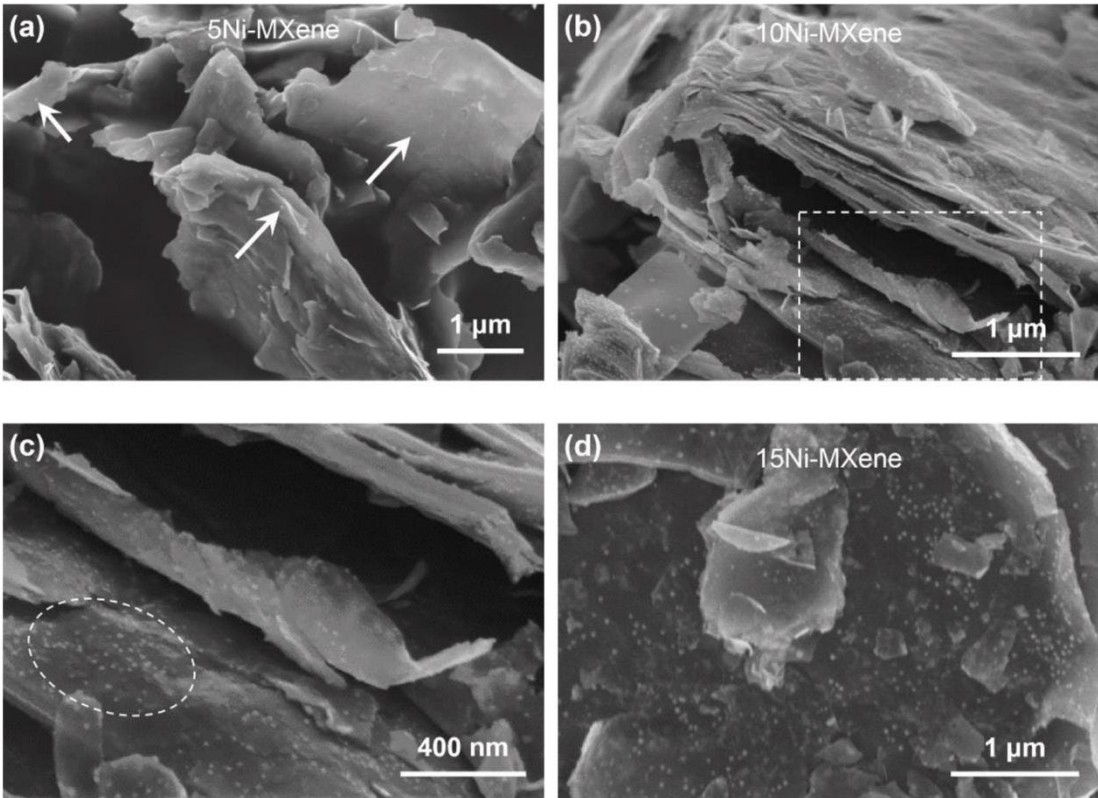

**Figure 27.** The secondary electron SEM images of Ni-MXene after annealing: (**a**) 5Ni-MXene, (**b**) 10Ni-MXene, (**c**) the enlarged image corresponding to the rectangular area marked in (**b**), and (**d**) 15Ni-MXene [75].

The morphology of 3(Ni-MXene)/Cu-3 h is shown in Figure 28a, with micron- and submicron-sized MXene molecules dispersed throughout the Cu grid. Their prior study with 3MXene/Cu-3 h yielded a similar outcome. The results show that adding Ni has less of an effect on MXene particle refining. The elemental EDS investigations are consistent with the matrix region noted in Figure 28a, which is shown in Figure 28b.

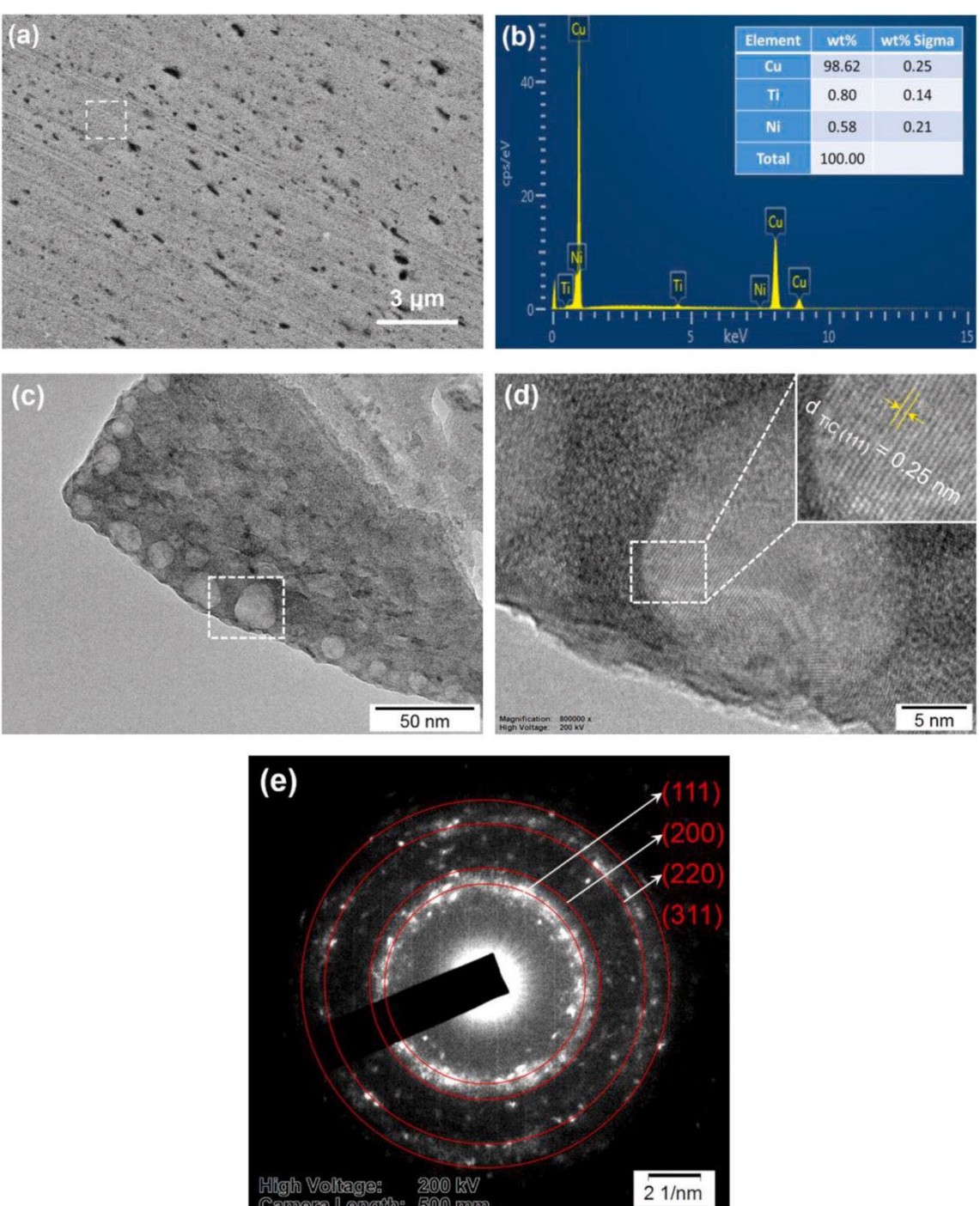

**Figure 28.** (**a**) The backscattered electron SEM image of the 3(Ni-MXene)/Cu-3 h, (**b**) EDS analysis of the rectangular area marked in (**a**), (**c**) typical TEM image of 3(Ni-MXene)/Cu-12 h, (**d**) the enlarged image corresponding to the rectangular area marked in (**c**), and (**e**) the SAED pattern corresponding to the area of (**c**) [75].

Aside from 0.8% Ti and 0.58% Ni components, the matrix is primarily made up of pure Cu. The Ni load of the hybrids and the proportion of hybrids fed to the compounds were determined, and the mass load of Ni component in the composite grid was determined to be around 0.17%, assuming that the Ni constituent had been distributed uniformly into the Cu matrix. The experimental value of 0.58% Ni concentration might be judged fair in light of the 0.21% variation. The scattered nano MXene molecules in the Cu grid might be linked to the 0.8% Ti element. An illustrative TEM picture of the 3(Ni-MXene)/Cu-12 h composite

is shown in Figure 28c. The nano MXene molecules in the Cu grid could easily be seen, and their dimension was approximately 30 nm. The highlighted area in Figure 28c is magnified in Figure 28d. The nano MXene fragment was finely bound to the grid, the interface was also transparent, and there were no flaws visible. The planar interval of the molecule was around 0.25 nm, which is virtually equivalent to the interplanar spacing of the TiC, according to the HRTEM image (111). The SAED specimen analogous to the region of Figure 28c is shown in Figure 29e. A typical polycrystalline diffraction result can be seen in the SAED pattern. The diffused nano particles had been additionally authenticated to be a cubic TiC structure based on calibration outcomes. The results are similarly compatible with those obtained previously with a 3MXene/Cu-12 h complex, implying that the inclusion of Ni has no effect on the MXene structural transition [75].

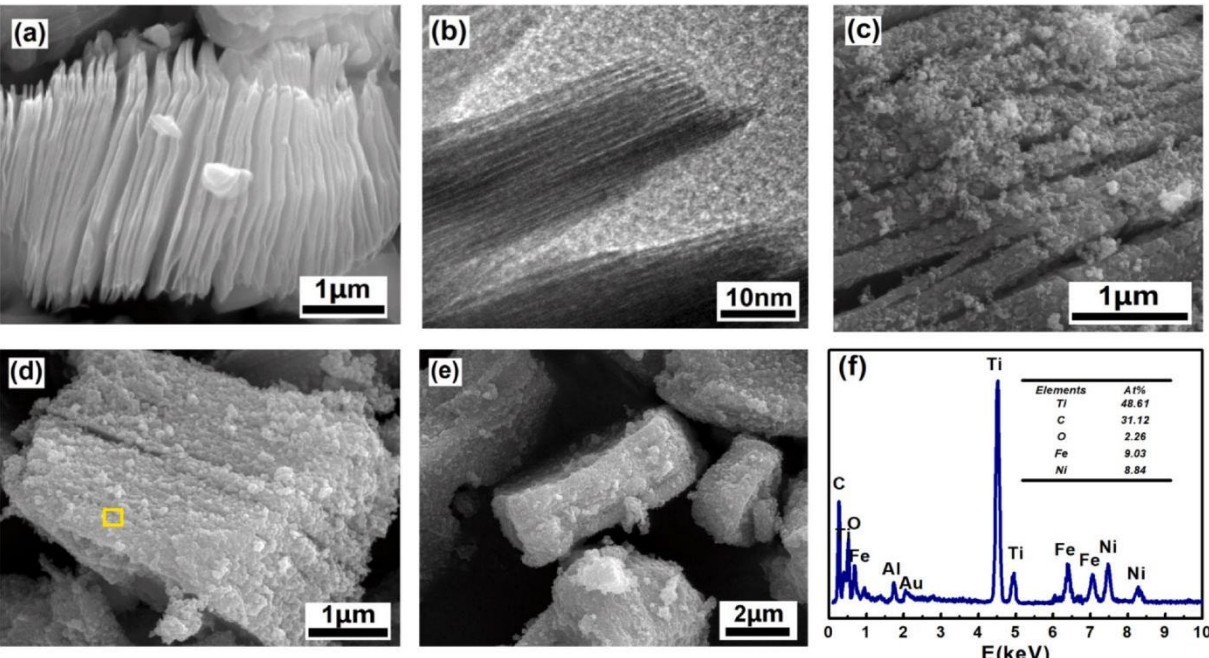

**Figure 29.** (**a**) SEM and (**b**) corresponding HRTEM images of $Ti_3C_2T_x$ MXene; SEM images of (**c**) FeNi/$Ti_3C_2T_x$ MXene-1, (**d**) FeNi/$Ti_3C_2T_x$ MXene-2 and (**e**) FeNi/$Ti_3C_2T_x$ MXene-3samples; (**f**) EDS spectrum of FeNi/$Ti_3C_2T_x$ MXene-2 [76].

Figure 29a indicates that the $Ti_3C_2T_x$ MXene that possessed laminated micromorphology had been termed thereafter the exfoliated $Ti_3AlC_2$. According to Figure 29b, the mean interlamellar space was around 10 nm. The addition of various FeNi nanoparticle concentrations to the laminated $Ti_3C_2T_x$ MXene is anticipated to adjust electromagnetic (EM) characteristics in favor of reduced magnetic loss and improved impedance match up. SEM pictures of FeNi/$Ti_3C_2T_x$ MXene composites with varied FeNi concentrations are also shown in Figure 29c–e. Many FeNi nanoparticles appear to be uniformly filled onto the surface of $Ti_3C_2T_x$ MXene or implanted in the gap connecting many layers, resulting in numerous heterostructures.

The nucleation and development of the magnetic FeNi alloy are aided by the existence of multiple aborted modified groups on the plane of $Ti_3C_2T_x$ MXene. Furthermore, as the FeNi load rises, the amount of nanoparticles coated on the $Ti_3C_2T_x$ MXene plane and interlayer without agglomeration increases. The EDS inspection of the FeNi/$Ti_3C_2T_x$ MXene-2 specimen (with 20 wt.% FeNi filling) is utilized to authenticate the elemental distribution of the composite, as shown in Figure 29f [76].

### 3.2. Mechanical Properties

It can be predicted that if the content of $Ti_3C_2T_x$ raised above 3 wt.%, then mechanical possessions of the $Ti_3C_2T_x$/Al composites might enhance (Figure 30a,b). When stress was

applied, small micro-voids formed and the Al matrix plastically deformed in the tensile test. As distortion occurs, these micro-voids expand and unite to make an elliptic fracture. The fracture of the Al matrix was instigated by the propagation of a crack along with the formation of dimples. Because of the tough interface that existed between $Ti_3C_2T_x$ and Al, the applied stress would be successively conveyed to the particles of $Ti_3C_2T_x$ throughout Al distortion.

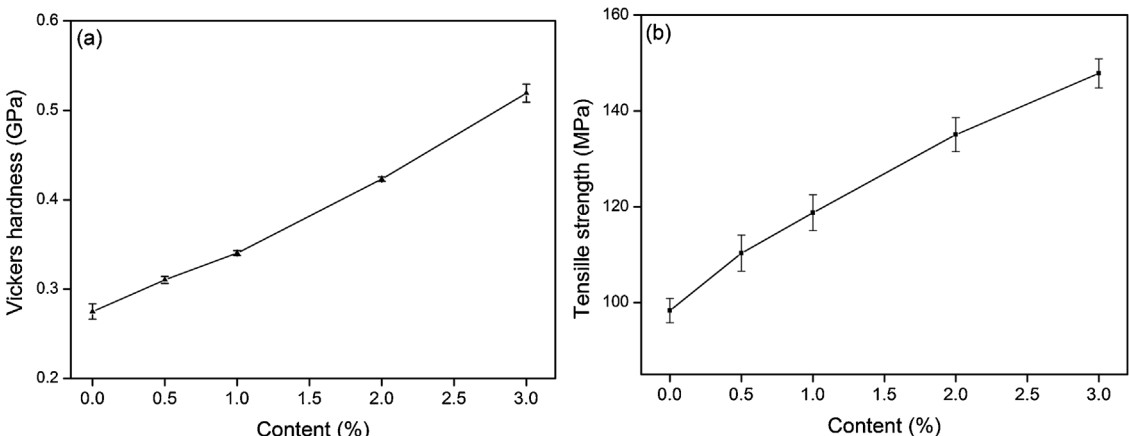

**Figure 30.** Mechanical properties of $Ti_3C_2T_x$/Al composites: Vickers hardness (**a**) and tensile strength (**b**) as a function of $Ti_3C_2T_x$ content. Reprinted with permission from [66]. Copyright 2020 Elsevier.

The reinforcement fracture took place when the maximum stress was reached. The delamination, kink coatings, and stair fracture surface of the fractured $Ti_3C_2T_x$ particles, on the other hand, spent a lot of strain energy during deformation. The mentioned fracture modes furnished the composites with upgraded mechanical properties. Pure Al exhibited a friction coefficient of about 0.49 with higher fluctuations, while the composite of $Ti_3C_2T_x$ displayed ordinary fluctuation of the friction coefficient, which is about 0.2. The hardness of pure Al was less than the $Ti_3C_2T_x$/Al composite, which is also supported by these results. The 3 wt.% $Ti_3C_2T_x$/Al composite experiences smaller plastic distortion and displays a lower friction coefficient in comparison with pure Al [66].

FLM/Al was tested for thermal expansion nature under cyclic thermal load at temperatures ranging from 323 to 573 K (Figure 31a–d). Because the cyclic behavior of the thermic evolution of the FLM/Al complex was linear and flexible, FLM-Al surfaces were thermally stable and durable. The results match the TEM of the cracked composite surface. The smashed FLM was usually discovered in sets, according to the TEM results of a detached shattered surface. This meant that the few-layered MXene (FLM) had traverse issues also subsequently diminished in the middle of the FLM receiving a load. This finding explains the effective load shift at the FLM/Al boundary caused by Al phase infiltration's anchor effect, as well as the presence of an $Al_2O_3$ phase that may have worked as a binding factor among the FLM and Al. Furthermore, unlike the fragile van der Waals interaction between the interwall of CNTs and GPLs, the FLM interlayers in AMCs had closely linked together, enhancing MXene's load-bearing capability. As a result, the FLM/Al compounds might have the appropriate mechanical properties. The ultimate tensile strength (UTS) of FLM/Al composites enhanced to an extreme value of 217.9 ± 9.5 MPa with an increased concentration of FLM at 0.26 vol.% and an elongation decrease to 15.3 ± 1.6% compared with unreinforced Al. It demonstrated that AMCs can be effectively reinforced with FLM. On the other hand, FLMs containing 0.39 vol.% have the ultimate tensile strength (UTS) of the compound reduced to 213.8 ± 10.4 MPa and the ductility seriously degenerated to 11.2 ± 3.2% [68].

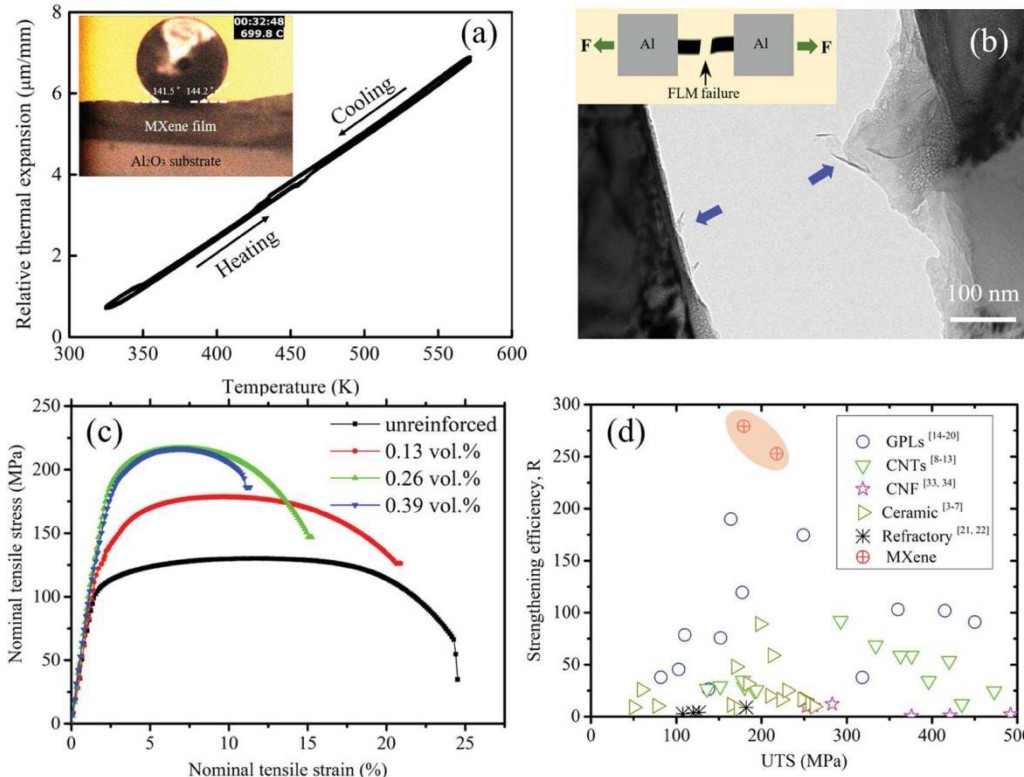

**Figure 31.** (**a**) Cyclic thermal expansion behavior of the 0.26 vol.% FLM/Al composite; (**b**) TEM image of the crack produced by bending the FLM/Al composite; (**c**) Nominal tensile stress–tensile strain curves for unreinforced Al and the FLM/Al composites; (**d**) Comparison of the strengthening efficiency of FLM and other reinforcement for AMCs. Inset in (**a**) shows the optical images of the Al bulk on the MXene film at 973K. Reprinted with permission from [68]. Copyright 2020 Taylor and Francis.

The intense binding of the $Ti_3C_2$ nanoplatelets (NPLs) with the metal matrix seen in the Mg-Li alloy reflects biaxial shear in the $Ti_3C_2$ nanoplatelets, which possessed substantial ramifications for the mechanical functioning of these MMCs. Over the standard Mg-Li alloy, the MMC produced by $Ti_3C_2T_x$ has a 128% improvement in yield toughness and a 57% rise in specific yield toughness (Figure 32a). This MMC can withstand a steadily rising load without showing symptoms of cracking. Furthermore, the repeated samples yielded repeatable results, implying that particle distribution is uniform. The MMC made with isotropic $TiC_{0.9}$ NCs can likewise withstand the load without showing symptoms of fracture formation. However, the yield toughness (71% for best specimen) and specific yield toughness (17% for best specimen) of this MMC are only moderately improved. Differences in intersections owing to interconnection among the nanoparticle planes and the metal matrix could explain this discrepancy in mechanical performance. The work of adhesion ($–W_{ad}$) computed by density functional theory (DFT) between metal and ceramic intersections is higher for polar ceramic features. $Ti_3C_2$ nanoplatelets maximize the region of polar $(111)_{fcc}$, and $(001)_{hcp}$ sides each volume of materials interacting with the metal matrix because of the KS adjustment correlation among $Ti_3C_2$ NPLs and the Mg-Li alloy.

Nonpolar surfaces contacting the metal matrix are found in $TiC_{0.9}$ nanocomposites (NCs). In comparison to isotropic $TiC_{0.9}$ NCs, robust chemical interaction at the terminal describes a higher load shift from the matrix to $Ti_3C_2$ NPLs. As previously stated, two-dimensional $Ti_3C_2$ NPLs' flexible shear in the middle of metal matrix reduces networks ill-matched at the intersection and, as a result, increases the amount of chemical bonds per unit area of the NPL matrix terminal. Unlike the MMCs with $Ti_3C_2$ NCs, the MMCs with isotropic $TiC_{0.9}$ NCs had not produced repeatable outcomes. The nonuniform distribution

of NCs is to blame for this. With attractive isotropic particles, the proportion of volume needed to accomplish colloidal gelation is larger than with seductive anisotropic particles.

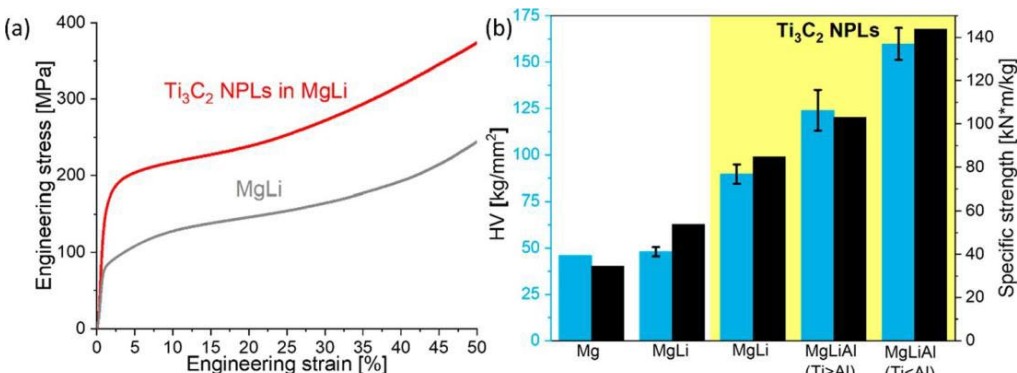

**Figure 32.** (**a**) Compressive engineering stress–strain curves for the Mg-Li alloy MMCs containing Ti$_3$C$_2$ NPLs (red) and Mg-Li alloy without any particles (gray). (**b**) Summary of the specific yield strength and Vickers hardness for the composites of Ti$_3$C$_2$ NPLs in Mg-Li and Mg-Li-Al alloys. Reprinted with permission from [73]. Copyright 2019 American Chemical Society.

The aluminum influence on Ti$_3$C$_2$ in Mg-Li composite was also investigated. Small amounts of Al are routinely employed in Mg-Li alloys to strengthen them and to promote the wetting of TiC fragments in Mg alloys. Figure 32b illustrates how the specific yield toughness and Vickers hardness (HV) of Ti$_3$C$_2$ NPLs in various Mg-Li composite quantities of additional aluminum are related at 0.98 N load. With an elemental ratio of Ti to Al 2.4/1, aluminum had not formed any intermetallic compounds with Mg or Li, implying that aluminum links with the NPL plane preferentially. The majority of the injected Al to the Ti$_3$C$_2$ NPL metal crossing is segregated, according to the EDX elemental mapping of a thin specimen. In accordance with Mg-Li alloy, the specific strength enhanced by 91% and the stiffness improved by 158%. That improvement in strength has followed with a widening of the (0002) reflection, potentially indicating that the injection of aluminum enhances individual Ti$_3$C$_2$ layer wetting. The composite, on the other hand, could only gain a maximum plastic strain of 22% prior to the initial crack that reached the specimen's plane, indicating greater fragile behavior compared to the no Al Mg-Li composites. In spite of the fact that aluminum had successfully enhanced the composite's strength without forming intermetallic compounds, more work is needed to increase the composite's plasticity. The inclusion of aluminum at a higher atomic ratio than Ti developed in the creation of an Al-Li intermetallic phase, which boosted the alloy's strength at the rate of its plasticity, with a maximum plastic shear of only 9.7%. They accepted that the process for producing stable metal–matrix composites, which is formed on colloidal gelation of liquid metals, could be applied to various metal alloy–MXene networks. Although Ti$_2$C, Ti$_3$CN, Ti$_4$N$_3$, and other Ti-terminated MXenes are likely to function similarly to Ti$_3$C$_2$, other transition metal-based MXenes should be researched. Zr$_3$C$_2$ MXene, in particular, has been proven to maintain its 2D lattice and morphology at higher temperatures than Ti$_3$C$_2$ MXene. Consequently, Zr$_3$C$_2$ MXene could be employed by supplementing MXene in alloys that need to be processed at even higher temperatures [73].

The change of ultimate tensile strength (UTS) as a trend of ball milling duration is shown in Figure 33a. The ultimate tensile strength of the 1MXene/Cu and 3MXene/Cu complex increases directly with grinding time. The UTS of the 3MXene/Cu composites increased from 202 to 314 MPa as the grinding duration rose from 3 to 12 h, which is equivalent to that of the 10Ti$_2$AlC/Cu composite. It has been demonstrated that Ti$_2$AlC can efficiently enhance Cu matrix through several mechanisms. The tempering impact of 3 vol.% MXene was closer to that of 10 vol.% Ti$_2$AlC, implying that nanoscale MXene fragments may have benefits in reinforcing the Cu grid. Figure 33b shows the relationship among prolongation and grinding time. With an increasing milling time, the prolongation

of the 1MXene/Cu and 3MXene/Cu complex likewise increases. When the grinding period reaches 9 h, the prolongation increases dramatically, principally for the 3MXene/Cu compound, where the prolongation jumps from 4.7% to 11.1%. It is also worth noting that the UTS of the 5MXene/Cu composite enhanced as grinding time sped up, later declining when grinding time exceeded 6 h. As mentioned in Figure 33a, the highest UTS of the 5MXene/Cu may hit 354 MPa. The elongation follows the same change law as the UTS, with a maximum elongation of 6.0% occurring after 6 h of milling, as illustrated in Figure 33b [74].

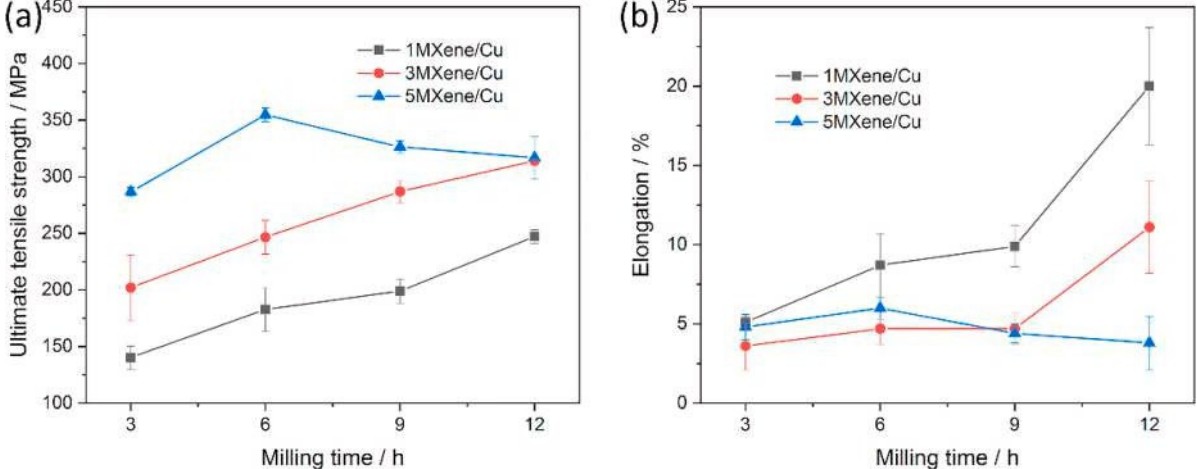

**Figure 33.** (**a**) The variation of the UTS as a function of milling time for the MXene/Cu composites with various MXene content. (**b**) The variation of the elongation as a function of milling time for the MXene/Cu composites with various MXene content [74].

The variation in 3(Ni-MXene)/Cu composites' ultimate tensile strength (UTS) and elongation as a trend of ball grinding time has been shown in Figure 34a,b, respectively. For comparison, the UTS and prolongation of the 3MXene/Cu complex has too been shown. When the milling period is prolonged from 3 h to 12 h, the UTS of 3(Ni-MXene)/Cu complex improved from 278 to 325 MPa. Furthermore, they are clearly greater compared to the 3MXene/Cu complex, particularly when reduced milling time is taken into account. According to their prior research, an aggregate of MXene particles is more likely to form composites with shorter milling times, resulting in a decrease in UTS. The current findings suggest that Ni has a remarkable effect on the structure of agglomerated MXene particles. As the grinding period rose from 3 to 6 h, the prolongation of 3(Ni-MXene)/Cu compound rose from 10.6% to 15.8%, and then remained essentially steady as the grinding time exceeded 6 h.

When compared to the 3MXene/Cu compound, the elongation of the 3(Ni-MXene)/Cu complex enhanced significantly. 3MXene/Cu-9h has a 4.7% prolongation, whereas 3(Ni-MXene)/Cu-9h has a 15.4% elongation, which is more than three times higher. At the milling period of 12 h, the highest UTS and prolongation of 3(Ni-MXene)/Cu emerge, demonstrating that the grinding duration has a significant impact on the Ni-MXene diffusion. The current findings show that increasing wettability allows Ni-MXene/Cu compounds to achieve optimal ductile characteristics in less grinding time. As a result, Ni-MXene/Cu composites have a lesser UTS than MXene/Cu complexes, and their prolongation is less, steady stabilizing after 6 h of milling time. In other words, Ni-functionalized MXene could help massive MXene fragments disperse in the sintering process by Cu and Ni interdispersion, preventing or delaying an interparticle premature crack starting and providing high UTS and elongation to composites [75].

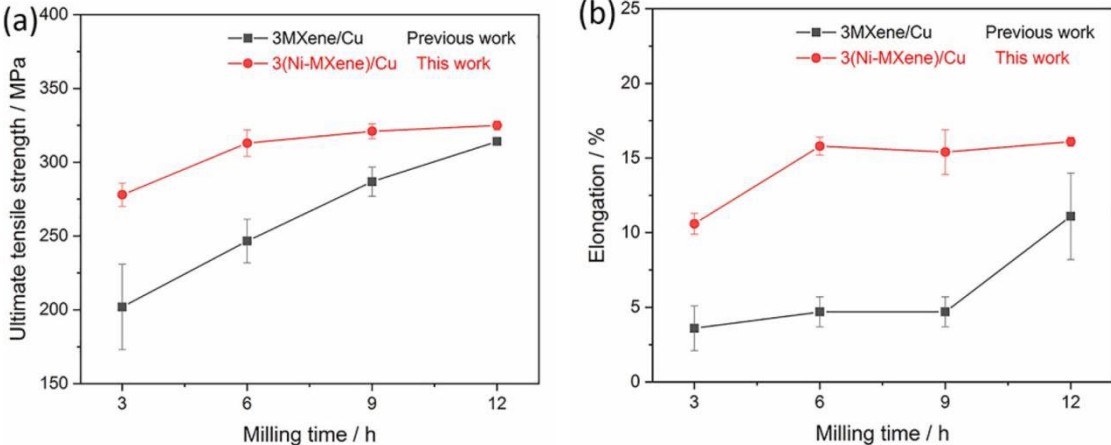

**Figure 34.** The UTS and elongation of 3(Ni-MXene)/Cu composites: (**a**) the variation of UTS with the milling time; (**b**) the variation of elongation with the milling time [75].

### 3.3. Electromagnetic Adsorption

The $Ti_3C_2T_x$/Ni spheres had attained a smaller reflection loss of $-47.06$ dB, having a width of 1.5 mm at 12.4 GHz, and the Effective Absorption Bandwidth (EAB) extends to 3.6 GHz (10.8–14.4 GHz). Magnetic loss of Ni and the conduction loss as well as the dielectric loss of $Ti_3C_2T_x$ leads to exciting properties of electromagnetic absorption. The laminated $Ti_3C_2T_x$//Ni-sphere hybrids' undistinctive laminated composition also contributed to increasing the electromagnetic absorption potentiality, that is, by virtue of the numerous reflections and scattering, to diminish microwave energy in the laminated $Ti_3C_2T_x$//Ni-sphere hybrids' laminated $Ti_3C_2T_x$//Ni-sphere hybrids' laminated $Ti_3C_2T_x$//Ni-sphere hybrids' laminated $Ti_3C_2T_x$//Ni. In short, it confirmed that all of the above-discussed properties of $Ti_3C_2T_x$/Ni-sphere hybrids have proved favorable in electromagnetic absorption [65].

$Ti_3C_2T_x$/CNZF composites have been considered as favorable candidates for electromagnetic (EM) wave absorption for stealth devices. Surface-concentrated particles have interplanar spacing of 0.479 and 0.248 nm and the average particle size is about 13 nm based on electron microscopic analysis. The $Ti_3C_2T_x$/CNZF has excellent RL and optimal thickness of about $-58.4$dB and 3.6 mm at 6.2 GHz than that of original $Ti_3C_2T_x$ and cobalt, nickel, zinc and iron (CNZF) ($-14.0$ dB at 6.9 GHz). The average permittivity of $Ti_3C_2T_x$/CNZF was mostly assigned to the unified polarization stemming from their stacked-layer, diverse and orderly mannered structures. The $Ti_3C_2T_x$/CNZF have indicated dramatic change in the tendency of permeabilities compared with the oxidized MXene. The MXene mainly revealed the dielectric loss capability for microwave attenuation, while $Co_{0.2}Ni_{0.4}Zn_{0.4}Fe_2O_4$ (CNZF) ferrite acquired classic magnetic loss property. In this manner, $Ti_3C_2T_x$/CNZF might have effective complementarities between permittivity and permeability and presented unique absorbing capability [77].

The production of ternary composites is influenced by the amount of silver nitrate treatment used. The two X- and $K_u$-band areas and the EMI shielding effectiveness for various silver, nanosized, filled $Ag$-$Ti_3C_2T_x$ and $Ag$-$Nb_2CT_x$ were studied, with the greater proportion of silver-nitrate-processed $Ag$-$Nb_2CT_x$ showing exceptional EMI shielding efficacy (72.04 dB at 18 GHz). The Ag-MXene hybrid nanostructure's EMI shielding performance could be adjusted by varying the silver charging. The discrete ternary hybrid nanomaterial can create a greatly varied substrate, increase surface area, and, most significantly, enhance electrical conductivity, which is critical for managing EMI shielding accomplishment through conduction and polarization loss. The conflicting impact among conductive loss and surface as well as interfacial polarization by the oxide nanoparticles existing on the MXene plane has a significant effect on the dissipation of EM waves. The suggested simple manufacturing method along with the outstanding EMI shielding ca-

pabilities of the Ag-MXene nanostructure may be useful in the enormous fabrication of carbon-based absorbing substances for an upcoming EMI shielding approach [69].

The main elements found were Ti, C, Fe, and Ni, and the calculated wt.% matched the specified values. The extra O observed could be due to a little amount of $TiO_2$ that is available during the hydrothermal process. There are no other contaminants found. At ambient temperature, the magnetic production of as-prepared $FeNi/Ti_3C_2T_x$ MXene composites with different FeNi concentrations is computed in an executed magnetic field range of −15 to 15 kOe. The magnetic hysteresis loops of $FeNi/Ti_3C_2T_x$ MXene-1, $FeNi/Ti_3C_2T_x$ MXene-2, and $FeNi/Ti_3C_2T_x$ MXene-3 specimens are shown in Figure 35.

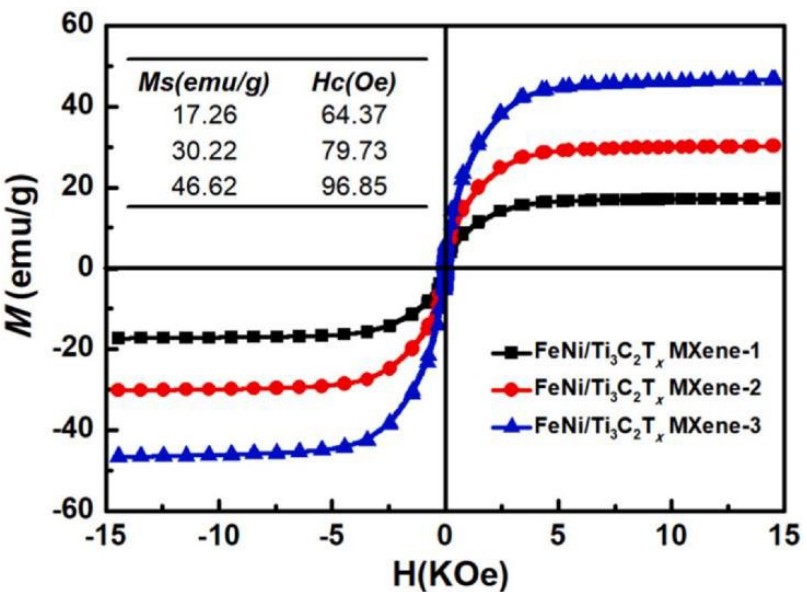

**Figure 35.** VSM of the as-synthesized $FeNi/Ti_3C_2T_x$ MXene samples with various FeNi nanoparticle loadings, the inset displays corresponding to $M_S$ and $H_C$ value [76].

All composite samples show conventional ferromagnetic activity, as well as low coercivity ($H_C$) and high saturation magnetization ($M_s$). Both the $M_s$ and $H_C$ values of the composite specimen appear to improve as the FeNi level increases, from 17.26 to 46.62 emu/g and 64.37 to 96.85 Oe, respectively, while the FeNi load rises up. The greater $M_s$ value is mostly due to the composites' excessive magnetic nanoparticle and magnetic moment. The larger FeNi alloy particle dimension generated by the improved FeNi nanoparticle concentration may have contributed to the higher $H_C$ value. FeNi/PAN and $Fe_3O_4/Ti_3C_2T_x$ composites show a similar phenomenon and come to the same conclusion. Consequently, the r value and magnetic loss capacity of the three $FeNi/Ti_3C_2T_x$ MXene compounds have strengthened in the same way, which is extremely beneficial for increased impedance match up and microwave absorption (MA) accomplishment. As shown in Figure 36a, the single $Ti_3C_2T_x$ MXene has an optimum $RL_{min}$ noted value of −17.2 dB at 15.52 GHz and an adsorption bandwidth of less than −10 dB (more than 90% microwave absorption) of just 2.4 GHz, implying bad MA accomplishment. Following FeNi alloy alteration, the $FeNi/Ti_3C_2T_x$ MXene composites definitely display superior MA accomplishment with regard to effective absorption bandwidth and absorption intensity. $FeNi/Ti_3C_2T_x$ MXene-1, $FeNi/Ti_3C_2T_x$ MXene-2, and $FeNi/Ti_3C_2T_x$ MXene-3 had effective absorption bandwidths of 5.0, 6.2, and 6.4 GHz, respectively, matching to the thicknesses of 1.4, 1.6, and 1.8 mm, as represented in Figure 36b–d. With a thickness of 1.7 mm, the ideal $RL_{min}$ value of $FeNi/Ti_3C_2T_x$ MXene-3 specimens placed at 16.64 GHz is at its most at −42.32 dB.

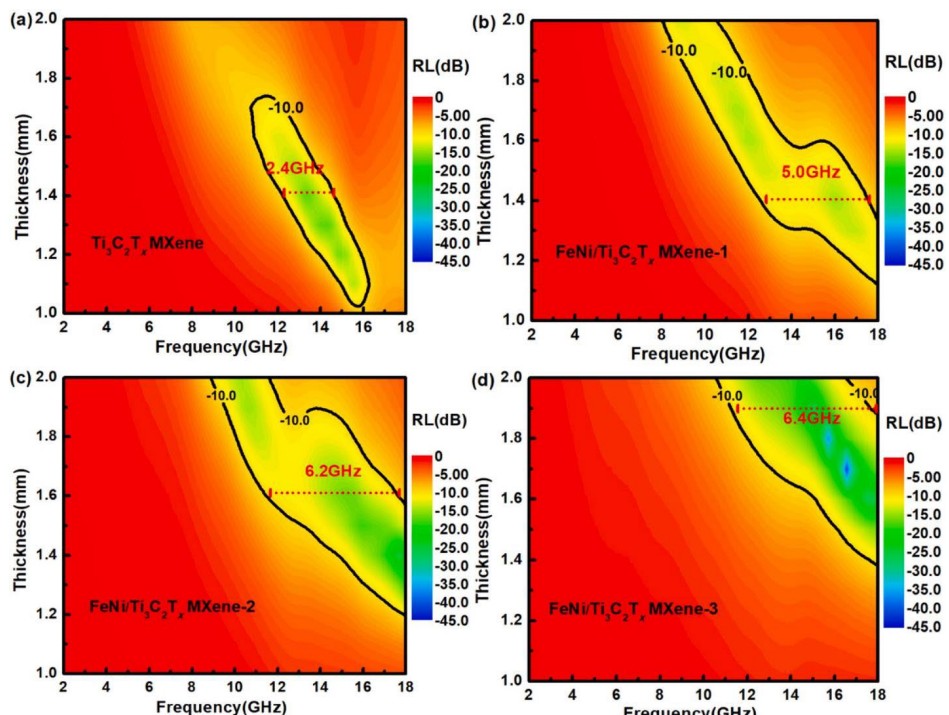

**Figure 36.** The color map of calculated RL values for the (**a**) Ti$_3$C$_2$T$_x$ MXene, (**b**) FeNi/Ti$_3$C$_2$T$_x$ MXene-1, (**c**) FeNi/Ti$_3$C$_2$T$_x$ MXene-2 and (**d**) FeNi/Ti$_3$C$_2$T$_x$ MXene-3 at thickness from 1.0 to 2.0 mm and over 2–18 GHz. For interpretation of the references to colour in this figure legend, the reader is referred to the web version of this article [76].

Figure 37 summarizes and depicts the microwave absorption (MA) processes for FeNi/ Ti$_3$C$_2$T$_x$ MXene-flled MAMs. Because of the excellent impedance correspondance of the FeNi/Ti$_3$C$_2$T$_x$ MXene-filled aligned loops promoted by a metal plate, the EM wave could primarily invade the inside and just a little electromagnetic (EM) wave return at the interface. The penetrating EM wave then suffers many reflections inside the aligned loops, which are arbitrarily spread with various laminated absorbers. As a result of the dielectric and magnetic losses, the EM energy has been efficiently absorbed and turned into heat energy. Conduction loss, electric dipolar orientation polarization, and interfacial polarization can all occur as a result of the electric field's influence on electronic, numerous exited defects, and the diversified interface, producing extensive dielectric loss. The magnetic loss is mostly due to the magnetic FeNi alloy nanoparticles' generated natural resonance effect. As a result of these findings, the FeNi/Ti$_3$C$_2$T$_x$ MXene magnetic–dielectric complex, as synthesized, could be a suitable choice for developing broadband microwave-absorbing materials (MAMs) and can be exploited in the area of radar stealth and EM resemblance [76].

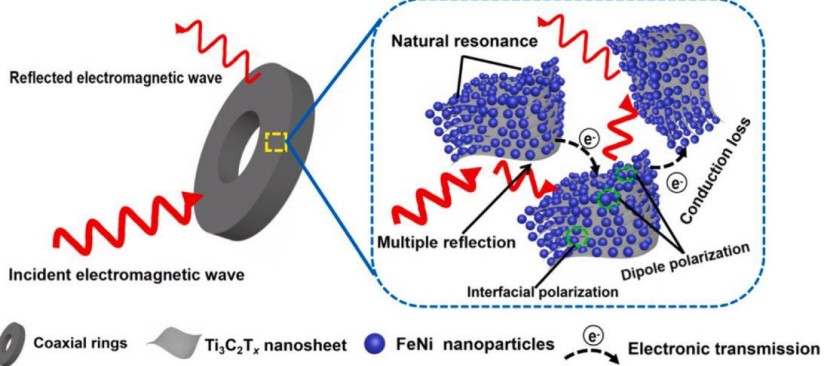

**Figure 37.** Schematic illustration of MA mechanisms for the FeNi/Ti$_3$C$_2$T$_x$ MXene-filled MAMs [76].

### 3.4. Wettability

FLM agglomerations are the result of Al particles' insufficient contact surface for anchoring a large amount of FLM sheets during powder mixing. As a result, FLM clusters served as favorable crack development locations under tensile pressures. A high-temperature wetting investigation of the FLM/Al intercourse was conducted to further understand the phenomenon. The mean contact angle of the Al spore on the MXene sheet is 142.9 degrees, indicating poor interfacial wettability. As a result, the extreme deformation of Al during densification can be attributed to FLM interlayers with an Al phase. FLM has a strengthening efficiency of 250–280, which is significantly greater than that of typical ceramics, refractory particles, fiber reinforcements, CNTs, and GPLs. The integrating properties of the undamaged FLM platelets are responsible for this superiority. FLM's excellent strengthening efficiency enables it a favorable reinforcement for great-specific-strength AMCs, and its reasonable benefits include lower material costs [68].

As shown in Figure 22, 2M $ZnSO_4$ electrolyte (6 L) was employed on exposed Zn foil and $Ti_3C_2T_x$ MXene surfaces. When compared to $Ti_3C_2T_x$ MXene paper, the bare Zn foil had a larger contact angle, implying that $Ti_3C_2T_x$ MXene paper has better wettability and electrolyte accessibility. Furthermore, the Ti@Zn electrode in the Zn | Ti@Zn cell has poor cycle stability and a low coulombic rate, whereas the $Ti_3C_2T_x$ MXene@Zn paper anode in the Zn | MXene@Zn cell has improved cycle stability and a coulombic efficiency of 94.13% throughout the plating/stripping process. At a current density of 5 mA·cm$^{-2}$, the $Ti_3C_2T_x$ MXene@Zn anode shows high cycle stability, proving the $Ti_3C_2T_x$ MXene@Zn anode's viability as a replacement for the typical Zn metal anode. The strong conductivity, superior wettability, interlayer, and specific interior chemical characteristics of $Ti_3C_2T_x$@Zn paper can all be attributed to the improved cycling stability [72].

After heating to 1200 °C, the optical picture of the hydrophilicity test between tiny Cu volume and MXene film is shown in Figure 38a. Surface tension has converted the original cubic Cu bulks into small Cu balls, and the MXene film can be easily identified due to partial breakup from the $Al_2O_3$ substrate subsequently heating. The weak binding contact among the Cu bulk and the MXene film is clearly visible in Figure 38b. When polishing the sample, the MXene film even came loose from the interface. The plane of MXene film is lower than Cu bulk, as can be seen. The Cu, Ti, and O elements mapping outcomes analogous to the region in Figure 38b are shown in Figure 38c–e. The Cu and Ti components, respectively, go properly with the Cu bulk and MXene film region. The Cu component displays limited dispersion at the contact, indicating that MXene and Cu have a low wettability. When x is smaller than 0.7, non-stoichiometric $TiC_x$ has superior wettability with Cu, according to research. The $Ti_3C_2$ MXene employed in this experiment is expected to be wetting with Cu. The function groups on the MXene surface are strongly linked to the current findings. Indeed, the oxygen adsorption on the reinforcements from the atmosphere is one of the most essential factors in preventing wetness. Figure 38f shows an extended picture of the MXene film close to the interface, where the MXene grains are broadly assembled, indicating that the Cu has not dispersed into the MXene film. A significant content of Cu has dispersed within Ni-MXene film. Furthermore, the Ni component dispersed and scattered homogeneously in the Cu bulk. Despite the fact that the Ni alteration has not removed the O functional group from the MXene plane, the alloying effect at high temperatures can cause Cu and Ni interdiffusion. Cu and Ni interdiffusion causes the MXene grains to bind side by side. The foregoing findings imply that using Ni nanoparticles to increase the wettability of MXene with Cu is a viable strategy [75].

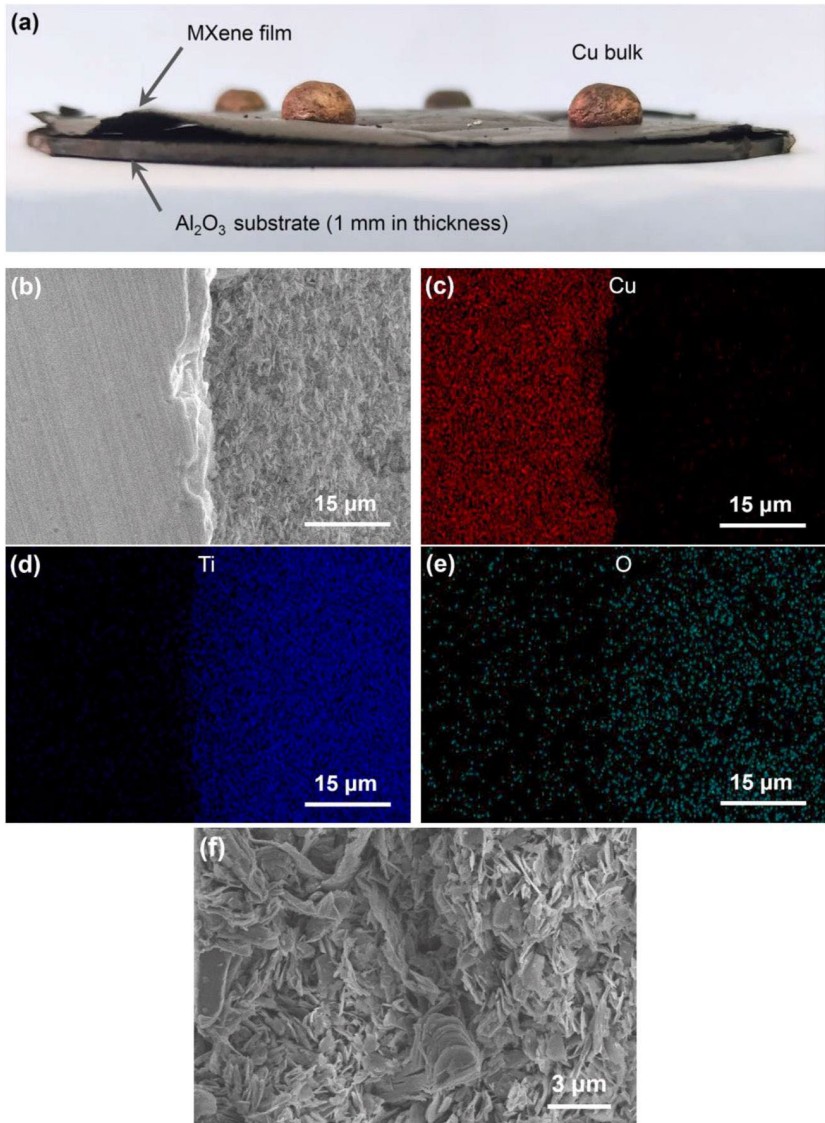

**Figure 38.** (**a**) The optical photograph of wettability test between Cu bulks and MXene film, and (**b**–**e**) the secondary electron SEM image and corresponding elemental mapping near the interface between Cu bulk and MXene film: (**b**) the SEM image, (**c**) Cu element, (**d**) Ti element, (**e**) O element, and (**f**) enlarged images of MXene film area [75].

## 4. Application of MXene/Metal Composite

In this section, we discussed the electrochemical, catalytic and anticorrosion behavior of MXene and composites of metal.

### 4.1. Electrochemical Performance

Xia, C. et al. reported that the tremendous sensing behavior of GOx/Au/MXene/Nafion/GCE has been inspected, which showed enhancement in the comparative activity of GOx in the existence of Au nanoparticles. The electrical conductivity might be increased, making the Au/MXene compound a promising electrochemical sensor/enzyme immobilization array due to the diffusion of Au nanoparticles on the plane of MXene nanosheet. The permeability of negatively charged substrates has been reduced by a negatively charged polyelectrolyte matrix named Nafion. Thus, the coating of Nafion on the GOx/Au/MXene/Nafion/GCE biosensor plane removed the effect of impeding stimulus and improved the selectivity of the sensor [62].

Wei, C. et al. investigated the electrochemical behavior of the $Ti_3C_2T_x$ MXene@Zn anode, it being a presenter for lithium metal cells to demonstrate their universality in preventing metal dendrite formation. For each cycle, the discharge areal proportion is set at 1 mAh·cm$^{-2}$, and the charging layoff potential is adjusted at 0.5 V. In the same condition, market available 2D flattened Cu foil is examined for comparison. The Coulombic effectiveness of the $Ti_3C_2T_x$ MXene@Zn@Li anode in Li metal batteries exhibits excellent reliability above 600 cycles, as can be seen in Figure 39. At a current density such as 1 mA·cm$^{-2}$ and the area capacity of 1 mAh·cm$^{-2}$, the $Ti_3C_2T_x$ MXene@Zn@Li anodes achieve a mean Coulombic effectiveness of 97.69% over 600 cycles. The Coulombic effectiveness of the $T_{i3}C_2T_x$ MXene@Zn anode is 96.79% during the 50th cycle and climbs to 99.1% during the 300th cycle, showing that the $Ti_3C_2T_x$ MXene@Zn anode has good reversibility. According to previous research, Zn might be employed as a crystalline nucleus to limit lithium dendrite formation, improving Coulombic efficiency, whereas the Cu@Li anode has a poorer Coulombic efficiency (below 90%), which fluctuates considerably after just 120 cycles. They examined the morphology evolution of $Ti_3C_2T_x$ MXene@Zn@Li anodes and Cu@Li anodes following the cycle to investigate the mechanism of $Ti_3C_2T_x$MXene@Zn@Li anode improvements. After plating/stripping towards 200 cycles at an areal magnitude of 1 mAh·cm$^{-2}$ and a current density of 1 mA·cm$^{-2}$, no evident dendrites can be seen in $Ti_3C_2Tx$ MXene@Zn@Li anodes, as shown in Figure 39f. In contrast, only after 120 cycles can damp Li development and an irregular plane be seen in Figure 39e for Cu@Li anodes, showing rapid dendritic development in the plating/stripping operation. The findings show that the $Ti_3C_2T_x$ MXene@Zn plays a good role in reducing Li dendrites. The low Coulombic effectiveness of flattened Cu electrodes is a common consequence of damp Li dendrite development that results in continual Li$^+$ dissipating and the subsequent creation of a solid electrolyte interphase (SEI) [72]. Guo, J. et al. examined the electrochemical behavior of MXene and MXene/10Ag composite, being anode materials for LIBs. Additionally, the particular CV curves for the first five cycles are expressed in Figure 40. In the initial lithiation phase, two large irreversible depletion peaks of 1.66 and 0.62 V had been found. The creation of a solid electrolyte interphase in association with Li$^+$ interpolation among the films of MXene and MXene/Ag electrode materials is thought to be the main cause. At 1.96 and 2.44 V, two wide anodic peaks were identified during the first delithiation phase. Additionally, a novel peak at 0.37 V had been discovered in the MXene/10Ag, which was assigned to lithium ion removal from the MXene/Ag electrodes.

The existence of a peak at 0.37 V could be due to the evolution of transition-state Ti in the reduction operation, which was after cycling substitutes for many stable Ti compounds. There was no discernible peak move in succeeding cycles, implying that charge storage in the two MXene and MXene/Ag electrode materials is due to Li$^+$ interpolation instead of a conversion reaction [78]. In 5 mM [Fe(CN)$_6$]3$^-$/4$^-$ carrying 0.1 M KCl, the electrochemical impedance spectroscopy (EIS) of various modified electrodes was investigated. The diameter of the semicircle corresponded to the charge transfer resistance ($R_{ct}$). The Nyquist plots of Au–Ag NSs/GCE and $Ti_2C$ MXene/Au–Ag NSs/GCE had linear lines with $R_{ct}$s close to zero, showing that Ti2C MXene/Au–Ag NSs/GCE exhibited acceptable conductivity in contrast to exposed GCE and $Ti_2C$ MXene/GCE (Figure 41) [79].

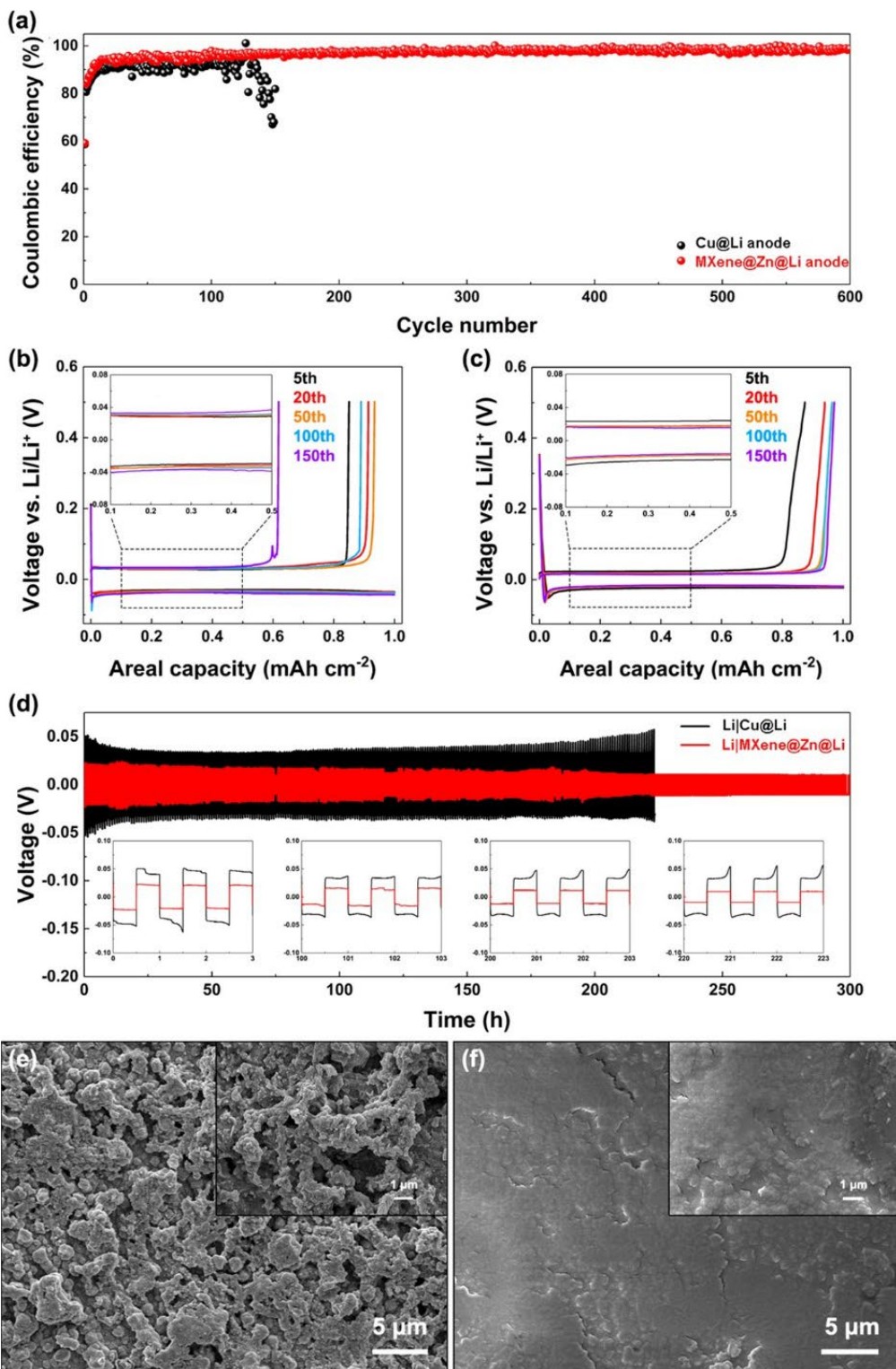

**Figure 39.** (**a**) The Coulombic efficiency of Li deposition on commercial Cu foil and $Ti_3C_2T_x$ MXene@Zn electrodes at a current density of 1 mA cm$^{-2}$ with a fixed areal capacity of 1 mAh cm$^{-2}$. The corresponding Li stripping/plating profiles for (**b**) planar Cu and (**c**) $Ti_3C_2T_x$ MXene @Zn electrodes at different cycles. (**d**) Voltage profiles of Li plating/stripping at a current density of 1 mA cm$^{-2}$ with an areal capacity of 1 mAh cm$^{-2}$ in symmetric Li | Li-Cu cells and Li | Li-MXene@Zn cells, respectively. Top view of SEM images of (**e**) Cu and (**f**) $Ti_3C_2T_x$ MXene@Zn electrodes after cycling. Reprinted with permission from [72]. Copyright 2019 American Chemical Society.

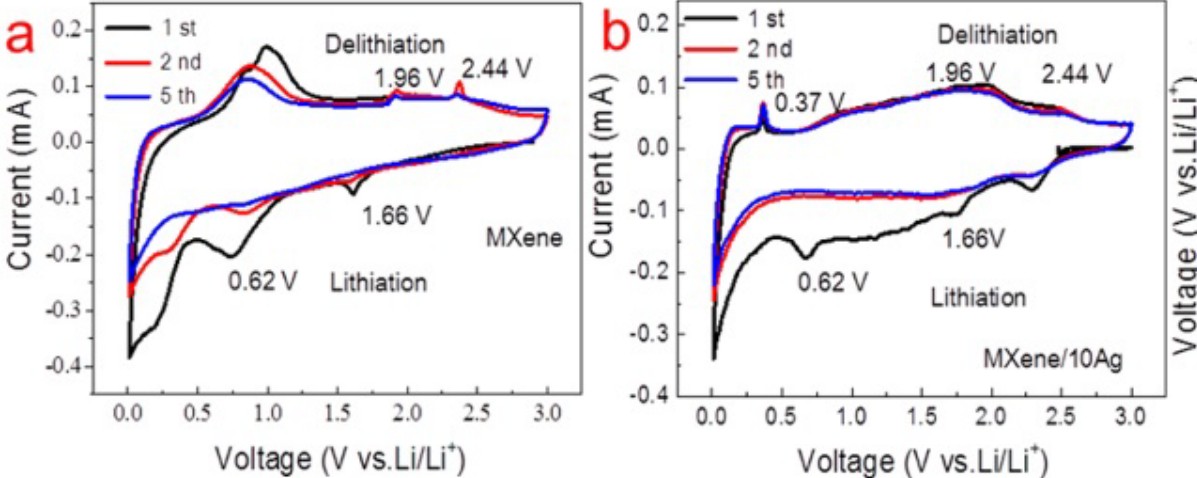

**Figure 40.** Cyclic voltammetry curves of the MXene (**a**) and MXene/10 g (**b**) samples at the initial 5 cycles. Reprinted with permission from [78]. Copyright 2016 American Chemical Society.

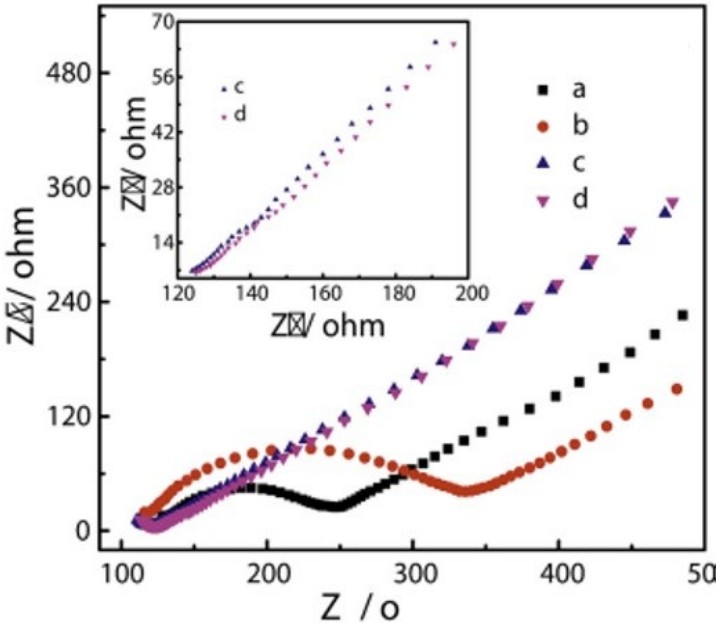

**Figure 41.** Nquist plots of bare GCE, Ti$_2$C MXene/GCE. Reprinted with permission from [79]. Copyright 2021 Elsevier.

Li, Y. et al. reported that a new, reliable, and sensitive biosensor based on MnO$_2$/Mn$_3$O$_4$ and MXene/Au NP complexes was effectively created for the electrochemical observation of OPs via an enzyme inhibitory pathway. The MnO$_2$/Mn$_3$O$_4$ hierarchical microcuboids, which are made up of vertically aligned, highly structured nanosheets, were originally made by calcining Mn-MOF like a precursor. Under ideal circumstances, AChE-Chit/MXene/Au NPs/MnO$_2$/Mn$_3$O$_4$/GCE exhibited excellent achievement for methamidophos detection, with a broad linear range of $10^{-12}$ M to $10^{-6}$ M and a lesser detection limit of $1.34 \times 10^{-13}$ M, which can be attributed to the collaborative impact of MnO$_2$/Mn$_3$O$_4$ and MXene/Au NPs composites. Furthermore, in real sample analysis, good recovery percentages (95.2–101.3%) were obtained. This sensitive sensing device has a lot of potential for on-site pesticide exposure studies and identifying other pollutants in the environment [80]. Xiong, S. et al. used a simple and versatile electrodeposition technique; they were able to successfully build stratified permeable Sb on MXene as durable, flexible, free-standing, and binder-free anodes for potassium-ion batteries (KIBs). This approach efficiently addresses the issue of the Sb anode during cycling by imparting electrodes with electrical conductivity, elec-

trochemical endurance, and structural reliability. During the potassiation/depotassisation phase, the hierarchical porous Sb might aid ion diffusion and buffer volume expansion. MXene paper, which is more conductive and flexible, acts as an elastic current collector, allowing ions to be transported and volume changes to be accommodated in the cycle process. MXene@Sb anodes for potassium-ion batteries (KIBs) have a greater reversible magnitude of 516.8 mAh·g$^{-1}$ at 50 mA·g$^{-1}$, a good rate capability of 270 mAh·g$^{-1}$ at 500 mA·g$^{-1}$, and a steady capacity retention of 79.1435% after 500 cycles with just 0.04172% capacity fading per cycle at 500 mA·g$^{-1}$. Moreover, the adaptable and simple electrodeposition technique was discovered to be capable of producing a variety of flexible and self-standing MXene@Metal (e.g., MXene@Bi, MXene@Sn) electrodes. This study could aid in the creation of flexible free-standing electrodes for rechargeable batteries, catalysts, and sensors, among other applications [81].

Shamna, I. et al. targeted his study on laccase, which was connected into the layers of MXenes via gold nanoparticles as mediators and employed as positive and negative terminals for electrochemical catechol oxidation. The results of a constructed Lac/Au/MXene/GCE electrode with a 0.3 mM catechol concentration at a scan rate of 100 mV·s$^{-1}$ at various pH values are shown in Figure 42. When comparing pH 4, 6, and 7, it was discovered that pH 5 had the highest peak current. Using a constructed Lac/Au/MXene/GCE electrode, an appropriate pH 5 was optimized for further catechol analysis. Because of the H$^+$ generated during catechol oxidation, E° of the dopant falls while pH rises. Anions (OH$^-$) inhibit enzymatic oxidation at higher pH levels.

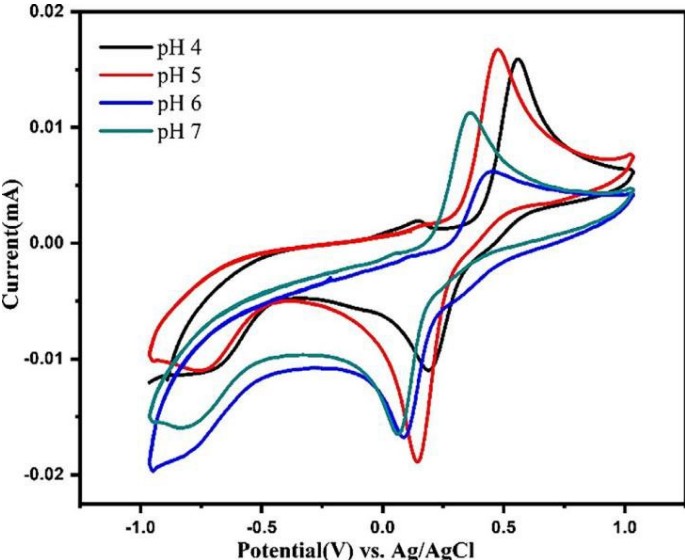

**Figure 42.** Cyclic voltammetry response of Lac/Au/MXene/GCE for 0.3 mM catechol concentration with a scan rate of 100 mV·s$^{-1}$ at pH 4, 5, 6, and 7 [82].

As a result, laccase cannot be used to perform catechol analysis at basic pH. For electrochemically reversible redox reactions, the peak current, ip, develops straight with the square root of the scan rate, in accordance with the Randles Sevcik formula. Figure 43 shows the CV response of Lac/Au/MXene/GCE at pH 5 with 0.30 mM catechol concentration at scan rates of 10, 20, 50, 100, 125, 150, 175, and 200 mV·S$^{-1}$, which illustrates the rise in loop area with an increasing scan rate and so agrees with the Randles Sevcik equation, which is a mathematical formula developed by Randles Sevcik.

$$Ip = 0.4463(F3/RT)^{1/2}An^{3/2}DR^{1/2}C_0v^{1/2} \tag{1}$$

where ip is the peak current (Ampere), R is the gas constant (8.314 J/mol K), F is Faraday's constant (96,485 C/mol, A refers to the electrode surface area (cm$^2$), for 10 mM K$_3$Fe(CN)$_6$

in 0.1 M KCl electrolyte, DR = $6.605 \times 10^{-6}$ cm$^2$·s$^{-1}$, n = 1, T is the absolute temperature (298K), $\nu$ is the scan rate (V/s) and $C_0$ is the concentration of $K_3Fe(CN)_6$ in mol/L.

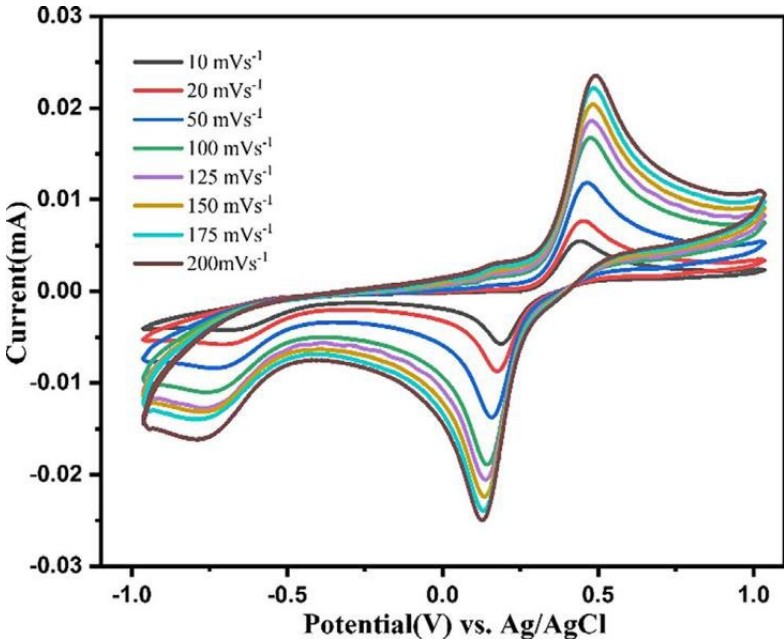

**Figure 43.** Cyclic voltammetry response of Lac/Au/MXene/GCE at pH 5 with 0.3 mM catechol concentration at scan rates of 10, 20, 50, 100, 125, 150, 175, and 200 mV·s$^{-1}$ [82].

According to the Randles Sevcik equation, ip is explicitly related to the massive concentration of catechol at a constant scan rate. The region of the redox loop rose with the rise in catechol concentration, indicating that the current enhanced with the rise in catechol concentration. As scan speeds rose, anodic peaks moved to high positive potentials, as cathodic peaks moved to high negative potentials. If the plot of peak current ranges straightly with the sweep rate (m), the limiting step is the transfer of electrons from analyte to electrode, according to the literature. There was a direct rise in peak current and a positive shift in peak potential as the scan rate was raised. On graphing the peak current (Ip) against the scan rate (v) $^{1/2}$, a straight line was obtained, conforming the equation Ipa (mA) = 0.00165 $v^{1/2}$ (mVs1)$^{-1/2}$ + 2.56831 ($R_2$ = 0.9998) in Figure 44. Their diffusion-controlled mechanism is indicated by this linear correlation. For the catechol concentrations of 0.3, 0.05, and 0.01 mM, Figure 45 compares the observation of catechol utilizing bare GCE, Au/MXene/GCE, and Lac/Au/MXene/GCE at a scan rate of 100 mV s1. The peak current related to catechol oxidation was higher for Lac/Au/MXene/GCE compared with Au/MXene/GCE and bare CE, according to the CV response. Even at 0.05 mM catechol, the enzyme laccase immobilized on Au/MXene increases on the account of catechol oxidation, resulting in a noticeable increase in peak current. From the foregoing, it may be deduced that it is a good candidate for sensing catechol traces [82].

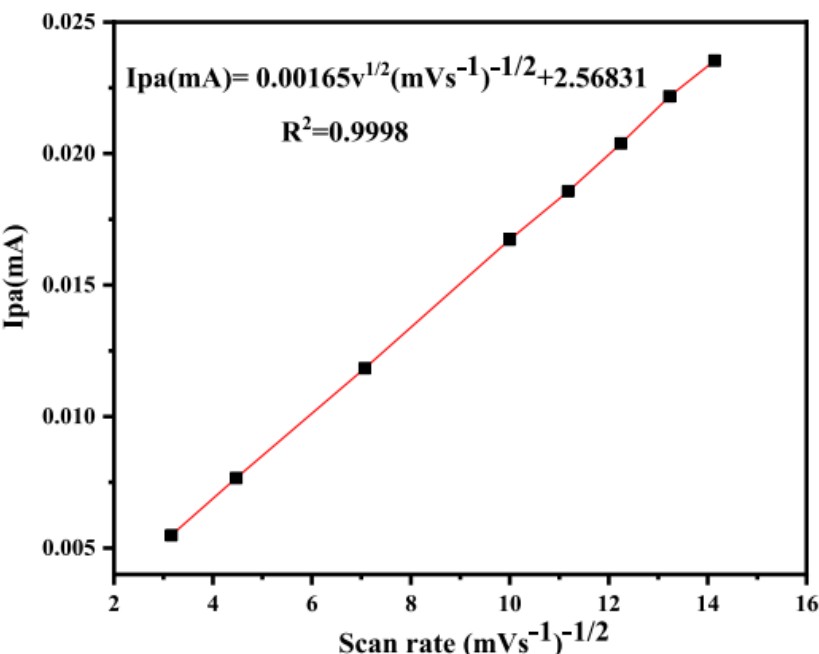

**Figure 44.** Dependence of anodic peak current on the square root of scan rate [82].

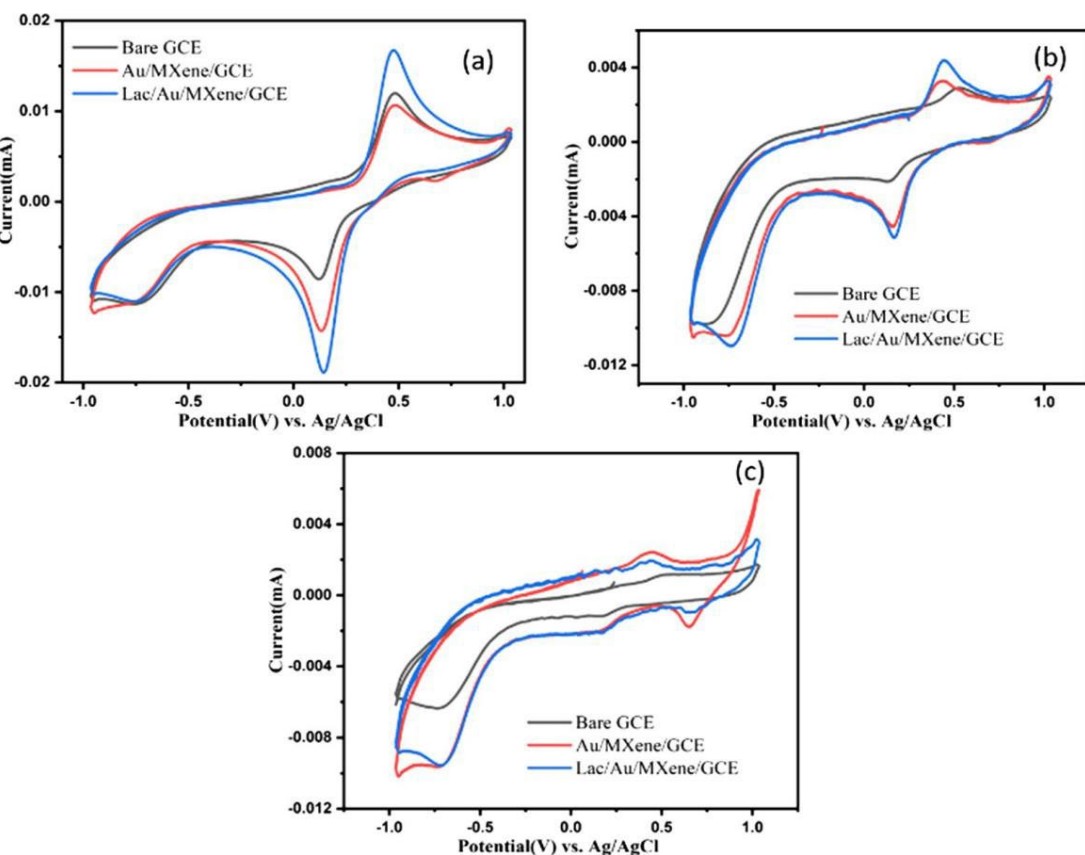

**Figure 45.** Comparison of the detection of catechol using bare GCE, Au/MXene/GCE and Lac/Au/MXene/GCE at a scan rate of 100 mV·s$^{-1}$ for the catechol concentration (**a**) 0.3 mM, (**b**) 0.05 mM, and (**c**) 0.01 mM [82].

### 4.2. Catalytic Performance

Xia, C. et al.'s research showed that Au nanoparticles attached on the plane of MXene nanofilms markedly boost up the process of transfer of electrons among GOX and GCE.

With a broad linear span of 0.1 to 18 mM, superior sensitivity of 4.2 $\mu A \cdot mM^{-1} \cdot cm^{-2}$, a lesser sensing limit of 5.9 M, and outstanding stability, repeatability, and reproducibility, the as-prepared biosensor electrode demonstrated remarkable electrocatalysis activity towards glucose sensation. All these investigations confirmed that the biosensor electrode is appropriate for measuring glucose concentrations in biological samples in the range of 0.1–1.8 mM in order to detect diabetes mellitus [62]. There were no obvious redox peaks formed under the Ar condition note from CV cures of four electrodes, indicating in the evaluated potential range that these composites were stable. In contrast, a significant distinctive decrease peak around 0.5–0.6 V, which correlates to ORR, has been detected. The MXene/NW-$Ag_{0.9}Ti_{0.1}$ catalyst with an initial $E_p$ ~0.564 V showed the best ORR activity, as shown in Table 2. When the cycle time increased, all $E_p$ values became negative. After 1000 cycles, the MXene/NW-$Ag_{0.9}Ti_{0.1}$ catalyst demonstrated the highest structural stability as well as reversibility. With faster rotation speeds, the current density rises. At a rotational speed of 1600 rpm, the polarization bandings of the further three specimens and a source of 20 wt.% Ag/C were additionally analyzed for comparison. At 1600 rmp, the onset voltage ($E_{ORR}$) and half-wave voltage ($E_{1/2}$) of the MXene/NW-$Ag_{0.9}Ti_{0.1}$ catalyst were 0.921 and 0.782 V, respectively. Each $E_{ORR}$ and $E_{1/2}$ move in a positive manner when compared to the other three samples.

**Table 2.** The important parameters determined from experimental results (1600 rpm) and some related data on Ag-based catalysts. Reprinted with permission from [83]. Copyright 2016 American Chemical Society.

| Samples | $E_{ORR}$, V | $E_{1/2}$, V | J mA·cm$^{-2}$ | n | Ep (V) | | | | Refs. |
|---|---|---|---|---|---|---|---|---|---|
| | | | | | 1 | 100 | 500 | 1000 | |
| MXene/N-Ag | 0.880 | 0.571 | 3.31 | 2.06 | 0.514 | 0.506 | 0.501 | 0.495 | [49] |
| MXene/NT-Ag | 0.901 | 0.631 | 3.34 | 2.19 | 0.526 | 0.523 | 0.512 | 0.503 | - |
| MXene/NW-$Ag_{0.9}Ti_{0.1}$ | 0.921 | 0.782 | 3.64 | 3.95 | 0.565 | 0.558 | 0.553 | 0.549 | - |
| MXene/SS-$Ag_{0.9}Ti_{0.1}$ | 0.881 | 0.554 | 2.78 | 3.15 | 0.508 | 0.504 | 0.504 | 0.499 | - |
| 20 wt.%/Ag/C | 0.88 | 0.57 | 3.29 | 3.71 | | | | | - |
| Supportless Ag nanowire | 0.92 | 0.78 | 3.51 | 3.85 | | | | | - |
| 20 wt.%/Ag/C | 0.85 | 0.56 | 3.29 | 3.70 | | | | | - |
| Ag nanorods | | 0.57 | | 3.80 | | | | | - |
| Ag-GNR | | 0.618 | | 3.51 | | | | | - |
| Ag/B-MWCNTs | | 0.69 | | 3.80 | | | | | - |
| Ag-$MnO_2$/graphene | | 0.67 | | 3.70 | | | | | - |
| Ag/GNP | | 0.72 | | 4 | | | | | - |
| Ag/$TiO_2$ | | 0.69 | | 4 | | | | | - |

The $E_{ORR}$ and $E_{1/2}$ of the MXene/NW-$Ag_{0.9}Ti_{0.1}$ catalyst was much higher compared with the reference Ag/C catalyst, implying that the MXene/NW-$Ag_{0.9}Ti_{0.1}$ specimen had excellent ORR activity. As indicated, the value is higher while comparing it with that recently published for pure silver nanowires [83]. Ding, A. et al. investigated the modified electrode. The absence of a redox peak at bare GCE and $Ti_3C_2$@GCE implies that the main electrodes are electroinactive against dopamine (DA) in the 0–0.5 V potential range. In contrast to the DNA/Pd/Pt@GCE, which has a poor current feedback, the $Ti_3C_2$/DNA/Pd/Pt@GCE has two balanced and precise redox apexes at 0.20 and 0.14 V (vs Ag/AgCl); thus, it could be ascribed to $Ti_3C_2$ high electrical conductivity. So as to analyze the impact of DNA on the electrochemical functioning of $Ti_3C_2$/DNA/Pd/Pt@GCE, $Ti_3C_2$/Pd/Pt functionalized electrode ($Ti_3C_2$/Pd/Pt@GCE) had been manufactured and executed to examine the electrochemical behavior towards DA. In the presence of DA, $Ti_3C_2$/Pd/Pt@GCE shows an electrochemical sign, as illustrated in Figure 46, although the response signal is weaker than $Ti_3C_2$/DNA/Pd/Pt@GCE. DNA-assisted nucleation could characterize the results; however, it is clear that $Pd^{2+}$ is inadequate to adsorb rightly on the hydrophobic surface of $T_{i3}C_2$, making nucleation on the $Ti_3C_2$ surface undesirable. When DNA is introduced to the $Ti_3C_2$ solution, the aromatic nucleobases of DNA are stacked on the $Ti_3C_2$ surface,

resulting in piling among the aromatic nucleobases of DNA and the hydrophobic nature of $Ti_3C_2$. $Pd^{2+}$ is adsorbed on the phosphate pillar of DNA when $PdCl_2$ solution is added, and PdNPs crystal nuclei are produced based on the $Pd^{2+}$ in the existence of reducing agents. Pd/Pt nanoparticles attach tightly to $Ti_3C_2$ with smooth scattering, increased density, and high surface area due to DNA medium, resulting in significant DA catalytic activity. $Ti_3C_2$ nanosheets functionalized with PdNPs, PtNPs, and Pd/PtNPs had also been examined for their catalytic activities. The electrodes changed with equivalent masses of nanocomposites ($Ti_3C_2$/DNA/Pd, $Ti_3C_2$/DNA/Pt, and $Ti_3C_2$/DNA/Pd/Pt), as manifested in Figure 46, have distinct current responses to DA. The current feedback at $Ti_3C_2$/DNA/Pd/Pt@GCE is clearly greater than at the other two electrodes, suggesting that the catalytic capacity of Pd/Pt nanoparticles is greater compared to PdNPs or PtNPs, which could be ascribed to the collaborative impact of PdNPs and PtNPs [64].

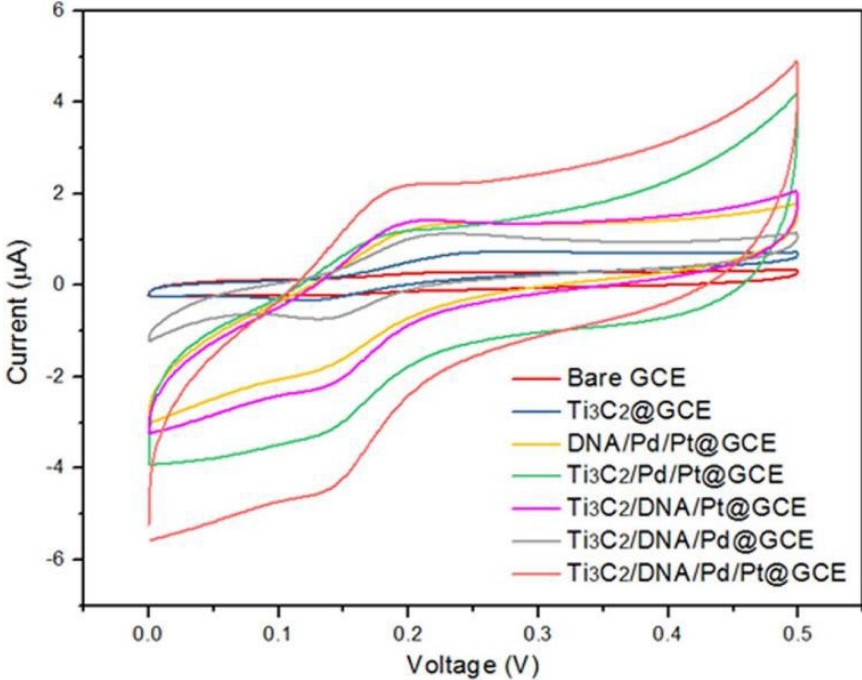

**Figure 46.** Cyclic voltammetry (CV) of bare GCE, $Ti_3C_2$@GCE, DNA/Pd/Pt@GCE, $Ti_3C_2$/Pd/Pt@GCE, $Ti_3C_2$/DNA/Pd@GCE, and $Ti_3C_2$/DNA/Pd/Pt@GCE in 0.1 M pH 7.0 PBS at a scan rate of 0.05 $Vs^{-1}$ with 1μm DA. Reprinted with permission from [64]. Copyright 2018 Elsevier.

Xing, R. et al. worked on the catalytic hydrogenation reaction of nitro-compounds, which was used to test the catalytic capacity of the MXene@AuNPs nanocomposites. The catalytic activity of the composite MXene@AuNPs20 was investigated through reducing 2-NA (2 mL, 5 mmol·$L^{-1}$) or 4-NP (2 mL, 5 mmol·$L^{-1}$) by new aqueous $NaBH_4$ (20 mL, 0.01 mol·$L^{-1}$) at ambient temperature, as measured by UV–Vis spectroscopy. The UV-Vis absorption peak of 4-NP was from 317 to 402 nm subsequently adding $NaBH_4$ to the solution, indicating the synthesis of 4-nitrophenolate. The color of the mixture of 4-NP and $NaBH_4$ does not change considerably over 24 h in the absence of a catalyst, specifying that the reaction has not taken place. The blend became colorless with time after a negligible quantity of composite MXene@AuNPs20 suspension was added. The usual peak of 4-NP at 402 nm eventually faded, indicating that 4-NP had been completely reduced (Figure 47a). Because $NaBH_4$ has a significantly greater concentration than 4-NP ($C_{NaBH_4}/C_{4-NP}$ = 400), the entire reaction operation of 4-NP might be considered a pseudo-first-order reaction. The catalytic reaction of 4-NP might be carefully examined as a pseudo-first-order reaction, where $C_t$ is the concentration at time t and $C_0$ is the initial concentration, according to a linear connection between $\ln(C_t/C_0)$ and time (t) of the MXene@AuNPs20 catalyst (Figure 47b). The pseudo-first-order reaction rate is 0.175 $min^{-1}$,

showing that the MXene@AuNPs20 compound has been a good 4-NP catalytic activity. As demonstrated in Figure 47c, the catalytic conversion remained greater than 92% after eight successive catalytic cycles, suggesting its better stability in contrast to other types of composite catalysts. They also noticed MXene@AuNPs5 and MXene@AuNPs60 complexes, catalytic blends and matched its data with a linear fit. The MXene@AuNPs5 composite can perfectly catalyze 4-NP blends in 1 h with a linear dependence. The MXene@AuNPs60 composite completely catalyzes the 4-NP combination moderately before time compared to the MXene@AuNPs5 composite still prior to the MXene@AuNPs20 composite, and order of the reaction closed to $1^{st}$ order, showing that the MXene@AuNPs20 composite is an excellent catalyst. The activity of gold nanoparticles was also tested by the catalytic depletion of 2-NA. The color of the $NaBH_4$ suspension and 2-NA solution had not changed considerably during 24 h in the absence of a catalyst, and the intensity alteration of the solution's distinctive absorption band at 415 nm was insignificant. The mixed solution turned colorless after a little quantity of the compound MXene@AuNPs20 was added, demonstrating that the MXene@AuNPs20 composite could nearly completely accelerate the solution of 2-NA mixture (Figure 47d). The MXene@AuNPs20 composite-catalyzed depletion of 2-NA can also be considered a pseudo-first-order reaction, similar to the MXene@AuNPs20 composite-catalyzed reduction of 4-NP (Figure 47e). The pseudo-first-order reaction proceeds at a rate of 0.116 $min^{-1}$, implying that the MXene@AuNPs20 compound has strong 2-NA catalytic activity. The pseudo-first-order reaction rate for 4-NP is substantially greater than for 2-NA, showing that 4-NP has a better catalytic effect than 2-NA. The catalyst's conversion rate was maintained above 95% after eight catalytic cycles (Figure 47f).

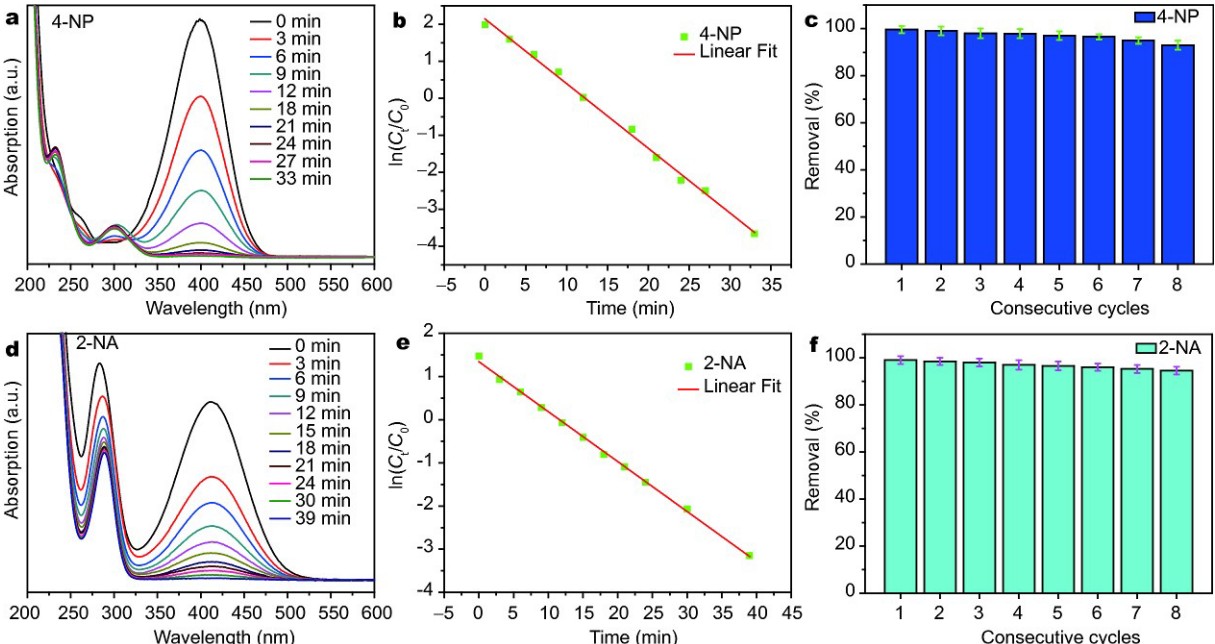

**Figure 47.** (**a**,**d**) Catalytic reduction of 4-NP or 2-NA with MXene@AuNPs20 composite. (**b**,**e**) The relationship between $\ln(C_t/C_o)$ and the reduction time (t) of the nanocomposite catalyst. (**c**,**f**) The recyclable catalysis capacities of MXene@AuNPs20 composite for the reduction reaction of 4-NP or 2-NA by $NaBH_4$ [84].

The MXene@AuNPs5 and MXene@AuNPs60 compounds could completely accelerate the 2-NA mixture in a definite amount of time; although, the time needed is a bit longer compared to the MXene@AuNPs20 composite, suggesting that the MXene@AuNPs20 composite is an extremely effective catalyst for the catalytic reaction of 2-NA. Furthermore, pure gold particles had a much lower catalytic impact on 4-NP or 2-NA than MXene@AuNPs composites, demonstrating that the MXene@AuNPs composite is an

effective catalyst [84]. Yuan, J. et al. monodispersed RhNi NPs successfully attached to MXene surfaces using a straightforward one-step wet-chemical method. The manufactured $Rh_{0.8}Ni_{0.2}$/MXene catalysts demonstrate the most reactive function to $N_2H_4H_2O$ decomposition with 100% $H_2$ selectivity and outstanding catalytic performance of 857 $h^{-1}$ at 50 °C by adjusting the percentage of Rh element in the Rh-Ni system. $Rh_{0.8}Ni_{0.2}$/MXene also outperforms $Rh_{0.8}Ni_{0.2}$/GO, $Rh_{0.8}Ni_{0.2}$/XC-72, $Rh_{0.8}Ni_{0.2}$/MCNTs, $Rh_{0.8}Ni_{0.2}$/$Al_2O_3$, and $Rh_{0.8}Ni_{0.2}$/$Al_2O_3$ in terms of catalytic performance. The parameters of metallic particle size, catalyst amount, support effect, and NaOH concentration have beneficial effects on the reaction rate of $N_2H_4H_2O$ reduction catalyzed over $Rh_{0.8}Ni_{0.2}$/MXene, according to kinetic experiments. Furthermore, the developed nano catalysts have a long lifetime of $N_2H_4H_2O$ breakdown. Other MXene-supported metal NPs can be easily prepared using the same simple synthetic technique [63]. In Liu, J. et al.'s study, the catalytic achievement of the $Ni/MoO_2@Mo_2CT_x$ composite was next tested in a palmitic acid hydrodeoxygenation (HDO) model reaction. The influence of various Ni loadings on palmitic acid HDO had been first inspected (Figure 48), and the outcomes show that a hypothetical Ni charging of 20 wt.% for $Ni/MoO_2@Mo_2CT_x$ is optimum in their reaction network because it has the greatest selectivity of hexadecane (C16) and complete palmitic acid conversion. The influence of temperature on palmitic acid HDO had been tested, as shown in Figure 48. When the temperature exceeds 240 °C, the conversion of palmitic acid rapidly increases. The primary result of the reaction at 240 °C is hexadecanol, which departs as the temperature extends to 280 °C with a 100% conversion. Consequently, at 280 °C, the ratio of hexadecane/pentadecane (C16/C15) is 1.39, resulting in a high selectivity of 97.09% towards alkane. This finding demonstrates that the $Ni/MoO_2@Mo_2CT_x$ catalyst has a higher activity for the scission of C–O bonds than for C–C bonds. When the temperature is increased to 300 °C, the $C_{16}/C_{15}$ ratio decreases and the selectivity of cracking products rises to 7.26% [85].

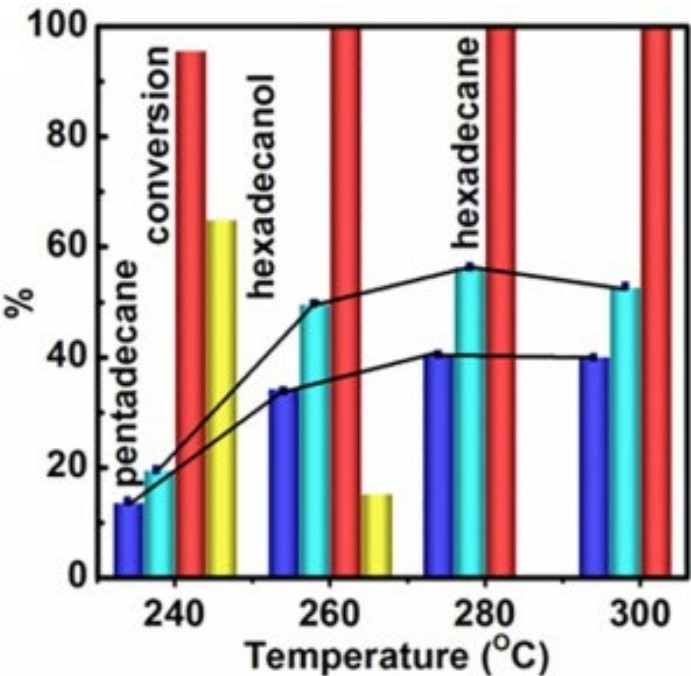

**Figure 48.** Conversion and selectivity for the main products as a function of reaction temperature over $Ni/MoO_2CT_x$ catalyst (4 MPa $H_2$, 240 min). Reprinted with permission from [85]. Copyright 2020 Elsevier.

### 4.3. Corrosion Resistance

Through one-step hydrothermally chemical conversion, a complex design MXenes/MgAl-LDH along with $Y(OH)_3$ had been in situ produced like a sharp corrosion-resistant layer on

AZ31 Mg alloy. Figure 49 shows that the greatest negative $E_{corr}$ ($-1.59$ VSCE) and highest $i_{corr}$ ($1.40 \times 10^{-5}$ A·cm$^{-2}$) were found on the bare AZ31 substrate, showing extremely low corrosion opposition. The $E_{corr}$ of the AZ31 + FLMs and AZ31 + Y + FLMs were around $-0.31$ and $-0.36$ V$_{SCE}$, respectively, indicating a large leap to good voltage, signifying the formation of a preservative layer on the plane of magnesium that reduces the probability of corrosion start. Furthermore, the $i_{corr}$ values of the exposed AZ31 coating ($1.40\ 10^{-5}$ A·cm$^{-2}$), AZ31 + FLMs ($9.85\ 10^{-7}$ A·cm$^{-2}$), and AZ31 + Y + FLMs ($9.12\ 10^{-9}$ A·cm$^{-2}$) declined in order, indicating that anticorrosive behavior improves.

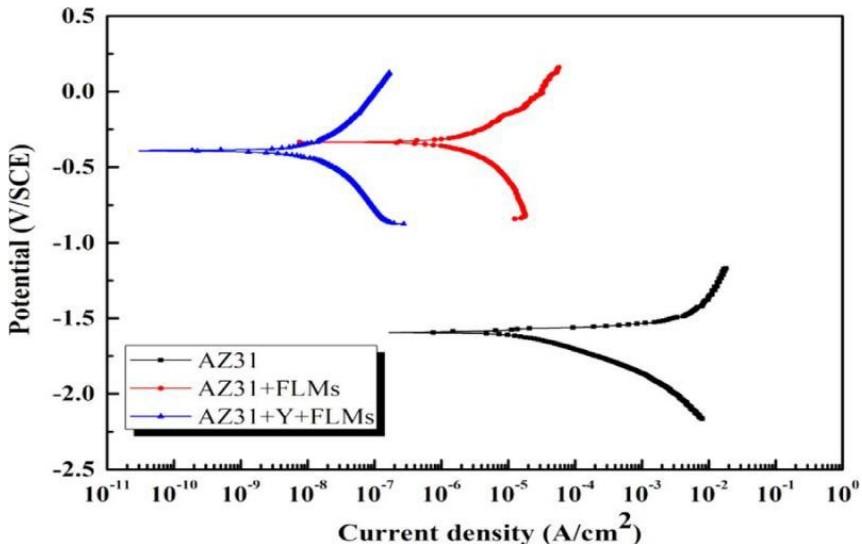

**Figure 49.** Potentiodynamic polarization curves of bare AZ31, AZ31 + FLMs and AZ31 + Y + FLMs measured after immersion for 1 h in 3.5 wt.% NaCl solution. Reprinted with permission from [86]. Copyright 2021 Elsevier.

Digital pictures of the exposed AZ31 dopant, AZ31 + FLMs, and AZ31 + Y + FLMs over the course of 15 days of absorption are shown in Figure 50. The erosion of the naked AZ31 Mg alloys was obviously drastic. The initial substratum for AZ31 + FLMs and AZ31 + Y + FLMs, on the other hand, are gray and blazing black, and their planes are homogeneous, condensed, continuing, and perfect. Local corrosion began at the bottom right side of the AZ31 + FLMs and grew to the center following 15 days of absorption. The microscopic tunnel gaps and edge stress corrosion are the primary causes of local corrosion. After 15 days of immersion, only minor rough erosion took place on the plane of AZ31 + Y + FLMs, which would be ascribed to binary factors: the magnificent physical obstruction influence of the coating, as evidenced by the thick morphology, and significant thickness of the sharp corrosion shielding procedure, as evidenced by the mobile seizing of corrosive species through chemical reaction.

The hydrogen evolution proportion could be acquired coincidently throughout the absorption trial in 3.5 wt.%NaCl aqueous solution, as illustrated in Figure 51. During 15 days of immersion, it was discovered that the hydrogen development proportion of the non-coated sample is substantially compared to the laminated sample, and that its slant increases progressively. As a result, the corrosion rate of the bare AZ31 substratum is more and moderately improves. On AZ31 + FLMs, a reduced hydrogen evolution proportion and flatten slants are identified in contrast to the naked AZ31 dopant, showing that the coating could successfully preserve the substrate from corrosion. After 15 days of immersion, the hydrogen evolution proportion of AZ31 + Y + FLM coating is only 0.026 mL·cm$^{-1}$, indicating excellent corrosion prevention. Furthermore, the slope, which represents corrosion rate, is nearly zero, indicating that the process of corrosion is reduced rather than increased, highlighting the corrosion inhibitor's action. As a result, the following is a ranking of

anticorrosion ability in increasing order: AZ31 + FLMs AZ31 + Y + FLMs AZ31 + Y + FLMs AZ31 + Y + FLMs AZ31 + Y + FLMs AZ31 + Y + FLMs AZ31 + Y + FLMs [86].

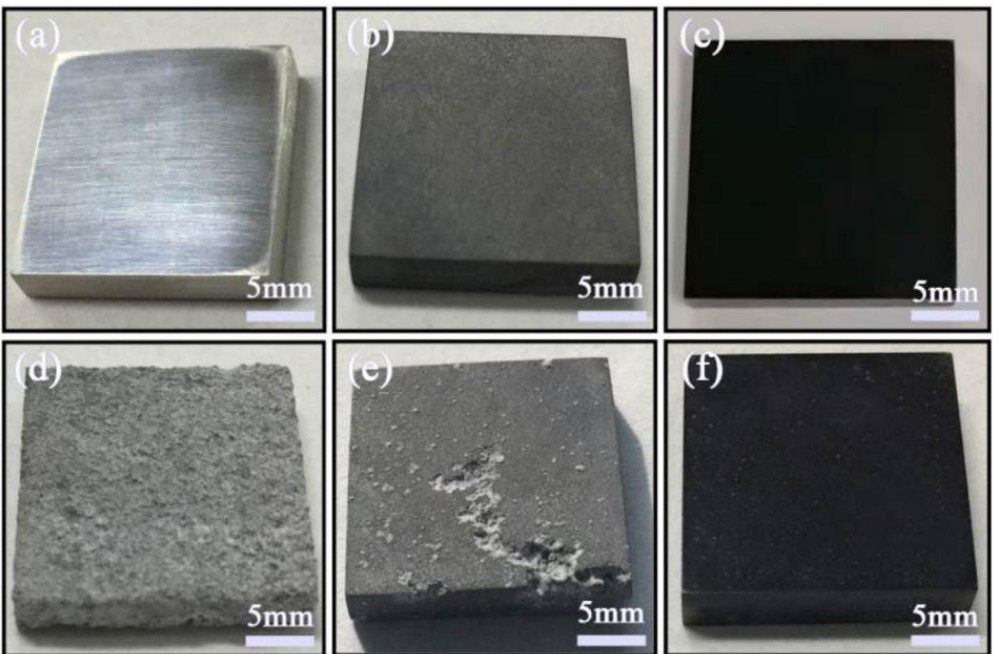

**Figure 50.** Photographs of (**a**,**d**) bare AZ31, (**b**,**e**) AZ31 + FLMs and (**c**,**f**) AZ31 + Y + FLMs before and after being immersed in 3.5 wt.% NaCl solution for 15 days. Reprinted with permission from [86]. Copyright 2021 Elsevier.

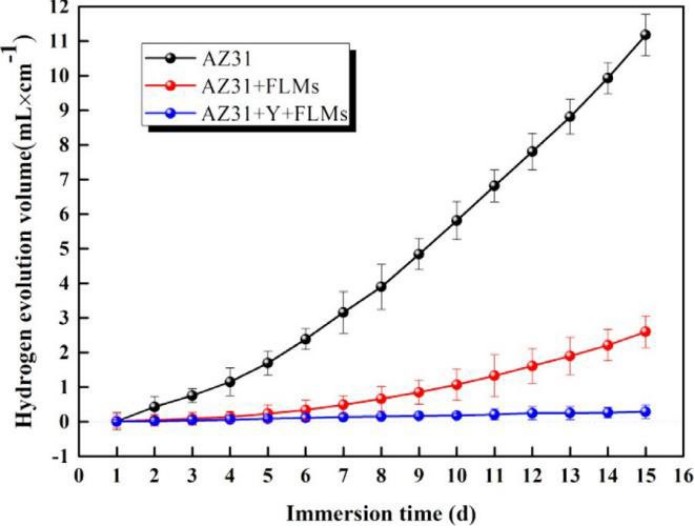

**Figure 51.** Images of hydrogen evolution volume for bare AZ31, AZ231 + FLMs and AZ31 + Y + FLMs before and after being immersed in 3.5 wt.% NaCl solution for 15 days. Reprinted with permission from [86]. Copyright 2021 Elsevier.

## 5. Critical Overview

The electromagnetic properties are crucial for absorption measurement that transforms the electromagnetic wave into heat energy. Facile preparation of MXene and metal composite is required for the exploration of new family members of MXene. The targeted ternary composite nanomaterial could produce a more diverse interface, a larger surface area, and, most significantly, enhance electrical properties, which is critical for managing EMI shielding effectiveness through conduction and polarization loss. There was a substantial contribution in the damping of EM waves due to the competing impact among

conducting impedance and surface and interfacial polarization by the oxide nanoparticles located on the MXene surface. Mechanical flexibility and metallic conductivity are the two important factors for energy storage devices. The fabrication of MXene@Zn composite has helped researchers to achieve desirable systems. Energy efficiency requirements in different industries have been overcome by utilizing lightweight and high-performance materials. Al incorporation into MXene can effectively produce materials with high strength, hardness and fracture toughness. Mechanical properties of materials are greatly dependent on dispersion of particles as the wettability has an important role in the dispersion of particles. Excellent wettability of materials improved mechanical properties. Researchers need to design novel three-component MXene heterostructure for multifunctional applications such as supercapacitors, catalytic performance, etc. (as shown in Table 3) The catalytic performance could improve through considering bimetallic nanoparticles. The synergistic effect between Rh/Ni and Pt/Pd nanoparticles accurately adopted surface electric state of nanoparticles. MXene can be used as supportive materials for dehydrogenation. MXene and metal-based anode material can overcome challenges related to batteries because of their large electrical conductivity and significant energy density. Combination of MXene with metal composites consisted excellent capacities with long cycle stability. Metals cause corrosion when exposed to a favorable oxygen-containing environment. The issue can be resolved by applying coating agents to prevent corrosion and explore MXene with other elements, which act as anticorrosive materials. Hard ceramic nanostructures together with soft metal can remarkably upgrade the properties, such as fatigue, strength and corrosion resistance.

**Table 3.** MXene–metal composites synthesis, properties and application.

| MXene–Metal Composites | Methods | Properties | Applications | References |
|---|---|---|---|---|
| $Au/Ti_3C_2T_x$ | chemical reduction | microstructure | electrochemical and catalytic performance | [62] |
| RhNi/MXene | one-step wet chemical | microstructure | catalytic performance | [63] |
| $Ti_3C_2/DNA/Pd/Pt$ | In-situ process | | sensor and catalytic performance | [64] |
| $Ti_3C_2T_x/Ni$ | In-situ hydrothermal | EMA | electromagnetic wave absorption | [65] |
| $Ti_3C_2T_x/Al$ | pressureless sintering followed by hot extrusion | microstructure and mechanical properties | solid lubricant | [66] |
| $Ti_3C_2@Au@CdS$ | self reduction | microstructure | photocatalytic hydrogen production activity | [67] |
| FLM/Al composite | self assembly protocol and powder metallurgy | microstructure, mechanical properties | automotive, aerospace, packaging industries | [68] |
| $Ag-Ti_3C_2T_x$ and $Ag-Nb_2CT_x$ | self chemical reduction | electromagnetic Interference | EM wave shielding | [69] |
| Pd@MXene | one-step soft solution processing | microstructure, surface-enhanced Raman spectroscopy | Sensors, catalysis, biomedical | [70] |
| $Ti_3C_2T_xMXene@Zn$ | facile in situ electroplating | flexibility, wettability, electronic conductivity | energy storage system | [72] |
| $Ti_3C_2T_x/Mg-Li$ | liquid metal gelation | mechanical properties | alloys, batteries and supercapacitor | [73] |

**Table 3.** *Cont.*

| MXene–Metal Composites | Methods | Properties | Applications | References |
|---|---|---|---|---|
| MXene/Cu | high energy ball milling | microstructure, mechanical | automotive and aerospace industries | [74] |
| Ni-MXene/Cu composites | high energy ball milling | microstructure, mechanical and wettability | automotive and aerospace industries | [75] |
| FeNi/Ti$_3$C$_2$T$_x$ | facile in situ hydrothermal | microstructure, magnetic and microwave absorption | Radar detection technology | [76] |
| Ag-Ti$_3$C$_2$T$_x$ and Ag-Nb$_2$CT$_x$ Composites | simultaneous self-reduction and oxidation | EMI shielding | wireless technologies and radar systems | [77] |
| MXene/Ag | direct reduction method | | lithium-ion batteries | [78] |
| Ti2C/Au-Ag | machine learning | | electrochemical and SERS intelligent analysis | [79] |
| MOF-derived MnO$_2$/Mn$_3$O$_4$ and Ti$_3$C$_2$ MXene/Au | enzymatic inhibition | | electrochemical pesticides detection | [80] |
| MXene@Sb | one-step electrodeposition approach | flexible | catalyst, batteries, sensors | [81] |
| Lac/Au/MXene/GCE | reduction process | | Electrochemical detection of catechol | [82] |
| MXene-Ag$_{0.9}$Ti$_{0.1}$ | self reduction | | electrocatalytic activity | [83] |
| MXene@AuNPs | self reduction | | catalytic performance | [84] |
| Ni/MoO$_2$@Mo$_2$CT$_x$ | wet impregnation method | | catalytic performance | [85] |
| MXene/MgAl-LDHs | in situ synthesis | | anticorrosion | [86] |

## 6. Summary

In this paper, we focused on composites of MXene and metal. Due to their prodigious physical and chemical properties, MXene-based metal composites have gained much attention. MXene/metal composites showed excellent results for synthesis preparation, showed properties such as microstructure, mechanical, thermal stability, and wettability, as well as boosted their wide range of applications in energy storage devices, catalytic activity, supercapacitors, and anti-corrosive and electrochemical performance. Synthesis of the bimetallic MXene complex via the one-step chemical approach has enabled the possible fabrication of other metals with MXene. MXene–metal composites have been demonstrated as efficient electromagnetic absorption materials that might be effective in the application of radar networks and wireless automations. The composite of MXene with aluminum showed excellent mechanical behavior and reduced frictional losses. The chemical stability investigation has led to the synthesis of composite at definite process states. This preliminary research suggests that MXene-reinforced MMCs with significantly better mechanical characteristics could be developed. It is normal to believe that improving MXene amount, size of particle, dispersion, and alloy composition will increase mechanical characteristics even further. Beyond excessive EMI shielding materials, the gained large conductivity and synthesis of ternary hybrid nanostructure offer promise for significances in energy storage, photocatalysis, and multifunctional importance. A bifunctional nanosensor has come up

with the latest approach for food and agro-product safety. Furthermore, diabetes mellitus has been detected by using a suitable electrode as a GOx/Au/MXene/Nafion/GCE biosensor to determine the amount of glucose in biological specimens. The manufacturing of sensors based on MXene nanocomposite has unlocked its application in the biomedical field. Nanocomposite based on MXene could efficiently apprehend the disintegration of solar water. More efforts have been made on MXene@Metal composite to fabricate a dendrite-free, metal-based storage cell as well as potassium ion devices. Moreover, the manufacturing of functional nanocomposite extends MXene–metal composite for proceeding implementation in structural alloys as well as batteries and supercapacitors. MXene/Ag composite proved to be a promising electrode material for batteries as well as possess better electrocatalytic activity in alkaline fuel. The development of $Ni/MoO_2@Mo_2CT_x$ catalyst has overcome the issues of transportation and can be employed as fuel in cars. In addition, the detection of methamidophos utilizing composite materials has opened more opportunities in the field of electrochemical sensors for examining different environmental contaminants such as pesticides and other harmful chemicals. Besides, MXene also unbarred routes for its utilization as an anticorrosive agent.

There are limitations when dealing with biosensor, such as the surface termination of fluorine in MXene is not favorable under various conditions. Therefore, it is suggested that the promising method might be adopted to remove such functional groups while constructing biosensor, which is advantageous for specific biomedical applications. The challenges associated with the agglomeration of particles required special attention, which affects the performance of catalysts. Using MXene composite as a catalyst is helpful to stop the agglomeration of the particles, improved surface area and created more active sites. It is notable that the controlled loading of the composite should be utilized, which otherwise might lead to reducing the exposed active sites, causing a decrease in degradation efficiency. It is worth noting that a catalyst should possess control surface morphology and dispersion, which is needed for improving hydrogen evolution reaction performance. Future work should focus on the design and synthesis methods, which should be investigated to obtain bimetallic nanoparticles and ternary nanocomposite materials for achieving the best results in properties and applications. The main focus should be on strength, flexibility, crack formation, durability, dispersion, hydrophilicity, current density, charge, transfer, cyclic stability, power density, catalytic activity and anticorrosion enhancement in different fields for enabling the composites of MXene and metal that require a strong bond between them for better electrical contacts. MXene/metal composite requires special attention to deal with redox reactions because the exposed metal sites on MXene are more likely to undergo redox reactions. Presently, more than 30 MXene have developed from MAX phase materials, but still a lot of MAX phases are not investigated for the synthesis of new MXene materials. To date, very few MXene members have been explored as a composite with metal. Some of their synthesized composites focus on properties, while others discuss applications in various fields. More research is required to develop new MXene from the MAX phase via exiting synthesis methods as well as investigate new methods and explore the new MXene materials with metal composite for their outstanding results. Limited research has been conducted on MXene and metal composite for exploring anticorrosion materials. It is highly recommended for researchers to synthesize novel MXene and investigate with metal composite for a wide range of applications.

**Author Contributions:** Q.F. and S.G.; data curation, L.D.; writing—original draft preparation, M.U.K., S.F., D.W. and Y.B.; writing—review and editing, C.H.; visualization, M.U.K.; supervision, C.H.; project administration, C.H.; funding acquisition, C.H. All authors have read and agreed to the published version of the manuscript.

**Funding:** This review article was supported by China Scholarship Council, the Natural Sciences Foundation of China (52032011 and 52072311), Outstanding Young Scientific and Technical Talents in Sichuan Province (2019JDJQ0009), the Fundamental Research Funds for the Central Universities (2682020ZT61, 2682021GF013, XJ2021KJZK042), the Opening Project of State Key Laboratory of

Green Building Materials, and the Project of State Key Laboratory of Environment-Friendly Energy Materials (20kfhg17).

**Institutional Review Board Statement:** Not applicable.

**Informed Consent Statement:** Not applicable.

**Data Availability Statement:** Data sharing is not applicable to this article.

**Conflicts of Interest:** The authors declare no conflict of interest.

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
