# Peer review of "Preparations and Applications of MXene–Metal Composites: A Review"

_coatings, doi:10.3390/coatings12040516_

Round 1
Reviewer 1 Report
Journal Coatings (ISSN 2079-6412) Manuscript ID coatings-1638059 Type Review Title Preparations and Applications of MXene-Metal Composites: A ReviewAuthors of this review article illustrated and explained the development of MXene based metal composites (preparations and applications). Also, highlighted the synthesis techniques utilized for preparation of MXene composites with metal in enhancing the properties of the composites suiting for wide range of application as an electrode substance for energy storage devices, electrochemical cells, supercapacitors, catalytic and anti-corrosive performance.
Adequate number of figures, tables were provided,
Most of the literature covered.
Except for polishing English
Conclusions are good enough
Article can be accepted for publication in Coatings Journal
Author Response
I am grateful to you for such appreciation regarding this review paper. This paper has been checked for grammar mistakes and been polished.
Reviewer 2 Report
- Add more current references for a review.
- Highlight better in introduction which is the novelty of the work?
- Remake the introduction, the chemical attack aspects must be moved.
- Must be redivided into subchapters: microstructure, mechanical properties, electromagnetic adsorption and wettability.
- Make a figure with MXene and metal composites discussed in the paper.
- Make a table in which to extract the main features of each example discussed to easily see all the differences.
- Complete the conclusions with the limitations of the proposed methodology. Also write future research.
- Generally, the quality of the writing could be improved.
Author Response
I really appreciate the way you provide guidance to improve this paper. I have edited more informative data in introduction along with recent references. Subchapters are also considered, and general figure has been included to better understand process of MXene-metal composites. Each composite is mentioned in new table with its methods, properties and applications. Conclusion has been edited limitations and future prospects as well. Recommendations provided by you helped me a lot to know new ways in the research and gain more knowledge. Further improvements would be highly appreciable.
Reviewer 3 Report
Authors have focused on detailed review knowledge over MXene and metal composites which will be helpful in further research progress in the
improvement of composites. This study unlocked the synthesis of MXene and metal composite and their improvement in properties. Also, this study presented the application of MXane and metal composite such as electrochemical performance, catalytic activity and corrosive performance. Authors should incorporate substantial changes to the provided manuscript in order to reach the acceptable publication standard. Please find comments as below:
- Abstract should clarify the new contribution to the subject in the field to distinguish from previous literature. Please express the best findings of this review study which will be helpful for the potential authors. Future scope/recommendations are required. The adapted methodology is unclear.
- Introduction: Authors should provide the detailed critical discussion to elucidate the existent research gap which has motivated them to conduct this review study. Advantages/disadvantages are necessary to be included. Some important recent literature are missing as below:
- 10.1039/C9NH00571D
https://doi.org/10.1016/j.est.2019.101115
https://doi.org/10.1016/j.ceramint.2019.12.257
https://doi.org/10.1016/j.solener.2020.07.060
https://doi.org/10.1016/j.solmat.2020.110754
https://doi.org/10.1021/acsami.8b21893
https://doi.org/10.1021/acsnano.7b08895 - Novelty/originality is written poorly and should be rewritten carefully.
- Quality of figures should be improved significantly.
- Advantages/disadvantages of various techniques along with the their performance should be included.
- Titles are some figures are written wrongly which is weird. For instance, figure 16 is related to UV-Vis analysis figure but the title explains the XRD analysis or Figure 17 provides the XRD analysis results but title is about FTIR. This mistake is severely unacceptable and all figures along with their titles should be checked again carefully!
- Results should be supported with valid info.
- Some of the sub topics only represents the literature without consideration of scientific analysis which should be amended.
- Conclusion should be restructures with main focus on the best findings of this review study.
- Comprehensive proof read is essential throughout the manuscript as there are a lot of typo/grammatical errors.
Author Response
I am very thankful to you for providing such valuable suggestions on this paper. Abstract and introduction have been edited with detailed information. The missing literatures mentioned are considered and added. Figures 16 and 17 are added correctly, and you may refer to this article (Satheeshkumar E, Makaryan T, Melikyan A, et al. One-step solution processing of Ag, Au and Pd@MXene hybrids for SERS[J]. Scientific reports, 2016, 6 (1): 1-9.). Quality of figures was also adjusted. Subchapters, MXene-metal composite figure, and table are added for better understanding by following reviewer’ comments. Grammar mistakes have been removed. Many thanks for your hard work again.
Round 2
Reviewer 2 Report
Overall, the work has been improved.
Author Response
Dear Reviewer,
Many thanks for your hard work and supports. The manuscript has been polished.
Best wishes
Reviewer 3 Report
The revised version of manuscript has been improved properly.
Author Response

(The authors gave the same response as above.)
